# PASO: Step Parallel Stochastic Optimization

**Jianrong Lu** [1]  **Zhuoya Gu** [1]  **Haobo Li** [1]  **Zhiyu Zhu** [2]  **Yechao Zhang** [3]  **Jianhai Chen** [1]  **Minghui Yang** [4]
**Junwei Liu** [5]  **Jian Wang** [4]  **Qinming He** [1]  **Hui Liu** [6]  **Junhui Hou** [7]

## Abstract

This paper approaches the fundamental challenge of accelerating the inherently autoregressive nature of gradient descent (GD) like SGD and Adam through a dynamic system perspective. Specifically, we introduce a unified framework that recasts the autoregressive GD process as solving a system of triangular nonlinear equations (TNEs), thereby enabling *step-parallel* training, where gradients for different GD steps are computed concurrently without sequential dependencies. Within this generic framework, we establish that: (1) the TNE system admits a unique solution corresponding precisely to the autoregressive GD iterative trajectory; (2) solving the TNEs system guarantees convergence to the GD iterative trajectory in at most the equal iterations. Building on these insights, we present *PASO*, the first step-parallel optimizer for accelerating a broad class of GD-based optimizers like SGD and Adam. Extensive experiments (*e.g.*, Llama-3.2-1B and diffusion model) validate that PASO achieves up to **21×** reduction in GD steps and **4.5×** speedup in wall-clock time, with no model quality loss. Source code is available at: https://github.com/Jianrong-Lu/PASO.git.

## 1. Introduction

Stochastic gradient descent (SGD) (Robbins & Monro, 1951) and its variants (Kingma & Ba, 2014; Duchi et al.,

[1]College of Computer Science and Technology, Zhejiang University, Hangzhou, China; [2]City University of Hong Kong (Dongguan), Dongguan, China; [3]College of Computing and Data Science (CCDS), Nanyang Technological University, Singapore; [4]Ant Group, Hangzhou, China; [5]School of Software and Microelectronics, Peking University, Beijing, China; [6]Department of Computing and Information Sciences, Saint Francis University, Hong Kong, China; [7]Department of Computer Science, City University of Hong Kong, Hong Kong, China. Correspondence to: Jianhai Chen <chenjh919@zju.edu.cn>.

*Proceedings of the 43$^{rd}$ International Conference on Machine Learning*, Seoul, South Korea. PMLR 306, 2026. Copyright 2026 by the author(s).

2011; Tieleman & Hinton, 2017; Loshchilov & Hutter, 2017; Liu et al., 2025b; Nesterov, 1983; Pagliardini et al., 2024; Hwang, 2024), continue to be fundamental optimization engines for training deep neural networks. These methods underpin breakthroughs across domains (Vaswani et al., 2017; He et al., 2016; Shi et al., 2022; Lu et al., 2024; 2023), exemplified by large language models (LLMs) (Brown et al., 2020; OpenAI, 2023; Touvron et al., 2023) and computer vision systems (Radford et al., 2021; Rombach et al., 2022; Lu et al., 2025). At its essence, SGD operates through iterative parameter adjustments where each iterative step follows the negative gradient direction of a randomly sampled data batch. To produce high-quality models, however, SGD typically necessitates an enormous number of iterative steps involving repeated forward and backward passes through massive datasets, resulting in prolonged training durations. For instance, training modern LLMs like DeepSeek-V3 (Liu et al., 2024) and GPT-4 (OpenAI, 2023) often demands hundreds of thousands to millions of iteration steps. As a result, training a large-scale model consumes millions of GPU hours, roughly amounting to 1-3 months or longer (OpenAI, 2023; Liu et al., 2024). Therefore, the staggering SGD steps caused by the exponential growth of model and dataset sizes have made a fundamental efficiency bottleneck.

Existing efforts to accelerate SGD training follow two main paradigms. The first develops more advanced optimizers(Zhang et al., 2025; Robert et al., 2025; Cheng & Glasgow, 2025; Hwang, 2024; Duchi et al., 2011; Liu et al., 2025b; Polyak, 1964) (e.g., Adam (Kingma & Ba, 2014)) to accelerate convergence through refined update rules. However, the reduction in iterative steps is modest at best, as these techniques continue to rely on the sequential step-by-step execution of SGD. The second strategy develops parallel SGD, where workers update the model synchronously or asynchronously. Synchronous methods, typical in distributed learning, require waiting for all nodes to compute gradients before each update, while asynchronous approaches (e.g., DC-ASGD (Zheng et al., 2017) and HOGWILD! (Recht et al., 2011a) ) allow workers to update global parameters independently. However, they risk harming model performance because of stale gradients. More critically, their parallelization is confined to intra-step operations, while leaving the inter-step sequential dependency

intact, thereby maintaining the total number of training iterations unchanged.

This paper investigates a critical question: *can we drastically reduce gradient descent steps by parallelizing the step execution without sacrificing model performance?* At first glance, the challenge seems insurmountable—GD is inherently sequential, bound by a rigid Markov dependency chain (Zinkevich et al., 2010). However, we demonstrate that it is possible to completely sever this dependency chain to enable fully parallel gradient computation across different GD steps. In particular, we approach this problem through the lens of non-linear equations, treating the points on the GD iteration trajectory as mutually independent unknown variables. This independence thus naturally eradicates the sequential dependencies between iterations. By solving this system of equations, we achieve fully parallel gradient computation across all steps. Empirical results show this approach converges in far fewer iterations compared to standard sequential GD. Furthermore, our framework is inherently orthogonal to existing approaches, allowing seamless integration with both sequential optimizers (e.g., Adam) and parallel GD variants (e.g., model, pipeline, and data parallelism).

In summary, this paper contributes the following theoretical and practical advancements:

- we present PASO, the first step-parallel training paradigm without sacrificing model quality, through transforming the autoregressive GD process into solving a system of triangular nonlinear equations;
- we establish that PASO converges to the GD trajectory points with iteration counts equal to or surpassing that of the autoregressive GD;
- our comprehensive evaluation (language modeling, image classification, and diffusion model) showcases that PASO reduces iteration steps by up to $21\times$ and accelerates wall-clock time by $4.5\times$, all without sacrificing quality.

**Conflict of Interest Disclosure.** The authors declare that they have no financial conflicts of interest related to this work.

## 2. Related Work

**Model Parallelism.** Model parallelism (Jia et al., 2019; Narayanan et al., 2021; Xu et al., 2021; Yuan et al., 2021; Rajbhandari et al., 2020; Ren et al., 2021; Xu et al., 2020; Gao et al., 2025) shards network parameters across multiple devices to circumvent memory constraints. The devices execute partial computations on local parameter subsets, aggregating results via collective communication (NVIDIA, 2023) to derive global gradients. Early works (Dean et al., 2012; Chilimbi et al., 2014; Xing et al., 2015) established

basic model partitioning. Recent frameworks like Mesh-TensorFlow (Shazeer et al., 2018), FSDP (Zhao et al., 2023), and Megatron-LM (Shoeybi et al., 2019) optimize execution for large-scale models. However, these methods only address model parallelization within one GD step.

**Pipeline Parallelism.** Pipeline parallelism (He et al., 2021; Kim et al., 2020; Li et al., 2021; Sun et al., 2025; Zhao et al.; Tang et al.; Ao et al., 2025; Anonymous, 2025; Liu et al., 2025a; Ajanthan et al.) partitions models into sequential stages, processing micro-batches in an interleaved manner to reduce worker idle time. GPipe (Huang et al., 2019) ensures consistency via gradient accumulation, whereas PipeDream (Narayanan et al., 2019) optimizes throughput using "1F1B" scheduling and weight stashing. Despite these gains, pipeline parallelism preserves the sequential dependency of GD, as parameter updates are constrained by the completion of prior gradient computations.

**Data Parallelism**. Data parallelism (Nagaraju et al., 2024) partitions the training data across multiple workers (e.g., GPUs or nodes), and each worker computes gradients on its local data subset. The gradients are then aggregated to update the model parameters through two primary mechanisms: synchronous SGD (SSGD) and asynchronous SGD (ASSGD). In SSGD (Zinkevich et al., 2010; Dekel et al., 2012; 2010; Ye et al., 2022; McMahan et al., 2017), workers compute gradients in parallel, but the parameter server waits for all workers to finish before applying the aggregated gradients to the model. This ensures consistency but may suffer from stragglers. ASSGD (Baudet, 1978; Bertsekas & Tsitsiklis, 2015; Cohen et al., 2021; Recht et al., 2011b; Feyzmahdavian & Johansson, 2023; Stich et al., 2021; Nguyen et al., 2022; Even et al., 2024; Recht et al., 2011a; Zhang et al., 2015) address this limitation by enabling independent parameter updates without synchronization. A prominent example is HOGWILD! (Recht et al., 2011a), which implements lock-free updates to the shared model parameters in memory. However, ASSGD faces challenges with gradient staleness (Dutta et al., 2018).

Existing paradigms predominantly optimize intra-step parallelization, thus inheriting the limitations of autoregressive GD. In contrast, PASO disrupts this dependency chain by introducing *step-level parallelism*, enabling the simultaneous execution of multiple distinct GD steps. We believe that PASO serves as a natural complement to intra-step methods, establishing a new avenue for parallel training. We also observe that OptEx (Shu et al., 2024) leaves kernelized gradient to estimate future multiple models; however, these lead to low accuracy and limit its scalability for large-scale model training. In App. H, we show that PASO significantly outperforms OptEx both theoretically and experimentally. Besides, we notice techniques exposing concurrency across the "time" dimension in the field of differential equations

like Parareal (Lions et al., 2001; Falgout et al., 2014; Gander & Guttel, 2013) via multilevel relaxations and coarse propagators. PASO distinguishes itself by modeling the GD process as TNEs system solving via fixed-point iteration.

## 3. Proposed Method

### 3.1. Preliminary: Stochastic Gradient Descent (SGD)

Given a mini-batch $\zeta$ of size $B$, the loss function for the batch is defined as the average loss over the samples in $\zeta$:

$$\mathcal{L}(w, \zeta) = \frac{1}{B} \sum_{x,y \in \zeta} \ell(w; x, y), \qquad (1)$$

where $\ell(w; x, y)$ denotes the loss for a single sample $(x, y)$. The model parameters $w$ are updated iteratively using the gradient of the batch loss:

$$w_t = w_{t-1} - \eta_{t-1} \nabla_{w_{t-1}} \mathcal{L}(w_{t-1}, \zeta_{t-1}), \qquad (2)$$

where $\eta_t$ is the learning rate at iteration $t$, $\zeta_t$ stands for the mini-batch used at iteration $t$, and $\nabla_w \mathcal{L}(w_t, \zeta_t)$ is the gradient of the batch loss. Other popular optimizers, such as Adaptive Moment Estimation (Adam), also update parameters iteratively. For brevity, a detailed description of these methods is provided in *Appendix* M.

### 3.2. Motivation

GD algorithms, such as SGD and Adam, use historical weights to compute the current weight, which is the essence of an autoregressive process as follows.

**Definition 3.1** (Autoregressive GD Procedure). Initiating with a model weight $w_0$, the GD process, like SGD and Adam, represents an autoregressive procedure in the specific form of

$$w_t = w_0 - \sum_{\tau=0}^{t-1} \eta_\tau\, g_\tau\big(w_\tau, \ldots, w_{\tau-r+1}; \zeta_\tau, \ldots, \zeta_{\tau-r+1}\big), \quad (3)$$

where $t \in [1, T]$ and $1 \leq r \leq \tau + 1$ and the general gradient term $g_\tau$ is determined by the specific optimizer. For example, the $g_\tau$ for SGD depends only on the most recent weight and mini-batch (i.e., $r = 1$):

$$g_{t-1}(w_{t-1}; \zeta_{t-1}) = \nabla_{w_{t-1}} \mathcal{L}(w_{t-1}, \zeta_{t-1}). \qquad (4)$$

The explicit $g_{t-1}$ formulations for more complex optimizers like Adam are detailed in Appendix M.

We observe that when all the model weights $w_0, \cdots, w_T$ are considered as unknown variables, the autoregressive GD procedure above transforms into a system of $T+1$ nonlinear equations (NEs). By providing an initial set of guesses for the true weights, this system of NEs can be solved in parallel since there are no dependencies among the $T + 1$ NEs. As a result, the model weights $w_0, \cdots, w_T$ can be computed concurrently.

### 3.3. Recasting Autoregressive GD as NEs Solving

Inspired by current parallel algorithms (Song et al., 2021; Shih et al., 2024; Lu et al., 2025), such a series of cascaded functions in Definition 3.1 can be regarded as a system of $T + 1$ NEs with a triangular structure. Denote by $\hat{w}_0, \cdots, \hat{w}_T$ the unknown variables corresponding to the iterative trajectory $w_0, \cdots, w_T$ generated from the autoregressive GD process in Definition 3.1.

**Definition 3.2** (Triangular NEs). We define the system of triangular NEs for the autoregressive procedure in Definition 3.1 as $\mathcal{F}(\hat{w}_0, \cdots, \hat{w}_T)$:

$$\mathcal{F}(\cdot) = \begin{cases} \hat{w}_0 - w_0 = 0, \\ \hat{w}_t - F_{t-1}(\hat{w}_0, \cdots, \hat{w}_{t-1}; \zeta_0, \ldots, \zeta_{t-1}) = 0, t \in [1, T], \end{cases}$$
$$(5)$$

where $F_{t-1}(\hat{w}_0, \cdots, \hat{w}_{t-1}; \zeta_0, \ldots, \zeta_{t-1})$ is defined as:

$$F_{t-1}(\cdot) = \hat{w}_0 - \sum_{\tau=0}^{t-1} \eta_\tau g_\tau(\hat{w}_\tau, \ldots, \hat{w}_{\tau-r+1}; \zeta_\tau, \ldots, \zeta_{\tau-r+1}),$$
$$(6)$$

where $g_\tau$ depends on the choice of the specific GD algorithms. In Appendix M, we include the explicit form of $g_\tau$ for various GD algorithms like AdamW.

This formulation offers several advantages. First, it decouples the dependencies among $w_t$, enabling synchronous calculation for all gradients $\nabla_{w_t} \mathcal{L}(w_t, \zeta_t), t \in [0, T-1]$. Second, the triangular NEs have been extensively studied in mathematics, providing access to a variety of well-established methods for solving such systems efficiently.

While we can now calculate the gradients across steps in parallel by solving the triangular NEs, an important question remains: do the solutions found via equation solving yield model weights comparable to those generated by the autoregressive GD process? Specifically, can we assert that $\hat{w}_t = w_t$ holds for all $t \in [0, T]$?

**Proposition 3.3** (Unbiased Estimation (see *App.* N for proof)). *The TNEs system in Eq.* (5) *possesses a unique solution that unbiasedly estimates the GD trajectory* $\{w_\tau\}_{\tau=0}^{T}$ *in Definition* 3.1.

This finding demonstrates that by solving for the TNEs, a model of comparable quality to that derived from the traditional autoregressive GD process can be obtained.

### 3.4. Solving the System of TNEs

The field of optimization provides various methods for solving a system of NEs. Since our primary goal is to study a fundamental step-parallel GD optimizer, we implement only the classical fixed-point iteration (FPI) method (Banach, 1922) and postpone more advanced alternatives for future exploration. Applying FPI to find the solution of an equation system involves reformulating the equation system into an iterative form. It is easy to know the iterative form of Eq. (5) corresponds to a system with $T$

iterative components in Eq. (7). Therefore, given an initial set of guesses $\hat{w}_0^{(0)}, \cdots, \hat{w}_T^{(0)}$, and randomly sampled $T$ mini-batches $\zeta_0, \ldots, \zeta_{T-1}$, the system of FPI for the TNEs is as follows:

$$\hat{w}_t^{(k)} = F_{t-1}(\hat{w}_0^{(k-1)}, \cdots, \hat{w}_{t-1}^{(k-1)}; \zeta_0, \ldots, \zeta_{t-1}), \quad (7)$$

where $t \in [0,\ T-1]$ and $\hat{w}_0^{(k)} = w_0, \forall k \in \{0, \cdots, K\}$; $K$ is the number of parallel iterations. For a more intuitive FPI system, see Definition O.4 in Appendix.

**Proposition 3.4** (Convergence Analysis (see *App. O* for proof)). *From any initial guess $\{\hat{w}_t^{(0)}\}_{t=0}^T$, the fixed-point iteration in Eq. (7) converges exactly to the autoregressive GD trajectory $\{w_t\}_{t=0}^T$ defined in Definition 3.1. Exact convergence happens in $K \leq T$ iterations.*

The above is based on a worst-case analysis and we empirically find the number of parallel iterations $K$ is significantly smaller than $T$, resulting in substantial empirical speedups. Besides, we also provide empirical validation that the PASO trajectory is functionally equivalent to the sequential ones, with the near-zero average L2 norm and variance between them confirming a high-fidelity reproduction (*App. D.4*). Besides, in *Appendix P*, we give the convergence analysis of PASO under the sliding window.

### 3.5. Computation-efficient Subequations Solving

Solving the above triangular NEs necessitates computing $T$ gradients $\{\nabla_{\hat{w}_t} \mathcal{L}(\hat{w}_t, \zeta_t)\}_{t=0}^{T-1}$ in parallel across the entire time horizon. For large values of $T$, this becomes computationally prohibitive when restricted to a limited number of computing nodes. To tackle this, our core idea is to perform the fixed-point iteration only on $p \leq T$ subequations per iteration via a sliding window technique in (Shih et al., 2024). Specifically, we perform parallel equation solving only on a subset of $T + 1$ NEs, within a sliding window of size $p$:

$$\hat{w}_{t+i}^{(k)} = F_{t-1+i}(\hat{w}_0^{(k-1)}, \cdots, \hat{w}_{t-1+i}^{(k-1)}; \zeta_0, \ldots, \zeta_{t-1+i}), i \in [0, p-1] \quad (8)$$

This window size can be tuned to match the number of available computing nodes. Additionally, the window slides forward dynamically, with the sliding distance determined by the number of equations for which solutions have been found in the current window.

### 3.6. Stopping Criterion

Let $\delta$ represent the convergence tolerance threshold, governing the allowable variation in solution values between successive iterations. Following existing practicies (Shih et al., 2024; Zhou et al., 2024; Lu et al., 2025), we define the stopping criterion as:

$$d(\hat{w}_t^{(k)}, \hat{w}_t^{(k-1)}) := \frac{1}{n} \left\| \hat{w}_t^{(k)} - \hat{w}_t^{(k-1)} \right\|^2 \leq \delta, \quad (9)$$

where $\|\cdot\|$ denotes the Frobenius norm; $n$ is model dimension. $\delta$ is updated adaptively via exponential moving

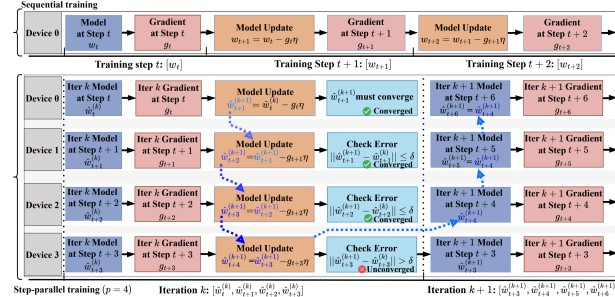

*Figure 1.* Illustration of our step-parallel training framework. PASO computes gradients in parallel within a $p$-sized window (here $p = 4$), followed by the communication of updated models. Throughout this process, each device maintains the computation graph for only a single step.

average (EMA):

$$\delta = \lambda \delta + (1 - \lambda) \cdot \mathcal{A}\Big(\big\{d(\hat{w}_{t+i}^{(k)}, \hat{w}_{t+i}^{(k-1)}) | i = 1, \cdots, p\big\}\Big), \quad (10)$$

where $\lambda$ is the EMA decay rate; $\mathcal{A}$ is a mean or median function.

### 3.7. Initialization and Complete Algorithm

The parallel iteration in Eq. (8) begins with a set of initial weights $\{\hat{w}_t^{(0)}\}_{t=0}^p$. We initialize all the model parameters using the default initial weight $w_0$: $\hat{w}_t^{(0)} = w_0, \forall t \in \{0, \cdots, p\}$. When the sliding window moves forward, any newly introduced parameter inherits the last converged weight from the window.

Alg.1 details PASO over a sliding window. After obtaining the GD update rule (Line 1) and initializing the weights $\{\hat{w}_t^{(0)}\}_{t=0}^{p-1}$ (Line 2), PASO initiates the parallel optimization loop at Line 4 in which a batch of weights $\{\hat{w}_t^{(k)}\}_{t=0}^{p-1}$ within a sliding window undergo synchronous updating. Line 5 compute the gradients, which are the basic computational units of parallelism. The updates are computed in Line 6, and Line 7 updates the current model weights, preparing for the next parallel iteration $k + 1$. Line 8 checks the variation between new weights and the current weights and then determines the stride to which the window can slide forward. Line 9 initializes new model parameters outside the window according to the sliding stride $s$. Fig. 1 visualizes the core implementation and Fig. 6 shows the pipeline.

We note that, as shown in App. A, we use step-based deterministic seeding mechanism and `DataLoader` with `Shuffle` to ensure that regardless of which worker GPU processes step $t$, or in what parallel iteration $k$ the computation occurs, the resulting mini-batch $\zeta_t$ is identical to the one sampled at step $t$ in the sequential regime.

**Algorithm 1** PASO: Step Parallel Stochastic Optimization within A Sliding Window

---

**Input :** Default model initial weight $w_0$, gradient descent steps $T$, learning rate $\{\eta_t\}_{t=0}^{T-1}$, random mini-batches $\{\zeta_t\}_{t=0}^{T-1}$, tolerance $\delta$, window size $p$, model dimension $n$, EMA decay rate $\lambda$.

1   Obtain GD update rule $g_t(\hat{w}_t, \cdots, \hat{w}_{t-r+1}; \zeta_t, \cdots, \zeta_{t-r+1})$. // E.g., Eq. (4) or Eq. (16)

2   Initialize $\{\hat{w}_t^{(0)} = w_0, t = 0, \cdots, p\}$   // Initialize $p$ model weights within the sliding window.

3   $t, k \leftarrow 0, 0; k \in [0, K]$

4   **while** $t < T$ **do**

5     $\nabla_{\hat{w}_{t+i}^{(k)}} \mathcal{L}(\hat{w}_{t+i}^{(k)}, \zeta_{t+i}), \forall i \in \{0, \cdots, p-1\}$ // Compute each gradient concurrently.

6     $g_{t+i}^{(k)}, \forall i \in \{0, \cdots, p-1\}$    // Calculate updates in parallel (e.g., via Eq. (4)).

7     $w_{t+i+1}^{(k+1)} \leftarrow \hat{w}_t^{(k)} - \sum_{j=t}^{t+i} \eta_j g_j^{(k)}, \forall i \in \{0, \cdots, p-1\}$    // Update weights at iteration $k$ via Eq. (7).

8     $s \leftarrow \min \left( \{i + 1; \hat{w}_{t+i+1}^{(k+1)} \text{ unsatisfying Eq. (9)}, \forall i \in \{0, \cdots, p-1\}\} \cup \{p\} \right)$    // The sliding stride.

9     $\hat{w}_{t+p+j}^{(k+1)} \leftarrow \hat{w}_{t+p}^{(k)}, \forall j \in \{1, \cdots, s\}$    // Initialize new model weights.

10    $\delta \leftarrow$ Eq. (10)    // Update tolerance via exponential moving average.

11    $t \leftarrow t + s, \quad k \leftarrow k + 1, \quad p \leftarrow \min(p, T - t)$

**Return:** $\hat{w}_T^{(K)}$

---

## 4. Efficiency Analysis

PASO introduces a novel *step-parallel* approach that is orthogonal to traditional parallelization paradigms. This naturally raises a key question: *under comparable computational and memory constraints, how does the speedup efficiency of PASO compare to that of conventional methods?* To this end, denote by $N$ the number of GPUs, $\alpha \triangleq t_{\text{comm}}/t_{\text{comp}}$ the communication-to-computation time ratio, and $m \triangleq T/pK$. Table 1 provides a comprehensive comparison against existing methods, indicating three main conclusions:

- **Acceptable Overhead:** The total computational cost of PASO ($mT$) is comparable to that of model and pipeline parallelism ($T$) and significantly lower than data parallelism ($NT$). This minimal overhead is a worthwhile trade-off for the performance gains.

- **Superior Speedup:** The speedup ratio of PASO is $\frac{N}{m(1+\alpha N/p)}$. Since $m \approx 1$ and $p > 1$, PASO's speedup is strictly greater than the $\frac{N}{1+\alpha N}$ achieved by other methods. This indicates that PASO can be approximately up to **p** times faster than existing parallel approaches.

- **Better Scalability:** When the window size equals the number of GPUs ($p = N$), PASO's speedup ratio simplifies to $\frac{N}{m(1+\alpha)}$. As $N$ increases, the denominator in PASO's speedup formula grows much more slowly than in other methods (where it is dominated by the $\alpha N$ term),

demonstrating PASO's superior scalability.

## 5. Experiments

### 5.1. Experiment Settings

**Datasets and Models.** We evaluate the efficacy of PASO across three distinct domains: language modeling, image classification, and diffusion models. For **language modeling**, we train a 124M-parameter GPT-2 (Radford et al., 2019) and Llama3.2-1B (Meta, 2024) on the WikiText-2 dataset (Merity et al., 2016). For **image classification**, we utilize a compact CNN (see App. D.2.1), ResNet50 (He et al., 2016), and Vision Transformer (ViT) (Wang, 2020) trained on the CIFAR-10 dataset (Krizhevsky et al., 2009). For the **diffusion task**, we follow the training pipeline in DDPM (Ho et al., 2020). We train a UNet model at $128 \times 128$ resolution image on the LSUN Church dataset (Yu et al., 2015). Comprehensive implementation details for these tasks are provided in App. C, D, E, respectively.

**Evaluation Metrics.** For the language modeling task, we evaluate the performance using testing perplexity (PPL). For the image classification task, we evaluate the performance using top-1 testing accuracy, testing precision, testing recall, and testing F1-score. For the diffusion model, we evaluate the quality of generative images using the FID score (Heusel et al., 2017).

**Hyperparameter Settings.** All models are trained using eight NVIDIA A100 GPUs (80GB). To evaluate hyperparameter sensitivity, we conduct a sweep over the following ranges: the tolerance threshold $\delta \in [10^{-6}, 10^{-4}]$, the EMA decay rate $\lambda \in [0.8, 0.9999]$, and the adaptivity scheme $\mathcal{A} \in \{mean, median\}$. *Detailed hyperparameter configurations for language modeling, image classification, and diffusion model are summarized in Tables 8, 10, and 12.*

### 5.2. Experimental Results

**Language Modeling Task**. Table 2 demonstrates that under near-identical token consumptions, PASO significantly accelerates the convergence of various optimizers without sacrificing model performance. Across both GPT-2 and Llama-3.2-1B architectures, PASO reduces the required iteration steps for sequential methods by a factor of $10.1 \sim 20.8$, resulting in an up to $4.4\times$ improvement in wall-clock time. Such efficiency gains imply that large-scale language model training tasks could see their durations reduced from months to weeks. Notably, this acceleration is achieved while maintaining nearly identical perplexity and per-GPU peak memory compared to sequential baselines. Figs 3 illustrate the perplexity curves relative to training iterations and cumulative token consumption, respectively.

**Image Classification Task**. Table 3 presents a comparative analysis of ResNet50 and Vision Transformer (ViT) mod-

*Table 1.* Comparison of computational cost, storage, and speedup ratio across parallel training methods. The analysis shows PASO's superior speedup potential and scalability. Denote by $N$ the number of GPUs, $\alpha \triangleq t_{\text{comm}}/t_{\text{comp}}$ the communication-to-computation time ratio, and $m \triangleq T/pK \approx 1$ empirically (see Fig. 10 in *Appendix*). Detailed derivations are available in *Appendix K*.

| Method | Total Gradient Computation | Memory per Device | Speedup ($S$) | Scalability ($\lim_{N \to \infty} S$) |
|---|---|---|---|---|
| Sequential | $T$ | 1 model + 1 optimizer | 1 | 1 |
| Data Parallel | $NT$ | 1 model + 1 optimizer | $\frac{N}{1+\alpha N}$ | $1/\alpha$ |
| Model Parallel | $T$ | $\sim \frac{1}{N}$ model + 1 optimizer | $\frac{N}{1+\alpha N}$ | $1/\alpha$ |
| Pipeline Parallel | $T$ | $\sim \frac{1}{N}$ model + 1 optimizer | $\frac{N}{1+\alpha N}$ | $1/\alpha$ |
| **Step Parallel (PASO)** | $\mathbf{pK = mT}$ | **1 model + 1 optimizer** | $\frac{\mathbf{N}}{\mathbf{m(1+\alpha N/p)}}$ | $\mathbf{p}/(\mathbf{m\alpha})$ |

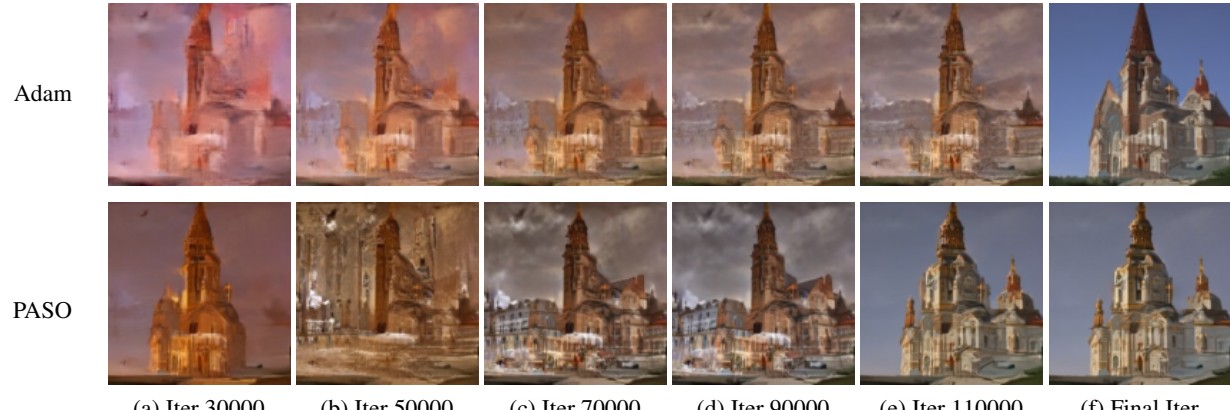

|  (a) Iter 30000 | (b) Iter 50000 | (c) Iter 70000 | (d) Iter 90000 | (e) Iter 110000 | (f) Final Iter |

*Figure 2.* Visual comparison of sampling results across different training iterations for diffusion models from Adam and PASO plus Adam. Each column represents the same training iteration. Please refer to Fig. 7 in App. E.8 for more visual comparisons.

*Table 2.* Quantitative comparisons of different methods on WiKiText-2, where models are initialized from pre-trained weights. The best results are highlighted in **bold**. "↑" (resp. "↓") means the larger (resp. smaller), the better. We define tokens as totally consumed tokens by the model. Memory (Mem.) refers to the peak GPU memory allocated during training. We report the mean and standard deviation across 5 runs. Listings 1 and 2 present the specific execution scripts required for reproducing the results. Tab. 8 gives the hyperparameter details.

| Method | GPT-2 ($B = 112, \eta = 6e-5$) | | | | | | Llama-3.2-1B ($B = 30, \eta = 1e-5$) | | | | | |
|---|---|---|---|---|---|---|---|---|---|---|---|---|
| | Iters ↓ | Mem. ↓ | Tokens ↓ | PPL ↓ | Time ↓ | Speedup ↑ | Iters ↓ | Mem. ↓ | Tokens ↓ | PPL ↓ | Time ↓ | Speedup ↑ |
| SGD | 1000 | 66.9G | 7344101 | $36.7_{\pm0.06}$ | 613s | 1.0× | 1000 | 57.8G | 2056338 | $14.4_{\pm0.11}$ | 749s | 1.0× |
| SGD+PASO | **99** (10.1×) | 66.9G | 8815938 | $36.9_{\pm0.11}$ | **176s** | **3.5×** | **72** (13.9×) | 60.1G | 2052591 | $14.4_{\pm0.19}$ | **237s** | **3.2×** |
| Adam | 1000 | 67.1G | 7344101 | $20.4_{\pm0.22}$ | 615s | 1.0× | 1000 | 60.1G | 2056338 | $11.1_{\pm0.20}$ | 767s | 1.0× |
| Adam+PASO | **48** (20.8×) | 67.3G | 7374839 | $20.7_{\pm0.07}$ | **139s** | **4.4×** | **72** (10.1×) | 60.1G | 2060908 | $11.1_{\pm0.05}$ | **254** | **3.0×** |
| AdamW | 1000 | 67.1G | 7344101 | $20.4_{\pm0.06}$ | 614s | 1.0× | 1000 | 60.1G | 2056338 | $11.1_{\pm0.22}$ | 773s | 1.0× |
| AdamW+PASO | **48** (20.8×) | 67.3G | 7374839 | $20.6_{\pm0.18}$ | **139s** | **4.4×** | **72** (10.1×) | 60.1G | 2060746 | $11.1_{\pm0.09}$ | **256s** | **3.0×** |

els, demonstrating that PASO effectively reduces iteration counts and wall-clock time without sacrificing model quality. A critical technical observation is that PASO achieves these accelerations while keeping the maximum peak GPU memory across all devices identical to the sequential baselines. For ResNet50, both the standard optimizers and their PASO-enhanced versions reach a peak per-GPU memory of 50,641 MB. Similarly, in ViT experiments, the maximum peak memory is maintained at approximately 17,999 MB. This parity is significant as it proves that our step-parallel implementation does not increase the local memory pressure on individual GPUs, making it a viable solution for large-scale model training. Fig 3 shows the accuracy curve.

**Diffusion Model Training Task**. Tab. 4 demonstrates that PASO reaches a comparable FID of 9.55 in just 136k iterations (2.46 days), whereas the baseline requires 600k iterations (7.39 days) to achieve an FID of 9.08. This substantial reduction in training time is achieved without any additional memory overhead per GPU, maintaining the same peak memory of 78.6G, which highlights the efficiency and practicality of the method for large-scale generative tasks. Figs. 3 and 2 give the FID curve and the visual comparisons.

**Step Parallelism vs. Data Parallelism.** Table 5 compares our proposed Step Parallelism method (PASO) against conventional Data Parallelism (DP). To ensure a fair comparison, given that our current PASO academic prototype lacks

*Table 3.* Quantitative comparisons of different methods on CIFAR-10. We report the mean and standard deviation over 5 runs with different random seeds. The models are trained from scratch. Listings 5 and 6 present the specific execution scripts required for reproducing the results. Tab. 10 gives the hyperparameter details.

| Method | ResNet50, $B = 2048, \eta = 5e-5$, cosine decay, warmup steps 10k, $p = 7, \delta = 0.1, \lambda = 0.999$ | | | | | | | |
|---|---|---|---|---|---|---|---|---|
| | Iters↓ | Accuracy ↑ | F1-score↑ | Precision↑ | Recall↑ | Peak memory ↓ | Time (s) ↓ | Speedup↑ |
| SGD | 240000 | $83.7_{\pm 0.21}$ | $83.7_{\pm 0.26}$ | $83.7_{\pm 0.14}$ | $83.7_{\pm 0.12}$ | 50550MB | 152909.6 ■ | 1.0× |
| SGD + PASO | **38398** (6.3×) | $83.7_{\pm 0.15}$ | $83.7_{\pm 0.23}$ | $83.7_{\pm 0.17}$ | $83.7_{\pm 0.19}$ | 50550MB | **45780.7** ■ | **3.3×** |
| Adam | 100000 | $90.1_{\pm 0.11}$ | $90.1_{\pm 0.15}$ | $90.1_{\pm 0.12}$ | $90.1_{\pm 0.13}$ | 50641MB | 62316.0 ■ | 1.0× |
| Adam + PASO | **14914** (6.7×) | $89.5_{\pm 0.25}$ | $89.5_{\pm 0.22}$ | $89.5_{\pm 0.24}$ | $89.5_{\pm 0.23}$ | 50641MB | **17771.6** ■ | **3.5×** |
| AdamW | 100000 | $90.3_{\pm 0.17}$ | $90.3_{\pm 0.16}$ | $90.3_{\pm 0.18}$ | $90.3_{\pm 0.17}$ | 50641MB | 62201.0 ■ | 1.0× |
| AdamW + PASO | **14921** (6.7×) | $89.8_{\pm 0.19}$ | $89.8_{\pm 0.21}$ | $89.8_{\pm 0.18}$ | $89.8_{\pm 0.20}$ | 50641MB | **17875.9** ■ | **3.5×** |
| Method | ViT, $B = 2048, \eta = 1.5e-4$, cosine decay, warmup steps 10k, $p = 7, \delta = 0.1, \lambda = 0.999$ | | | | | | | |
| | Iters↓ | Accuracy ↑ | F1-score↑ | Precision↑ | Recall↑ | Peak memory↓ | Time (s) ↓ | Speedup↑ |
| SGD | 240000 | $71.7_{\pm 0.25}$ | $71.7_{\pm 0.14}$ | $71.7_{\pm 0.20}$ | $71.7_{\pm 0.22}$ | 17962MB | 145489.0 ■ | 1.0× |
| SGD + PASO | **43111** (5.6×) | $71.5_{\pm 0.21}$ | $71.5_{\pm 0.18}$ | $71.5_{\pm 0.26}$ | $71.5_{\pm 0.20}$ | 17962MB | **40911.3** ■ | **3.6×** |
| Adam | 100000 | $77.7_{\pm 0.15}$ | $77.5_{\pm 0.12}$ | $77.5_{\pm 0.14}$ | $77.7_{\pm 0.13}$ | 17999MB | 59709.2 ■ | 1.0× |
| Adam + PASO | **14999** (6.7×) | $77.9_{\pm 0.20}$ | $77.8_{\pm 0.18}$ | $77.8_{\pm 0.19}$ | $77.9_{\pm 0.21}$ | 17999MB | **13958.1** ■ | **4.3×** |
| AdamW | 100000 | $78.0_{\pm 0.10}$ | $77.9_{\pm 0.11}$ | $77.9_{\pm 0.10}$ | $78.0_{\pm 0.10}$ | 17999MB | 59876.7 ■ | 1.0× |
| AdamW + PASO | **15019** (6.7×) | $78.4_{\pm 0.14}$ | $78.3_{\pm 0.13}$ | $78.2_{\pm 0.15}$ | $78.4_{\pm 0.14}$ | 17999MB | **13830.9** ■ | **4.2×** |

(a) LLM task, Llama3.2-1B  (b) Classification task, ViT  (c) Diffusion task, UNet  (d) Perplexity w.r.t. tokens on Adam  (e) Perplexity w.r.t. tokens on AdamW

*Figure 3.* The comparison of perplexity, FID, and accuracy curves. Note that Figs. (d) and (e) give comparison of perplexity curves w.r.t. the cumulative token consumption on Llama 3.2-1B over WiKiText-2.

*Table 4.* Quantitative comparison on diffusion model where we train a UNet model from scratch on the LSUN Church dataset and generate 50k images for FID computation. Listing 7 and 8 present the training scripts and FID evaluation scripts for reproducing the results. Tab. 12 gives the hyperparameter details.

| Method | Iters | Peak Mem. | FID | Time (days) | Speedup |
|---|---|---|---|---|---|
| Adam | 600k | 78.6G | 9.08 | 7.39 | 1× |
| Adam+PASO | 136k | 78.6G | 9.55 | 2.46 | 3.0× |

specific low-level optimizations, we implemented a corresponding unoptimized DP baseline sharing identical computation and communication routines. Additionally, we benchmark PASO against industry-grade DP frameworks, specifically PyTorch `DistributedDataParallel` (DDP) and `DeepSpeed`, both of which leverage highly optimized execution and communication kernels.

As shown in Table 5, PASO achieves a $2.75\times$ speedup over the unoptimized DP baseline (139s vs. 381s) under identical execution and communication implementations as well as almost equivalent token consumption. Remarkably, even without specialized execution and communication optimizations, PASO matches the performance of highly optimized indus-

try standards such as DDP and DeepSpeed ($4.4\times$ speedup). This highlights the inherent architectural advantages of *step parallelism*. Furthermore, PASO reduces the total number of iteration steps by a factor of $21\times$. This suggests that integrating advanced kernel-level optimizations from existing industrial frameworks (e.g., ZeRO (Rajbhandari et al., 2020) in DeepSpeed) into PASO could theoretically push the acceleration ratio toward the $21\times$ upper bound as communication and computation overheads are minimized. Crucially, PASO's *inter-step* parallelism complements existing *intra-step* parallel methods (data/model/pipeline parallelism). Their complementary nature suggests that combining them could yield significantly greater acceleration. We note that the lower PPL achieved by DeepSpeed is primarily attributed to the superior numerical stability and precision of its industry-standard FusedAdam kernels and ZeRO-3 optimization framework.

**Impact of the Window Size** $p$. Tab. 6 illustrates the influence of the window size $p$ on the speedup of AdamW. As $p$ increases, the number of iterations needed for convergence significantly drops, from 143 to 48, yielding a step reduction ranging from $7.0\times$ to $21\times$. This suggests that we can achieve up to $21\times$ walk-clock time acceleration without

*Table 5.* Comparison of step parallelism (PASO) vs. data parallelism (DP) on GPT-2 (WikiText-2). Hyperparameter are detailed in Tab. 13 (App. F), and execution scripts are provided in Listing 9.

| Method | Iters | Tokens | PPL | Time (s) | Speedup |
|---|---|---|---|---|---|
| Adam | 1k | 7344k | 20.4 | 614 | 1.0× |
| PASO (Academic Prototype) [a] | 48 (21×) | 7374k | 20.6 | 139 | 4.4× |
| DP (Academic Prototype) [a] | 1k (1×) | 7349k | 20.5 | 381 | 1.6× |
| DP (PyTorch DDP) [b] | 1k (1×) | 7349k | 20.5 | 140 | 4.4× |
| DP (DeepSpeed) [b] | 1k (1×) | 7349k | 19.2 | 138 | 4.4× |

[a] Unoptimized implementation sharing identical execution and comm. methods; [b] Industry-grade optimization.

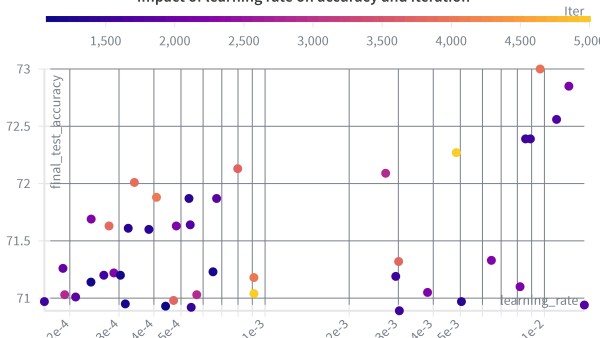

*Figure 4.* Impact of learning rate on accuracy and iterations on CNN model over CIFRA-10. The total steps are 10000. Darker points indicate faster convergence.

loss of model quality. However, given our limited resource of 8 GPUs, increasing $p$ inadvertently elevates the computational load on each device, causing the wall-clock speedup to plateau rather than scale proportionally with $p$. We believe that with more computing nodes, the time speedup of PASO could be substantially unleashed.

*Table 6.* Impact of $p$ on perplexity (PPL). Listing 3 presents the execution scripts for reproducing the results. Tab. 8 gives the hyperparameter details.

| | Iters | Peak Mem. | Tokens | PPL | Time (s) | Speedup |
|---|---|---|---|---|---|---|
| AdamW | 1000 | 67.1G | 7344101 | 20.4 | 614 | 1× |
| $p = 7$ | 143 | 67.1G | 7367159 | 20.4 | 152 | 4.0× |
| $p = 10$ | 100 | 67.1G | 7368243 | 20.5 | 171 | 3.6× |
| $p = 14$ | 72 | 67.1G | 7407890 | 20.3 | 140 | 4.4× |
| $p = 17$ | 59 | 67.3G | 7328407 | 20.8 | 157 | 3.9× |
| $p = 21$ | 48 | 67.3G | 7374839 | 20.6 | 139 | 4.4× |

**Impact of Batch Size**. As shown in Tab. 7 on GPT-2 model with learning rate $6e - 5$, the speedup of PASO becomes more significant as the batch size increases, while maintaining performance comparable to the baseline. This is because a larger batch size allows for better utilization of the GPU's computing capabilities for large-scale matrix operations, thereby improving parallel efficiency.

**Impact of Learning Rate**. Figure 4 illustrates that PASO performs robustly across a practical range of learning rates from $4 \times 10^{-4}$ to $1 \times 10^{-2}$. These rates are comparable

to those in standard optimizers (e.g., Adam's default of $1 \times 10^{-3}$), demonstrating that the model can converge effectively without additional limitations on learning rates.

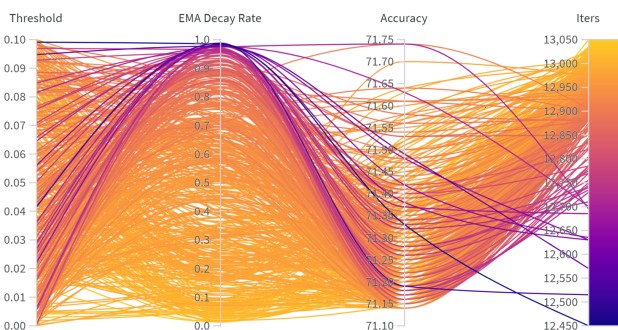

*Figure 5.* The impact of $\delta$ and $\lambda$ over CIFAR-10 by running 1200 experiments. We use PASO with $p = 7$ to accelerate Adam with 60000 steps. Darker lines indicate runs with fewer iterations.

**Impact of Tolerance $\delta$ and EMA Decay $\lambda$.** Fig. 5 demonstrates PASO's robustness to variations in $\delta$ and $\lambda$. Specifically, $\delta \in [10^{-7}, 0.1]$ and $\lambda \in [0.9, 0.9999]$ consistently yield significant iteration reductions with comparable model quality, indicating insensitivity to these hyperparameters. More detailed analysis on PASO's hyperparameters are shown in App. J.

### 5.3. Limitations and Discussion

While PASO achieves $3\times$–$4.5\times$ runtime acceleration, it remains significantly below its $6 \times -21\times$ step-level speedup. This performance gap is driven by two factors: (1) constrained GPU resources that inherently restrict acceleration, and (2) our current preliminary academic prototype implementation, which suffers from inefficient CPU-mediated inter-GPU communication (e.g., requiring gradient transfer to pass through a CPU intermediary), as well as redundant and low-efficient computations.

We note that these limitations are engineering artifacts of the research prototype rather than fundamental flaws in PASO. Future improvements could substantially increase efficiency

*Table 7.* Impact of batch size ($B$) on perplexity (PPL). Top to bottom: $B = 52, 72, 92, 112$. Listing 4 shows the execution scripts for reproduction. Tab. 8 gives the hyperparameter details.

| Method | Iters | Peak Mem. | Tokens | PPL | Time (s) | Speedup |
|---|---|---|---|---|---|---|
| AdamW | 1000 | 31.6G | 3420290 | 21.2 | 294 | 1× |
| AdamW+PASO | 48 | 32.0G | 3428529 | 21.5 | 92 | 3.2× |
| AdamW | 1000 | 43.4G | 4725181 | 20.6 | 400 | 1× |
| AdamW+PASO | 48 | 43.9G | 4775172 | 20.9 | 106 | 3.8× |
| AdamW | 1000 | 55.3G | 6020913 | 20.4 | 507 | 1× |
| AdamW+PASO | 48 | 55.8G | 6038468 | 20.7 | 122 | 4.1× |
| AdamW | 1000 | 67.1G | 7344101 | 20.6 | 614 | 1× |
| AdamW+PASO | 48 | 67.3G | 7374839 | 20.6 | 139 | **4.4×** |

through (1) more advanced algorithms (Anderson, 1965; Walker & Ni, 2011; Pulay, 1980) like Anderson acceleration, (2) system-level optimizations, such as highly optimized execution and communication kernels in DeepSpeed, and (3) combination with existing intra-step parallel training methods. These advancements could position PASO as a promising paradigm for more efficient parallel training, with broader implications for large model training.

## 6. Conclusion

We presented PASO, the first step-parallel framework that accelerates training by modeling the autoregressive process as a TNE system solving. We theoretically demonstrated that PASO maintains the exact trajectory of sequential stochastic optimization while potentially accelerating convergence. Experiments demonstrate that PASO achieves up to $4.4\times$ speedup while maintaining model quality.

## Acknowledgements

This work was supported by the National Natural Science Foundation of China under Grant 62422118, the Key R&D Program of Zhejiang Province under Grant 2025C01084, and the Hong Kong UGC under grants UGC/FDS11/E03/24, UGC/FDS11/E03/25.

### ETHICS&REPRODUCIBILITY STATEMENTS

**Ethics Statement.** This work presents a fundamental methodology for accelerating stochastic optimization algorithms. To the best of our knowledge, it does not raise any immediate ethical concerns. The research is theoretical and empirical in nature, based on mathematical analysis and standard benchmark tasks. We do not employ any private or sensitive data. However, we acknowledge that any optimization technology has the potential for dual use. We encourage the community to utilize this work responsibly.

**Reproducibility Statement.** We are committed to fostering reproducible research. To this end:

- The theoretical claims in this paper, including the uniqueness of the solution to the triangular nonlinear equations and the convergence guarantees, are supported by formal proofs provided in appendix.

- The empirical results are obtained using standard datasets and benchmarks. To ensure reproducibility, we haved open-source the complete source code of the PASO framework, including scripts for reproducing the experiments.

- The code package include detailed documentation, instructions for setting up the computational environment, and scripts to replicate the reported speedup and performance comparisons against sequential baselines.

- All hyperparameters and experimental settings are explicitly documented in the paper.

We believe these measures will enable other researchers to verify our findings and build upon this work.

## Impact Statement

This paper presents work whose goal is to advance the field of machine learning. There are many potential societal consequences of our work, none of which we feel must be specifically highlighted here.

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

# Contents

## A. Detailed Implementation of the PASO Framework

The practical realization of PASO (Step-Parallel Stochastic Optimization) is designed to ensure strict functional equivalence to sequential gradient descent (GD) while maximizing hardware utilization through step-parallelism. This section details the core components: deterministic data management, and the parallel communication architecture.

### A.1. Implementation Scope and Prototypes

**Research Prototype vs. Optimal Realization.**   It should be noted that the current implementation serves as a research prototype designed primarily to validate the core theoretical framework and convergence properties of PASO. Consequently, it is not an exhaustive engineering optimization and contains certain design simplifications, such as the reliance on a single master node to orchestrate all model replicas within the sliding window.

**Minimal Resource Requirements.**   We emphasize that the PASO framework fundamentally requires each device to maintain a single model and its corresponding optimizer state at any given step. To explore the potential for a more decentralized and communication-efficient realization, we have initiated a preliminary implementation using the NCCL backend (see our `cv_nccl.py` module). This exploration aims to further reduce the synchronization overhead and decouple the window management from a centralized master, demonstrating the scalability of PASO for larger distributed

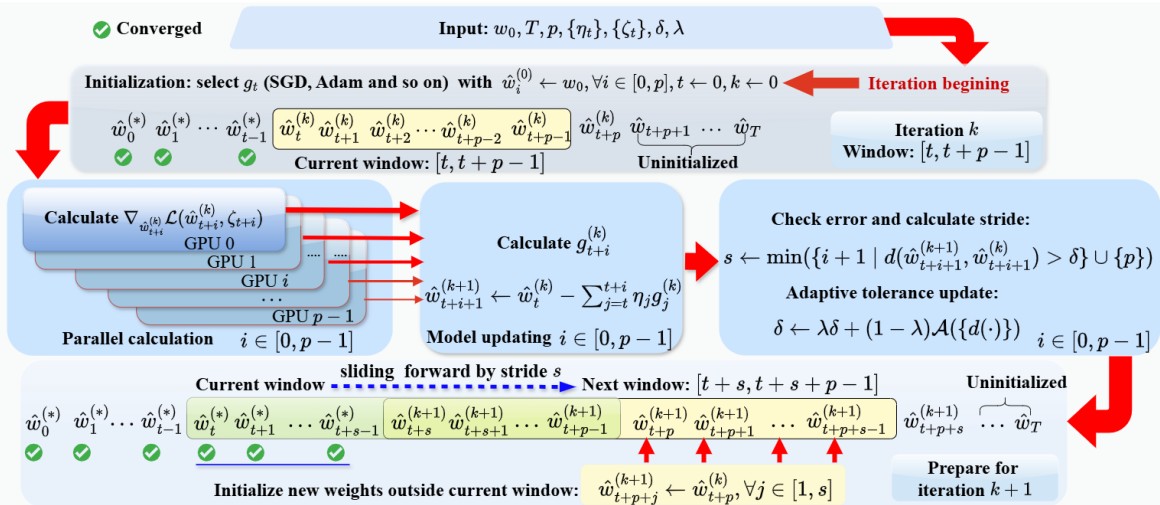

*Figure 6.* Illustration of Step-parallel Training Paradigm PASO. During iteration $k$, PASO performs simultaneous weight updates across steps within a $p$-size sliding window through parallel gradient computations. The process consists of: (1) computing update terms $g^{(k)}$ based on current weights $\hat{w}^{(k)}$ after calculating their graidients in parallel; followed by (2) determining new weights $\hat{w}^{(k+1)}$.

systems.

### A.2. Deterministic Data Sampling and Allocation

To guarantee that the parallel training trajectory is identical to the sequential baseline, PASO employs a *Step-Based Deterministic Seeding* mechanism.

**Identity via Seeding.** In standard sequential training, a mini-batch $\zeta_t$ is sampled at step $t$ following a stochastic trajectory. In our implementation, for any given step $t$ within a sliding window of size $p$, the corresponding mini-batch $\zeta_t$ is determined by an explicit seed function:

$$Seed_t = \text{Global\_Seed} + t \tag{11}$$

As shown in our implementation (`take_step()`), the system resets the `numpy` and `torch` random seeds using $Seed_t$ immediately before fetching data from the `DataLoader`. This ensures that regardless of which worker GPU processes step $t$, or in what parallel iteration $k$ the computation occurs, the resulting mini-batch $\zeta_t$ is identical to the one sampled at step $t$ in a sequential regime.

### A.3. Resource and Memory Efficiency

To maintain a low memory footprint, PASO shards the computational load. While the sliding window considers $p$ steps, each worker GPU only hosts the model parameters and a single mini-batch relevant to its assigned step. By utilizing `pin_memory=True` and `persistent_workers=True`, the implementation minimizes data transfer latency between the CPU host and GPU devices. The peak memory requirement per individual GPU remains equivalent to that of a sequential baseline, making PASO scalable to large-scale models without requiring proportional increases in per-card VRAM.

## B. Illustration of Step-parallel Training Paradigm PASO

Fig. 6 gives the illustration of PASO. During iteration $k$, PASO performs simultaneous weight updates across steps within a $p$-size sliding window through parallel gradient computations. The process consists of: (1) computing update terms $g^{(k)}$ based on current weights $\hat{w}^{(k)}$ after calculating their graidients in parallel; followed by (2) determining new weights $\hat{w}^{(k+1)}$.

# C. Experiment Details for LLM Tasks

This section provides a technical specification of the experimental environment, dataset configurations, and the exact execution parameters used for the Large Language Model (LLM) tasks.

## C.1. Dataset and Tokenization

The **WikiText-2** dataset, a widely recognized benchmark in the field of natural language processing, is employed for all language modeling evaluations in this study. Derived from a curated collection of verified "Good" and "Featured" articles on Wikipedia, it comprises approximately million tokens with a vocabulary size of unique words, ensuring a high level of lexical diversity and linguistic complexity compared to earlier datasets like Penn Treebank. The dataset is utilized under a causal language modeling (CLM) objective, where the primary task is to predict the probability of a succeeding token given its preceding context. By preserving the original punctuation, casing, and multi-paragraph structure of the source long-form articles, WikiText-2 serves as a rigorous testbed for evaluating a model's ability to capture long-range dependencies and maintain semantic coherence across extended sequences.

### C.1.1. TOKENIZER AND PREPROCESSING

- **GPT-2 Configuration:** We use the pre-trained GPT-2 tokenizer. To facilitate batch training, the `eos_token` is utilized as the `pad_token`.

- **Llama-3.2-1B Configuration:** We utilize the tiktoken-based tokenizer associated with the Llama-3 architecture.

- **Sequence Formatting:** All text is tokenized and truncated to a maximum sequence length of 512 tokens. During training, labels are generated by shifting the input sequences. Padding tokens are assigned a label value of $-100$, ensuring they are bypassed by the Cross-Entropy loss function.

### C.1.2. DATA LOADING

For optimal throughput, the `DataLoader` is configured with `pin_memory=True` and `persistent_workers=True`, utilizing 8 worker threads per process. In the parallel paradigm of PASO, updates are computed across a sliding window of size $p$. If `pin_memory` and `persistent_workers` are disabled, the system incurs a significant initialization overhead at every iteration due to the repeated creation of worker threads. Given the frequent synchronization required in parallel step computation, these latencies would exceed the computational gains, effectively nullifying the efficiency provided by the parallel window.

## C.2. Metrics and Evaluation

- **Perplexity (PPL):** The standard evaluation metric for language models, computed as $\exp(\mathcal{L})$, where $\mathcal{L}$ is the average negative log-likelihood.

- **Token Throughput:** We define total tokens as the cumulative count of non-padding tokens processed during training.

- **Peak Memory:** Reported as the maximum GPU memory allocated during the training window.

## C.3. Execution Commands of Quantitative Comparison in Table 2

The primary results in Table 2 are derived from the following execution commands.

### C.3.1. GPT-2 EXPERIMENTS

The commands below initialize the GPT-2 model with pre-trained weights for 1,000 training steps.

*Listing 1.* GPT-2 Serial and Parallel (PASO) Execution Commands for Reproducing GPT-2 Results in Table 2

```
# Serial Adam/AdamW Baseline
python llm.py --local_model_dir ./gpt2 --modelscope_name "gpt2" --max_steps 1000
    ↪ --batch_size 112 --learning_rate 6e-5 --threshold 0.1 --ema_decay 0.999
    ↪ --training_mode serial --P 21 --optimizer_type adam/adamw --lr_scheduler_type
    ↪ constant
```

```
# Parallel Adam/AdamW (PASO)
python llm.py --local_model_dir ./gpt2 --modelscope_name "gpt2" --max_steps 1000
    ↪ --batch_size 112 --learning_rate 6e-5 --threshold 0.1 --ema_decay 0.999
    ↪ --training_mode parallel --P 21 --optimizer_type adam/adamw --lr_scheduler_type
    ↪ constant

# Serial SGD Baseline
python llm.py --modelscope_name "gpt2" --max_steps 1000 --batch_size 112
    ↪ --learning_rate 6e-5 --threshold 1e-7 --ema_decay 0.9 --training_mode parallel
    ↪ --P 12 --optimizer_type sgd --lr_scheduler_type constant
# Parallel Adam (PASO)
python llm.py --modelscope_name "gpt2" --max_steps 1000 --batch_size 112
    ↪ --learning_rate 6e-5 --threshold 1e-7 --ema_decay 0.9 --training_mode parallel
    ↪ --P 12 --optimizer_type sgd --lr_scheduler_type constant
```

### C.3.2. LLAMA-3.2-1B EXPERIMENTS

For the 1B-parameter Llama model, the commands are as follows.

*Listing 2.* Llama-3.2-1B Execution Commands for Reproducing Llama-3.2-1B Results in Table 2

```
# Parallel AdamW/Adam/SGD (PASO)
python llm.py --local_model_dir "./Llama-3.2-1B" --modelscope_name "Llama-3.2-1B"
    ↪ --max_steps 1000 --batch_size 30 --learning_rate 1e-5 --threshold 0.1 --ema_decay
    ↪ 0.999 --training_mode parallel --P 14 --optimizer_type adamw --lr_scheduler_type
    ↪ cosine
python llm.py --local_model_dir "./Llama-3.2-1B" --modelscope_name "Llama-3.2-1B"
    ↪ --max_steps 1000 --batch_size 30 --learning_rate 1e-5 --threshold 0.1 --ema_decay
    ↪ 0.999 --training_mode parallel --P 14 --optimizer_type adam --lr_scheduler_type
    ↪ cosine
python llm.py --local_model_dir "./Llama-3.2-1B" --modelscope_name "Llama-3.2-1B"
    ↪ --max_steps 1000 --batch_size 30 --learning_rate 1e-5 --threshold 0.1 --ema_decay
    ↪ 0.999 --training_mode parallel --P 14 --optimizer_type sgd --lr_scheduler_type
    ↪ cosine

# Serial AdamW/Adam/SGD Baseline
python llm.py --local_model_dir "./Llama-3.2-1B" --modelscope_name "Llama-3.2-1B"
    ↪ --max_steps 1000 --batch_size 30 --learning_rate 1e-5 --threshold 0.1 --ema_decay
    ↪ 0.999 --training_mode serial --P 14 --optimizer_type adamw --lr_scheduler_type
    ↪ cosine
python llm.py --local_model_dir "./Llama-3.2-1B" --modelscope_name "Llama-3.2-1B"
    ↪ --max_steps 1000 --batch_size 30 --learning_rate 1e-5 --threshold 0.1 --ema_decay
    ↪ 0.999 --training_mode serial --P 14 --optimizer_type adam --lr_scheduler_type
    ↪ cosine
python llm.py --local_model_dir "./Llama-3.2-1B" --modelscope_name "Llama-3.2-1B"
    ↪ --max_steps 1000 --batch_size 30 --learning_rate 1e-5 --threshold 0.1 --ema_decay
    ↪ 0.999 --training_mode serial --P 14 --optimizer_type sgd --lr_scheduler_type
    ↪ cosine
```

### C.4. Execution Commands for Impact of Window Size $p$ in Table 6

To reproduce the results in Table 6, we vary the parallel window size $p$ while keeping other parameters constant.

*Listing 3.* Command for Window Size (p) Sensitivity Analysis for Reproducing Results in Table 6

```
# Parallel AdamW (PASO)
python llm.py --local_model_dir "./gpt2" --modelscope_name "gpt2" --max_steps 1000
    ↪ --batch_size 112 --learning_rate 6e-5 --threshold 0.1 --ema_decay 0.999
    ↪ --training_mode parallel --P 21 --optimizer_type adamw --lr_scheduler_type
    ↪ constant
python llm.py --local_model_dir "./gpt2" --modelscope_name "gpt2" --max_steps 1000
    ↪ --batch_size 112 --learning_rate 6e-5 --threshold 0.1 --ema_decay 0.999
```

```
    ↪ --training_mode parallel --P 17 --optimizer_type adamw --lr_scheduler_type
    ↪ constant
python llm.py --local_model_dir "./gpt2" --modelscope_name "gpt2" --max_steps 1000
    ↪ --batch_size 112 --learning_rate 6e-5 --threshold 0.1 --ema_decay 0.999
    ↪ --training_mode parallel --P 14 --optimizer_type adamw --lr_scheduler_type
    ↪ constant
python llm.py --local_model_dir "./gpt2" --modelscope_name "gpt2" --max_steps 1000
    ↪ --batch_size 112 --learning_rate 6e-5 --threshold 0.1 --ema_decay 0.999
    ↪ --training_mode parallel --P 10 --optimizer_type adamw --lr_scheduler_type
    ↪ constant
python llm.py --local_model_dir "./gpt2" --modelscope_name "gpt2" --max_steps 1000
    ↪ --batch_size 112 --learning_rate 6e-5 --threshold 0.1 --ema_decay 0.999
    ↪ --training_mode parallel --P 7 --optimizer_type adamw --lr_scheduler_type constant
# Serial AdamW Baseline
python llm.py --local_model_dir "./gpt2" --modelscope_name "gpt2" --max_steps 1000
    ↪ --batch_size 112 --learning_rate 6e-5 --threshold 0.1 --ema_decay 0.999
    ↪ --training_mode serial --P 21 --optimizer_type adamw --lr_scheduler_type constant
python llm.py --local_model_dir "./gpt2" --modelscope_name "gpt2" --max_steps 1000
    ↪ --batch_size 112 --learning_rate 6e-5 --threshold 0.1 --ema_decay 0.999
    ↪ --training_mode serial --P 17 --optimizer_type adamw --lr_scheduler_type constant
python llm.py --local_model_dir "./gpt2" --modelscope_name "gpt2" --max_steps 1000
    ↪ --batch_size 112 --learning_rate 6e-5 --threshold 0.1 --ema_decay 0.999
    ↪ --training_mode serial --P 14 --optimizer_type adamw --lr_scheduler_type constant
python llm.py --local_model_dir "./gpt2" --modelscope_name "gpt2" --max_steps 1000
    ↪ --batch_size 112 --learning_rate 6e-5 --threshold 0.1 --ema_decay 0.999
    ↪ --training_mode serial --P 10 --optimizer_type adamw --lr_scheduler_type constant
python llm.py --local_model_dir "./gpt2" --modelscope_name "gpt2" --max_steps 1000
    ↪ --batch_size 112 --learning_rate 6e-5 --threshold 0.1 --ema_decay 0.999
    ↪ --training_mode serial --P 7 --optimizer_type adamw --lr_scheduler_type constant
```

## C.5. Execution Commands of Impact of Batch Size $B$ in Table 7

The batch size scalability results in Table 7 can be reproduced by the following scripts:

*Listing 4.* Command for Batch Size (B) Scalability Analysis for Reproducing Results in Table 7

```
# Parallel AdamW (PASO)
python llm.py --local_model_dir "./gpt2" --modelscope_name "gpt2" --max_steps 1000
    ↪ --batch_size 112 --learning_rate 6e-5 --threshold 0.1 --ema_decay 0.999
    ↪ --training_mode parallel --P 21 --optimizer_type adamw --lr_scheduler_type
    ↪ constant
python llm.py --local_model_dir "./gpt2" --modelscope_name "gpt2" --max_steps 1000
    ↪ --batch_size 92 --learning_rate 6e-5 --threshold 0.1 --ema_decay 0.999
    ↪ --training_mode parallel --P 21 --optimizer_type adamw --lr_scheduler_type
    ↪ constant
python llm.py --local_model_dir "./gpt2" --modelscope_name "gpt2" --max_steps 1000
    ↪ --batch_size 72 --learning_rate 6e-5 --threshold 0.1 --ema_decay 0.999
    ↪ --training_mode parallel --P 21 --optimizer_type adamw --lr_scheduler_type
    ↪ constant
python llm.py --local_model_dir "./gpt2" --modelscope_name "gpt2" --max_steps 1000
    ↪ --batch_size 52 --learning_rate 6e-5 --threshold 0.1 --ema_decay 0.999
    ↪ --training_mode parallel --P 21 --optimizer_type adamw --lr_scheduler_type
    ↪ constant
# Serial AdamW Baseline
python llm.py --local_model_dir "./gpt2" --modelscope_name "gpt2" --max_steps 1000
    ↪ --batch_size 112 --learning_rate 6e-5 --threshold 0.1 --ema_decay 0.999
    ↪ --training_mode serial --P 25 --optimizer_type adamw --lr_scheduler_type constant
python llm.py --local_model_dir "./gpt2" --modelscope_name "gpt2" --max_steps 1000
    ↪ --batch_size 92 --learning_rate 6e-5 --threshold 0.1 --ema_decay 0.999
    ↪ --training_mode serial --P 25 --optimizer_type adamw --lr_scheduler_type constant
python llm.py --local_model_dir "./gpt2" --modelscope_name "gpt2" --max_steps 1000
    ↪ --batch_size 72 --learning_rate 6e-5 --threshold 0.1 --ema_decay 0.999
    ↪ --training_mode serial --P 25 --optimizer_type adamw --lr_scheduler_type constant
python llm.py --local_model_dir "./gpt2" --modelscope_name "gpt2" --max_steps 1000
```

```
↪ --batch_size 52 --learning_rate 6e-5 --threshold 0.1 --ema_decay 0.999
↪ --training_mode serial --P 25 --optimizer_type adamw --lr_scheduler_type constant
```

### C.6. Hyperparameter Settings for Language Modeling tasks.

Table 8 provides a unified summary of the hyperparameters applied across all Large Language Model experiments.

*Table 8.* Hyperparameter Settings for Language Modeling tasks.

| Hyperparameter | GPT-2 | Llama-3.2-1B |
|---|---|---|
| Max Steps ($T$) | 1,000 | 1,000 |
| Sequence Length | 512 | 512 |
| Learning Rate ($\eta$) | $6 \times 10^{-5}$ | $1 \times 10^{-5}$ |
| LR Scheduler | Constant | Cosine Annealing |
| SGD Momentum | $\beta = 0.9$ | $\beta = 0.9$ |
| Adam Momentum | $\beta_1 = 0.9, \beta_2 = 0.999$ | $\beta_1 = 0.9, \beta_2 = 0.999$ |
| AdamW Momentum | $\beta_1 = 0.9, \beta_2 = 0.999$ | $\beta_1 = 0.9, \beta_2 = 0.999$ |
| SGD Weight Decay | 0 | 0 |
| Adam Weight Decay | 0 | 0 |
| AdamW Weight Decay | 0.01 | 0.01 |
| Warmup Steps | 100 | 100 |
| Data Precision | bfloat16 | bfloat16 |
| Batch Size ($B$) | 112 | 30 |
| DataLoader Workers | 8 | 8 |
| Pin Memory | True | True |
| DataLoader Workers Persistent | True | True |
| **PASO Hyperparameters for SGD** | **Values** | |
| Window Size ($p$) | 12 | 14 |
| Threshold ($\delta$) | $1 \times 10^{-7}$ | 0.1 |
| EMA Decay ($\lambda$) | 0.9 | 0.999 |
| $\delta$ Adaptive Updating Method | Mean | Mean |
| **PASO Hyperparameters for Adam and AdamW** | **Values** | |
| Window Size ($p$) | 21 | 14 |
| Threshold ($\delta$) | 0.1 | 0.1 |
| EMA Decay ($\lambda$) | 0.999 | 0.999 |
| Adaptivity Method | Mean | Mean |

## D. Experiment Details for Image Classification Tasks

In this section, we provide the detailed configurations for the image classification experiments conducted on the CIFAR-10 dataset. Our experiments evaluate the performance and efficiency of different optimization strategies across two representative architectures: Vision Transformers (ViT) and ResNet50.

### D.1. Dataset and Data Augmentation

We utilize the **CIFAR-10** dataset, a foundational benchmark in computer vision consisting of color images distributed uniformly across mutually exclusive classes, including categories such as airplanes, automobiles, birds, cats, deer, dogs, frogs, horses, ships, and trucks. The dataset is partitioned into a training set of images and a test set of images, providing a balanced environment for evaluating multiclass image classification algorithms. To improve model robustness and prevent overfitting, we apply the following data augmentation and preprocessing pipeline during the training phase:

- **Random Cropping**: Images are padded by 4 pixels on each side and then randomly cropped to .

- **Random Horizontal Flip**: Images are flipped horizontally with a probability of 0.5.

- **Normalization**: All input images are normalized using the CIFAR-10 dataset-specific mean and standard deviation .

## D.2. Model Architectures

We employ two distinct types of architectures to validate the generalizability of our training approach:

### D.2.1. CNN MODEL

We evaluate our approach by training a compact Convolutional Neural Network (CNN), following a standard architecture with convolutional and pooling layers, as shown in Table 9.

*Table 9.* CNN Architecture

| Layer Type | Parameter Configuration |
|---|---|
| Conv2D | Input channels 3, output channels 32, kernel size 3×3, padding 1 |
| ReLU | Activation function |
| MaxPool2D | Pooling kernel 2×2, stride 2 |
| Conv2D | Input channels 32, output channels 64, kernel size 3×3, padding 1 |
| ReLU | Activation function |
| MaxPool2D | Pooling kernel 2×2, stride 2 |
| Flatten | Flatten to 64×8×8 vector |
| Linear | Input dimension 4096 (64×8×8), output dimension 128 |
| ReLU | Activation function |
| Linear | Input dimension 128, output dimension 10 |

### D.2.2. VISION TRANSFORMER (ViT)

The ViT model is configured for input resolution. To adapt the architecture for small-scale image data, we use a patch size of . The specific structural parameters are:

- **Embedding Dimension**: 512

- **Depth (Number of Layers)**: 6

- **Attention Heads**: 8

- **MLP Dimension**: 512

### D.2.3. RESNET50 WITH GROUP NORMALIZATION (RESNET50-GN)

For the convolutional neural network baseline, we use the ResNet50 architecture. However, we modify the standard architecture by replacing all Batch Normalization (BN) layers with Group Normalization (GN) using 32 groups. This modification is motivated by the fact that BN relies on batch-wise statistics, which can introduce instability and performance degradation in certain parallel training regimes or large-batch settings. GN provides a more stable alternative by normalizing across groups of channels, making the training process more robust to batch size variations and synchronization noise.

## D.3. Hyperparameter Settings

All models are trained for a maximum of 100,000 steps using a large batch size of 2048 to facilitate efficient hardware utilization. We use a Cosine Annealing learning rate scheduler with a warmup phase of 10,000 steps.

The detailed hyperparameter settings for both ViT and ResNet50 are summarized in Table 10.

*Table 10.* Hyperparameter settings for Image Classification tasks on CIFAR-10. All experiments are implemented using Pytorch. Our publicly available code provides all details.

| Hyperparameter | CNN | Vision Transformer (ViT) | ResNet50 |
|---|---|---|---|
| Max Steps ($T$) | 600,00 | 100,000 (240k for SGD) | 100,000 (240k for SGD) |
| Batch Size ($B$) | 4096 | 2,048 | 2,048 |
| Learning Rate ($\eta$) | $1.0 \times 10^{-3}$ | $1.5 \times 10^{-4}$ | $5.0 \times 10^{-5}$ |
| Warmup Steps | 0 | 10,000 | 10,000 |
| LR Scheduler | Constant | Cosine Annealing | Cosine Annealing |
| Normalization | BatchNorm | LayerNorm | GroupNorm ($G = 32$) |
| SGD Momentum | $\beta = 0.9$ | $\beta = 0.9$ | $\beta = 0.9$ |
| Adam Momentum | $\beta_1 = 0.9, \beta_2 = 0.999$ | $\beta_1 = 0.9, \beta_2 = 0.999$ | $\beta_1 = 0.9, \beta_2 = 0.999$ |
| AdamW Momentum | $\beta_1 = 0.9, \beta_2 = 0.999$ | $\beta_1 = 0.9, \beta_2 = 0.999$ | $\beta_1 = 0.9, \beta_2 = 0.999$ |
| SGD Weight Decay | 0 | 0 | 0 |
| Adam Weight Decay | 0 | 0 | 0 |
| AdamW Weight Decay | 0.01 | 0.01 | 0.01 |
| Loss Type | Cross Entropy | Cross Entropy | Cross Entropy |
| Validation Interval | 100 Steps | 100 Steps | 100 Steps |
| **PASO Settings for SGD** | **Values** | | |
| Window Size ($p$) | 7 | 7 | |
| Tolerance Threshold ($\delta$) | 0.01 | 0.1 | |
| EMA Decay ($\lambda$) | 0.9 | 0.999 | |
| $\delta$ update method | median | median | |
| **PASO Settings for Adam/AdamW** | **Values** | | |
| Window Size ($p$) | 7 | 7 | |
| Tolerance Threshold ($\delta$) | 0.01 | 0.1 | |
| EMA Decay ($\lambda$) | 0.999 | 0.999 | |
| $\delta$ update method | median | median | |

## D.4. Empirical Validation of Optimization Trajectory Equivalence

The objective of this study is to verify that the PASO algorithm faithfully reproduces the optimization path of various standard sequential optimizers. The experimental design is as follows:

- **Model and Task:** We train a CNN on the CIFAR-10 dataset. The total iterations is 10000 GD steps.

- **Optimizer Comparison:** For each base optimizer (SGD, Adam, AdamW), we conduct two parallel training procedures: one using the standard sequential optimizer and another using its PASO-enhanced version. Critically, all training runs commence from an identical set of randomly initialized weights, learning rate, and batch size to ensure a fair comparison.

- **Evaluation Metric:** At each training step $t$, we compute the squared L2 norm of the difference between the model weight vectors produced by the two methods, defined as:

$$d^t = ||w_{\text{PASO}}^t - w_{\text{Sequential}}^t||^2$$

where $w_{\text{PASO}}^t$ and $w_{\text{Sequential}}^t$ represent the model weights obtained by the PASO variant and its standard sequential counterpart at step $t$, respectively. This metric, $d^t$, quantifies the instantaneous deviation between the two optimization trajectories in the parameter space. To provide a comprehensive assessment, we report the mean and variance of $d^t$ across the entire training process for each optimizer.

**Results and Analysis.** The statistical summary of the trajectory divergence $d^t$ for all optimizers over the complete training process is presented in Table 11.

The results demonstrate that for all three optimizers, the mean and variance of the divergence $d^t$ remain exceptionally small throughout the training process. The consistently minimal values across all optimizers empirically confirm that PASO

*Table 11.* Statistical summary of trajectory divergence ($d^t$) between PASO and sequential optimizers.

| Optimizer | Mean of $d^t$ | Variance of $d^t$ |
|---|---|---|
| SGD and SGD + PASO | $3.14 \times 10^{-6}$ | $6.71 \times 10^{-12}$ |
| Adam and Adam + PASO | $3.56 \times 10^{-3}$ | $5.75 \times 10^{-6}$ |
| AdamW and AdamW + PASO | $3.38 \times 10^{-3}$ | $4.70 \times 10^{-6}$ |

faithfully reproduces the optimization trajectory of the standard sequential optimizer, regardless of the specific optimization algorithm employed. This high-fidelity replication ensures that the convergence properties and final solution quality of the original optimizer are preserved. Note that while the average L2 norm for Adam and AdamW appear larger than SGD's, they remain highly insignificant when considered in context. For a model with millions of parameters, an average squared L2 norm difference on the order of $10^{-3}$ corresponds to an extremely small per-parameter discrepancy. For example, for a model with $n \approx 5 \times 10^6$ parameters, this corresponds to a *root mean squared error (RMSE)* per parameter of approximately $\sqrt{3.56 \times 10^{-3}/5 \times 10^6} \approx 2.7 \times 10^{-5}$. Consequently, the trajectories of PASO and sequential optimizers are functionally equivalent.

### D.5. Execution Commands of Quantitative Comparison in Table 3

The primary results in Table 3 are derived from the following execution commands.

#### D.5.1. RESNET-50 EXPERIMENTS

The commands below train the ResNet-50 model from scratch for 100,000 training steps.

*Listing 5.* ResNet-50 Serial and Parallel (PASO) Execution Commands for Reproducing ResNet-50 Results in Table 3

```
# Serial Adam Baseline
python cv.py --dataset cifar10 --model_name resnet50 --max_steps 100000 --batch_size
    ↪ 2048 --learning_rate 0.00005 --threshold 0.1 --ema_decay 0.999 --training_mode
    ↪ serial --P 7 --optimizer_type adam --lr_scheduler_type cosine --warmup_steps 10000
# Serial AdamW Baseline
python cv.py --dataset cifar10 --model_name resnet50 --max_steps 100000 --batch_size
    ↪ 2048 --learning_rate 0.00005 --threshold 0.1 --ema_decay 0.999 --training_mode
    ↪ serial --P 7 --optimizer_type adamw --lr_scheduler_type cosine --warmup_steps
    ↪ 10000
# Serial SGD Baseline
python cv.py --dataset cifar10 --model_name resnet50 --max_steps 240000 --batch_size
    ↪ 2048 --learning_rate 0.00005 --threshold 0.1 --ema_decay 0.999 --training_mode
    ↪ serial --P 7 --optimizer_type sgd --lr_scheduler_type cosine --warmup_steps 10000

# Parallel Adam (PASO)
python cv.py --dataset cifar10 --model_name resnet50 --max_steps 100000 --batch_size
    ↪ 2048 --learning_rate 0.00005 --threshold 0.1 --ema_decay 0.999 --training_mode
    ↪ parallel --P 7 --optimizer_type adam --lr_scheduler_type cosine --warmup_steps
    ↪ 10000
# Parallel AdamW (PASO)
python cv.py --dataset cifar10 --model_name resnet50 --max_steps 100000 --batch_size
    ↪ 2048 --learning_rate 0.00005 --threshold 0.1 --ema_decay 0.999 --training_mode
    ↪ parallel --P 7 --optimizer_type adamw --lr_scheduler_type cosine --warmup_steps
    ↪ 10000
# Parallel SGD (PASO)
python cv.py --dataset cifar10 --model_name resnet50 --max_steps 240000 --batch_size
    ↪ 2048 --learning_rate 0.00005 --threshold 0.1 --ema_decay 0.999 --training_mode
    ↪ parallel --P 7 --optimizer_type sgd --lr_scheduler_type cosine --warmup_steps
    ↪ 10000
```

#### D.5.2. VIT EXPERIMENTS

For the ViT model, which is trained from scratch, the commands are as follows.

*Listing 6.* ViT Serial and Parallel (PASO) Execution Commands for Reproducing ViT Results in Table 3

```
# Serial Adam Baseline
python cv.py --dataset cifar10 --model_name resnet50 --max_steps 100000 --batch_size
    ↪ 2048 --learning_rate 0.00005 --threshold 0.1 --ema_decay 0.999 --training_mode
    ↪ serial --P 7 --optimizer_type adam --lr_scheduler_type cosine --warmup_steps 10000
# Serial AdamW Baseline
python cv.py --dataset cifar10 --model_name resnet50 --max_steps 100000 --batch_size
    ↪ 2048 --learning_rate 0.00005 --threshold 0.1 --ema_decay 0.999 --training_mode
    ↪ serial --P 7 --optimizer_type adamw --lr_scheduler_type cosine --warmup_steps
    ↪ 10000
# Serial SGD Baseline
python cv.py --dataset cifar10 --model_name resnet50 --max_steps 240000 --batch_size
    ↪ 2048 --learning_rate 0.00005 --threshold 0.1 --ema_decay 0.999 --training_mode
    ↪ serial --P 7 --optimizer_type sgd --lr_scheduler_type cosine --warmup_steps 10000

# Parallel Adam (PASO)
python cv.py --dataset cifar10 --model_name resnet50 --max_steps 100000 --batch_size
    ↪ 2048 --learning_rate 0.00005 --threshold 0.1 --ema_decay 0.999 --training_mode
    ↪ parallel --P 7 --optimizer_type adam --lr_scheduler_type cosine --warmup_steps
    ↪ 10000
# Parallel AdamW (PASO)
python cv.py --dataset cifar10 --model_name resnet50 --max_steps 100000 --batch_size
    ↪ 2048 --learning_rate 0.00005 --threshold 0.1 --ema_decay 0.999 --training_mode
    ↪ parallel --P 7 --optimizer_type adamw --lr_scheduler_type cosine --warmup_steps
    ↪ 10000
# Parallel SGD (PASO)
python cv.py --dataset cifar10 --model_name resnet50 --max_steps 240000 --batch_size
    ↪ 2048 --learning_rate 0.00005 --threshold 0.1 --ema_decay 0.999 --training_mode
    ↪ parallel --P 7 --optimizer_type sgd --lr_scheduler_type cosine --warmup_steps
    ↪ 10000
```

## E. Experiment Details for Diffusion Training Tasks

This section details the experimental configuration, dataset processing, and execution parameters for the Diffusion model training tasks, specifically comparing standard sequential training with the PASO parallel acceleration framework.

### E.1. Dataset and Data Augmentation

We utilize the **LSUN Church** (Church Outdoor) dataset for all image generation experiments. The dataset is processed to ensure high-fidelity image synthesis at a resolution of $128 \times 128$.

### E.1.1. PREPROCESSING AND "TOKENIZATION" EQUIVALENT

While diffusion models operating in pixel space do not utilize discrete tokenizers, the image processing pipeline serves as the input representation stage:

- **Image Rescaling:** All input images are resized to $128 \times 128$ pixels.

- **Normalization:** Pixel values are mapped from the range $[0, 255]$ to $[0, 1]$ and subsequently auto-normalized within the `GaussianDiffusion` module to facilitate stable gradient flow.

- **Augmentation:** We apply `RandomHorizontalFlip` and `CenterCrop` during training to enhance the diversity of the training distribution and prevent over-fitting.

### E.1.2. HIGH-THROUGHPUT DATA LOADING

The `DataLoader` is configured to maximize GPU utilization and minimize synchronization overhead between the parallel workers in the PASO window:

- **Worker Configuration:** We assign 12 worker threads per process (`num_workers=12`).

- **Memory Management:** To accelerate host-to-device transfers, `pin_memory=True` is enabled.

- **Persistence:** We set `persistent_workers=True`. In the PASO framework, since multiple steps are computed in a sliding window $P$, the persistent workers prevent the overhead of re-initializing the data pipeline at the start of every parallel rollout, which is essential for maintaining the reported $3.0\times$ speedup.

- **Shuffling:** Stochasticity is ensured by setting `shuffle=True` for the training set.

## E.2. Model Architecture and Diffusion Settings

We employ a **U-net** backbone within a **Gaussian Diffusion** framework in the source code [1] from DDPM (**?**) to train the diffusion model and compute FID. Specifically, following DDPM (**?**), our neural network architecture follows the backbone of PixelCNN++ (Salimans et al., 2017), which is a U-Net (Ronneberger et al., 2015) based on a Wide ResNet (He et al., 2016). We replaced weight normalization (Salimans & Kingma, 2016) with group normalization (Wu & He, 2018) to make the implementation simpler. All models have two convolutional residual blocks per resolution level and self-attention blocks at the $16 \times 16$ resolution between the convolutional blocks. Diffusion time $t$ is specified by adding the Transformer sinusoidal position embedding into each residual block.

### E.2.1. UNET HYPERPARAMETERS

The UNet architecture follows a hierarchical structure with the following specifications:

- **Base Dimension:** $64$.

- **Dimension Multipliers:** $(1, 2, 4, 4, 8, 8)$.

- **Attention Layers:** Full self-attention is integrated at the $5^{th}$ and $6^{th}$ resolution levels, while linear attention is used in intermediate blocks to balance computational cost and modeling capacity.

### E.2.2. DIFFUSION PROBABILISTIC PROCESS

- **Noise Schedule:** A linear beta schedule is used with $1,000$ diffusion timesteps for training.

- **Objective:** The model is trained under the `pred_noise` (epsilon-prediction) objective.

- **EMA:** An Exponential Moving Average (EMA) of model weights is maintained with a decay rate of $0.9999$ and updated every $10$ iterations to ensure the stability of the generated samples.

## E.3. Training Process

During the training process, following DDPM (**?**), we applied Exponential Moving Average (EMA) to the model parameters with a decay factor of $0.9999$. We configured the system to save the EMA-smoothed model every 10,000 steps; this model is subsequently used for FID calculation during the sampling phase.

## E.4. Sampling Process

During the sampling process, we generate 50,000 images and utilize the source code from DDPM (**?**) to calculate the FID score. Specifically, we employ a 100-step DDIM (Song et al., 2020) solver to sample the 50,000 images.

## E.5. Metrics and Evaluation Details

- **FID (Fréchet Inception Distance):** Quantitative evaluation is performed by calculating the FID between $50,000$ generated samples and the ground-truth LSUN Church dataset [2]. Feature extraction is performed using a pre-trained InceptionV3 network.

- **Training Efficiency:** Measured by total wall-clock time (days), iteration counts, and peak GPU memory (VRAM).

---

[1]https://github.com/hojonathanho/diffusion
[2]https://huggingface.co/datasets/tglcourse/lsun_church_train

## E.6. Execution Commands for Reproducing Table 4

The following commands were used to reproduce the quantitative results in Table 4. The experimental environment was synchronized using a fixed seed of 42.

### E.6.1. TRAINING EXECUTION

The training scripts for both sequential (Adam) and parallel (PASO) modes are provided below.

*Listing 7.* Execution scripts for Diffusion Training Tasks in Table 4

```
# Serial Training (Adam Baseline)
python diffusion.py --results_folder diffusion_v2_serial --dataset lsun_church
    ↪ --model_name unet --training_mode serial --lr_scheduler_type cosine
    ↪ --warmup_steps 10000 --save_and_sample_every 10000

# Parallel Training (Adam + PASO)
python diffusion.py --results_folder diffusion_v2_parallel --dataset lsun_church
    ↪ --model_name unet --threshold 0.000001 --paso_ema_decay 0.99 --training_mode
    ↪ parallel --P 7 --lr_scheduler_type cosine --warmup_steps 10000
    ↪ --save_and_sample_every 10000
```

### E.6.2. FID EVALUATION EXECUTION

After training checkpoints were saved at the specified milestones, FID was calculated using the following distributed sampling script:

*Listing 8.* FID Evaluation Commands in Table 4

```
# Parallel Mode FID Calculation
python sweep_fid.py --results_folder ./PASO/diffusion_v2_parallel --data_folder
    ↪ ./dataset/church_outdoor_images --milestones 10000 50000 100000 150000 200000
    ↪ 250000 300000 350000 400000 450000 500000 550000 600000 --csv_name
    ↪ parallel_fid.csv --batch_size 512 --num_samples 50000

# Serial Mode FID Calculation
python sweep_fid.py --results_folder ./PASO/diffusion_v2_serial --data_folder
    ↪ ./dataset/church_outdoor_images --milestones 10000 50000 100000 150000 200000
    ↪ 250000 300000 350000 400000 450000 500000 550000 600000 --csv_name serial_fid.csv
    ↪ --batch_size 512 --num_samples 50000
```

## E.7. Hyperparameter Settings for Diffusion Training

Table 12 summarizes the hyperparameter configurations for both the diffusion process and the PASO acceleration framework.

## E.8. Visual Comparison for Diffusion Model

To ensure a fair assessment, we generated samples using a fixed set of 50 random seeds, creating a strict one-to-one correspondence between the outputs of Adam and Adam plus PASO.

The comprehensive visual results are presented in Figure 7. From these visualizations, two primary conclusions can be drawn:

- **Consistency in Convergence Path:** The images generated by PASO are visually nearly identical to those produced by Adam under the same random seeds. This strongly suggests that PASO introduces negligible deviation to the optimization trajectory, adhering to the convergence path of the serial Adam optimizer.

- **Preservation of Generative Quality:** The perceptual quality of the generated samples in terms of structural coherence, texture details, and overall fidelity is comparable between the two methods. This indicates that the step parallelization strategy of PASO does not compromise the generative capabilities or degrade the quality of the model.

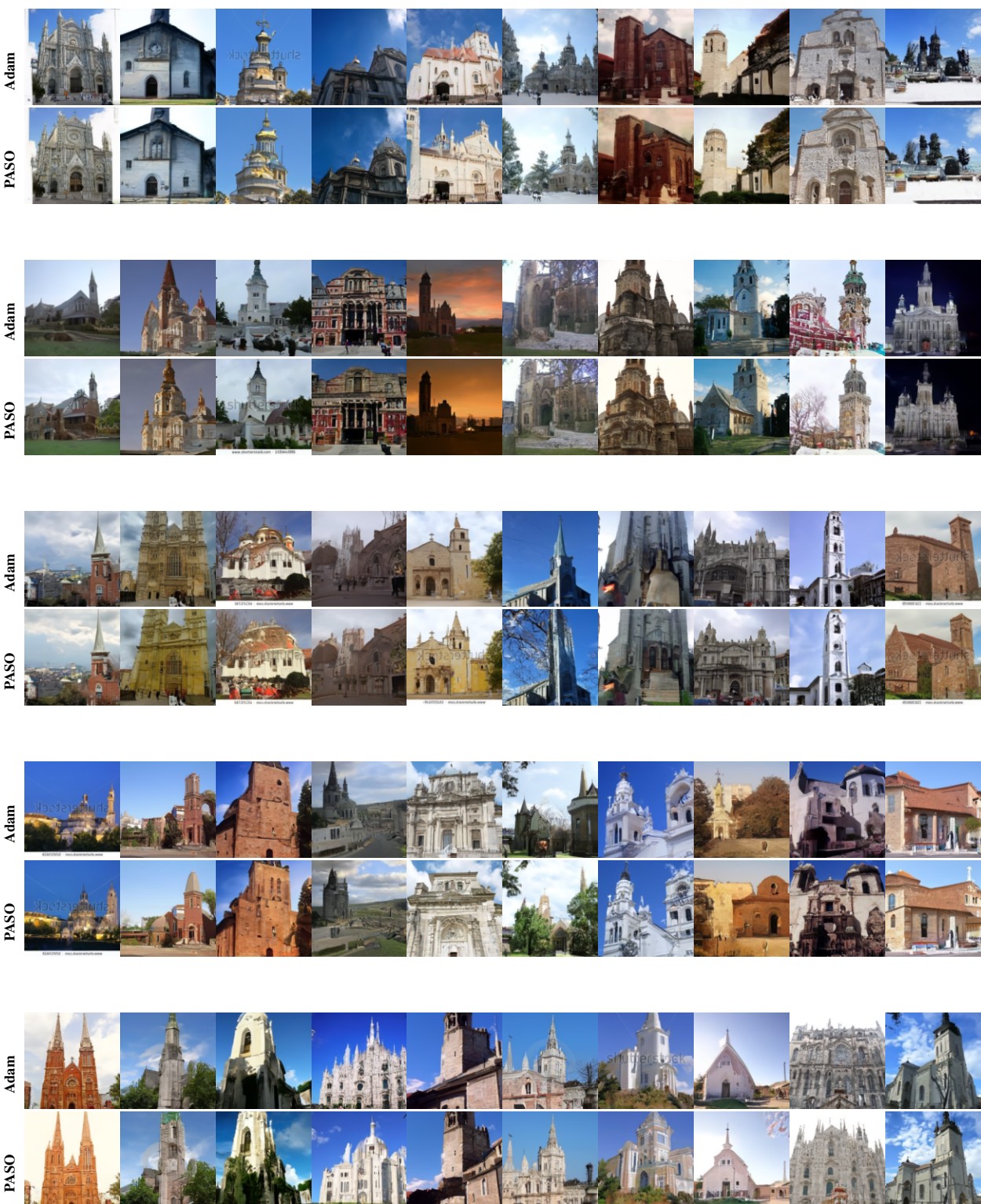

*Figure 7.* Visual comparison of generation results between **Adam** and **PASO+Adam** methods. Images are paired by identical random seeds to demonstrate consistency and difference. A total of 50 image pairs are shown across 5 blocks.

*Table 12.* Hyperparameter settings for Diffusion Training tasks in Table 4.

| Hyperparameter | Value |
|---|---|
| Max Steps ($T$) | 600,000 |
| Image Size | $128 \times 128$ |
| Batch Size ($B$) | 160 |
| Learning Rate ($\eta$) | $2 \times 10^{-5}$ |
| LR Scheduler | Cosine Annealing |
| Warmup Steps | 10,000 |
| Optimizer | Adam |
| Adam Betas | $\beta_1 = 0.9, \beta_2 = 0.99$ |
| Gradient Clipping | 1.0 |
| EMA Model Data Augmentation (Horizontal Flip) | True |
| DataLoader Workers | 12 |
| Pin Memory | True |
| Persistent Workers | True |
| UNet Architecture | Hierarchical Structure |
| UNet Base Dimension | 64 |
| UNet Dimension Multipliers | (1, 2, 4, 4, 8, 8) |
| UNet Full Attention Layers | the $5^{th}$ and $6^{th}$ resolution levels. |
| UNet Linear Attention Layers | Intermediate Blocks |
| Noise Schedule | Liner beta schedule is used with $1,000$ diffusion timesteps |
| Training Objective | Noise Prediction |
| EMA Model Decay Rate | 0.9999 |
| EMA Model Updating Interval | 10 steps |
| Sampling Solver | DDIM |
| Sampling Steps | 100 |
| **PASO Parameters** | **Values** |
| Window Size ($P$) | 7 |
| Threshold ($\delta$) | $1 \times 10^{-6}$ |
| EMA Decay ($\lambda$) | 0.99 |
| Adaptivity Type | Median |

## F. Experiment Details: Step-parallelism v.s. Data-parallelism in Table 5

This section provides the comprehensive experimental configuration used to generate the results in Table 5. All experiments were performed using 8 NVIDIA A100 GPUs (80GB) interconnected via NVLink.

**Common Configuration**   To ensure consistency across all methods (Serial Adam, PASO, DP, DDP, and DeepSpeed), we utilized the GPT-2 (124M parameters) architecture. The training was conducted on the Wikitext-2 dataset with a fixed total batch size of 112 and a maximum sequence length of 512. The optimizer was set to Adam with a constant learning rate of $6 \times 10^{-5}$.

**PASO and Naive DP Settings**   Both PASO and *DP w/o Optimization* are implemented as academic prototypes. They rely on Python `multiprocessing.Queue` for communication between the master process and worker processes. In PASO, the master GPU manages the parameter updates and synchronization for a parallel window size of $P = 21$, using an error threshold $\epsilon = 0.1$ and an EMA decay of 0.999 for step adaptivity. *DP w/o Optimization* utilizes the same queue-based architecture to aggregate gradients from 8 GPUs before performing a centralized update.

**Optimized Baseline Settings**   The *DDP* implementation uses the standard PyTorch `DistributedDataParallel` wrapper with the NCCL backend, which employs bucketed All-Reduce for efficient gradient synchronization. The *DeepSpeed* implementation utilizes the ZeRO-3 optimization stage, which partitions gradients and optimizer states across

GPUs, effectively reducing memory redundancy and overlapping communication with computation kernels.

*Table 13.* Detailed Experimental Parameters for Comparison Methods in Table 5

| Category | PASO (Unoptimized) | DP (Unoptimized) | DP (DDP) | DP (DeepSpeed) |
|---|---|---|---|---|
| **Model Architecture** | GPT-2 | | | |
| **Dataset** | Wikitext-2 | | | |
| **Hardware** | $8 \times$ NVIDIA A100 (80GB); All GPU pairs use NVLink 3.0; PCIe is bypassed for inter-GPU data transfer | | | |
| **Total Batch Size** | 112 | 112 | 112 | 112 |
| **Max Sequence Length** | 512 | | | |
| **Optimizer** | Adam ($\beta_1 = 0.9, \beta_2 = 0.999$) | | | |
| **Learning Rate** | $6 \times 10^{-5}$ (Constant Scheduler) | | | |
| **Precision** | BFloat16 | | | |
| **Communication Backend** | MP Queue (CPU) | MP Queue (CPU) | NCCL (GPU) | NCCL (GPU) |
| **Comm. Frequency** | Every Iteration | Every Iteration | Every Iteration | Every Iteration |
| **Optimization Level** | Naive Prototype | Naive Prototype | Bucketed All-Reduce | ZeRO-Stage 3 |
| **Window Size ($P$)** | 21 | N/A | N/A | N/A |
| **Adaptivity Threshold** | 0.1 | N/A | N/A | N/A |
| **Adaptivity EMA Decay** | 0.999 | N/A | N/A | N/A |

### F.1. Execution Commands for Data-parallelism Comparision Commands in Table 5

Below we give the detailed running scripts for reproducing the results in Table 5. In this scripts, we include execution commands for comparision between data-parallelism and our step-parallelism.

*Listing 9.* Step-parallellism v.s. Data-parallelism Comparision Commands in Table 5

```
# Adam Baseline
python llm.py --local_model_dir "./gpt2" --modelscope_name "gpt2" --max_steps 1000
    ↪ --batch_size 112 --learning_rate 6e-5 --training_mode serial --optimizer_type
    ↪ adam --lr_scheduler_type constant
# Step Parallelism w/o Optimization (PASO)
python llm.py --local_model_dir "./gpt2" --modelscope_name "gpt2" --max_steps 1000
    ↪ --batch_size 112 --learning_rate 6e-5 --threshold 0.1 --ema_decay 0.999
    ↪ --training_mode parallel --P 21 --optimizer_type adam --lr_scheduler_type constant
# Data Parallelism w/o Optimization
python llm.py --local_model_dir "./gpt2" --modelscope_name "gpt2" --max_steps 1000
    ↪ --batch_size 112 --learning_rate 6e-5 --training_mode naive_dp --optimizer_type
    ↪ adam --lr_scheduler_type constant
# Data Parallelism w/ Full Optimization (Pytorch's DDP)
python llm.py --local_model_dir "./gpt2" --modelscope_name "gpt2" --max_steps 1000
    ↪ --batch_size 112 --learning_rate 6e-5 --training_mode ddp --optimizer_type adam
    ↪ --lr_scheduler_type constant
# Data Parallelism w/ Full Optimization (DeepSpeed)
deepspeed --num_gpus=8 llm.py --local_model_dir "./gpt2" --modelscope_name "gpt2"
    ↪ --max_steps 1000 --batch_size 112 --learning_rate 6e-5 --training_mode deepspeed
    ↪ --optimizer_type adam --lr_scheduler_type constant --stage 3
```

## G. Complexity Analysis for OptEx and PASO

To understand the trade-offs inherent in step-parallel optimization, this section provides a comparative complexity analysis of PASO and OptEx. We analyze the total computational, space, and communication costs incurred during each "main iteration" of the respective algorithms.

- For PASO, a "main iteration" refers to a single parallel iteration $k$ of its fixed-point solver (Algorithm 1).

- For OptEx, a "main iteration" refers to a single sequential iteration $t$ that dispatches $N$ parallel gradient computations.

**Definitions**. We define the following variables for our analysis:

- $d$: The dimensionality of the model's parameter vector.

- $p$: The parallelism degree (window size) of PASO.

- $N$: The parallelism degree (number of parallel steps) of OptEx.

- $T_0$: The size of the local gradient history used by OptEx.

- $G$: The number of available processing units (e.g., GPUs).

- $t_{grad}(d)$: The computational cost of a single gradient calculation (one forward and backward pass), which is typically proportional to $d$. We simplify its cost to $\mathcal{O}(d)$.

- $t_{comm}(k, d)$: The communication cost to exchange $d$-dimensional vectors among $k$ nodes (e.g., via All-Gather or All-Reduce).

For simplicity, we assume an ideal scenario where the parallelism degree matches the number of available GPUs (i.e., $G = p$ for PASO and $G = N$ for OptEx).

### G.1. OptEx Complexity Analysis

**Total Computation (per iteration $t$).** The total computational cost for OptEx is the sum of its core gradient computation and its overhead:

1. **Core Gradient Computation**: $N$ gradients are computed in parallel. With $G = N$ GPUs, this takes $\mathcal{O}(\lceil N/G \rceil \cdot t_{grad}(d)) = \mathcal{O}(d)$.

2. **Overhead Computation**: This consists of two serial steps:
   - *Kernelized Gradient Estimation*: Building the estimation model $\mu_t$ from $T_0$ history points involves $T_0 \times T_0$ kernel matrix operations (e.g., inversion), costing $\mathcal{O}(T_0^3)$, plus matrix-vector operations costing $\mathcal{O}(T_0 d)$.
   - *Multi-Step Proxy Updates*: Serially computing $N - 1$ proxy steps to find the inputs for the parallel computation. Each step requires evaluating the kernel model, costing $\mathcal{O}(T_0 d + T_0^2)$. The total cost for this stage is $\mathcal{O}(N(T_0 d + T_0^2))$.

Combining these, the total computational cost is $\mathcal{O}(d + (T_0^3 + T_0 d) + (NT_0 d + NT_0^2))$. This is dominated by the overhead, yielding $\mathcal{O}(T_0^3 + NT_0(d + T_0))$.

**Total Space (excl. base model)** OptEx must store the $T_0$ historical gradients ($\mathcal{O}(T_0 d)$), the $T_0 \times T_0$ kernel matrix ($\mathcal{O}(T_0^2)$), and the $N$ inputs for the proxy updates ($\mathcal{O}(Nd)$). The total space is $\mathcal{O}(T_0 d + T_0^2 + Nd)$.

**Total Communication (per iteration $t$)** The primary communication involves gathering the $N$ computed gradients from the $N$ GPUs, which costs $\mathcal{O}(t_{comm}(N, d))$. The proxy updates are serial and do not add to the inter-node communication.

### G.2. PASO Complexity Analysis

**Total Computation (per iteration $k$)** The total computational cost for PASO is the sum of its parallel gradient computation and its serial update overhead:

1. **Core Gradient Computation**: $p$ gradients are computed in parallel. With $G = p$ GPUs, this takes $\mathcal{O}(\lceil p/G \rceil \cdot t_{grad}(d)) = \mathcal{O}(d)$.

2. **Overhead Computation**: This consists of serial updates on the host/main node after the parallel gradients are collected:
   - *Calculate Update Term $g^{(k)}$ (Line 6)*: For stateful optimizers like Adam, this forms a serial dependency chain of length $p$, costing $\mathcal{O}(pd)$.
   - *Update Model Weights $\hat{w}^{(k+1)}$ (Line 7)*: Updating all $p$ weights requires a cumulative sum, resulting in $\sum_{i=0}^{p-1}(i + 1) \approx \mathcal{O}(p^2)$ vector additions. The total cost is $\mathcal{O}(p^2 d)$.
   - *Check Error (Line 8)*: Computing $p$ L2-norm differences costs $\mathcal{O}(pd)$.

Combining these, the total computational cost is $\mathcal{O}(d + pd + p^2 d + pd) = \mathcal{O}(p^2 d)$.

**Total Space (excl. base model)** PASO needs to store the $p$ weights in the current window ($\mathcal{O}(pd)$) and their corresponding optimizer states (e.g., $p$ sets of $m, v$ moments for Adam, also $\mathcal{O}(pd)$). The total space is $\mathcal{O}(pd)$.

**Total Communication (per iteration $k$)** The primary communication involves gathering the $p$ computed gradients from the $p$ GPUs, which costs $\mathcal{O}(t_{comm}(p, d))$. The serial updates occur locally on the main node.

### G.3. Summary and Discussion

*Table 14.* Total per-iteration complexity comparison of OptEx and PASO, assuming parallelism degree matches the number of available GPUs ($G = p = N$). The speedup rate for OptEx is cited from the original paper, while ours is obtained from our Table 1 1).

| Metric | OptEx | PASO |
|---|---|---|
| Parallelism Degree | $N$ | $p = N$ |
| Total Computation | $\mathcal{O}(T_0^3 + NT_0(d + T_0))$ | $\mathcal{O}(N^2 d)$ |
| Total Space (Overhead) | $\mathcal{O}(T_0 d + T_0^2 + Nd)$ | $\mathcal{O}(Nd)$ |
| Total Communication | $\mathcal{O}(t_{comm}(N, d))$ | $\mathcal{O}(t_{comm}(N, d))$ |
| Speedup rate | $\mathcal{O}(\sqrt{N})$ | $\mathcal{O}(N)$ |

We summarize the complexity analysis in Table 14. The analysis reveals several key differences:

1. **Computational Cost**: OptEx's computation is highly sensitive to the history size $T_0$, featuring a $\mathcal{O}(T_0^3)$ term independent of model dimension $d$. In contrast, PASO's cost is independent of any history $T_0$ but scales quadratically with its parallelism degree $p$ ($\mathcal{O}(p^2 d)$). For large models (where $d$ is a dominant factor), the comparison simplifies to $\mathcal{O}(NT_0 d)$ for OptEx versus $\mathcal{O}(p^2 d)$ for PASO.

2. **Space Cost**: PASO demonstrates a significant advantage in space complexity. Its space overhead $\mathcal{O}(pd)$ scales linearly with its parallelism, whereas OptEx requires storing both the history and the proxy inputs, resulting in a larger $\mathcal{O}((T_0 + N)d)$ footprint.

3. **Communication Cost**: The communication costs are analogous and scale with their respective parallelism degrees, $N$ and $p$.

4. **Faster Speedup**: PASO achieves a linear speedup rate of $\mathcal{O}(N)$ with respect to the number of GPUs $N$. In contrast, OptEx only achieves a sub-linear speedup of $\mathcal{O}(\sqrt{N})$. This indicates that as the number of parallel processors (GPUs) increases, PASO's efficiency scales proportionally, whereas OptEx's performance gains diminish significantly.

This analysis suggests that PASO and OptEx have similar computational and communicational complexities. However, OptEx relies on a kernel estimation method that necessitates storing $T_0$ historical gradient records, which makes its space complexity much higher than that of PASO (about $\frac{T_0 + N}{N}$ times). This requirement, leading to a memory overhead proportional to $\mathcal{O}(T_0 d)$, makes OptEx infeasible for large-scale models where storage (memory) is often the primary bottleneck.

## H. Comparison with OptEx

We conduct a direct empirical comparison against **OptEx** (Shu et al., 2024). Following (Shu et al., 2024), we evaluate all methods on the three synthetic benchmark functions used in their official implementation[3]: the Sphere function, the Ackley function, and the Rosenbrock function (please refer to Appendix B.2.1 in the **OptEx** paper for the details of the three functions).

### H.1. Experimental Setup

We strictly adhere to the experimental setup provided in the OptEx codebase. As stated, the original OptEx source code only provides a complete and verifiable implementation for these three synthetic tasks. Due to the complexity of OptEx's kernel

---

[3]https://github.com/youyve/OptEx

*Table 15.* Quantitative comparisons of different methods. The best results are highlighted in **bold**. "↑" (resp. "↓") means the larger (resp. smaller), the better. We report the mean and standard deviation of the optimality gap over 5 independent runs.

| Method | Sphere function, $d = 1e4, \eta = 1e-3, T = 1000$ | | |
| --- | --- | --- | --- |
| | Optimality gap ↓ | Iters ↓ | Speedup ↑ |
| Adam | $1.95 \pm 0.12$ | 1000 | $1.0\times$ |
| OptEx | $1.95 \pm 0.08$ | 504 | $1.98\times$ |
| PASO | $1.95 \pm 0.15$ | **214** | **4.67×** |

| Method | Ackley function, $d = 1e4, \eta = 1e-3, T = 1000$ | | |
| --- | --- | --- | --- |
| | Optimality gap ↓ | Iters ↓ | Speedup ↑ |
| Adam | $0.40 \pm 0.18$ | 1000 | $1.0\times$ |
| OptEx | $0.40 \pm 0.06$ | 162 | $6.17\times$ |
| PASO | $0.40 \pm 0.11$ | **86** | **11.63×** |

| Method | Rosenbrock function, $d = 1e4, \eta = 1e-3, T = 1000$ | | |
| --- | --- | --- | --- |
| | Optimality gap ↓ | Iters ↓ | Speedup ↑ |
| Adam | $5.50 \pm 0.09$ | 1000 | $1.0\times$ |
| OptEx | $5.50 \pm 0.17$ | 479 | $2.08\times$ |
| PASO | $5.50 \pm 0.04$ | **210** | **4.76×** |

function estimation, we limit our comparison to these functions to ensure a correct and fair implementation of the OptEx baseline.

For all experiments, we use the Adam optimizer (Kingma & Ba, 2014) as the base solver. The standard, sequential Adam optimizer serves as our **Vanilla** baseline. We integrated our PASO framework directly into the *optex_cmp.py* script from the OptEx repository, ensuring that all methods (Vanilla, OptEx, and PASO) share the same experimental harness, initialization, and objective function calls.

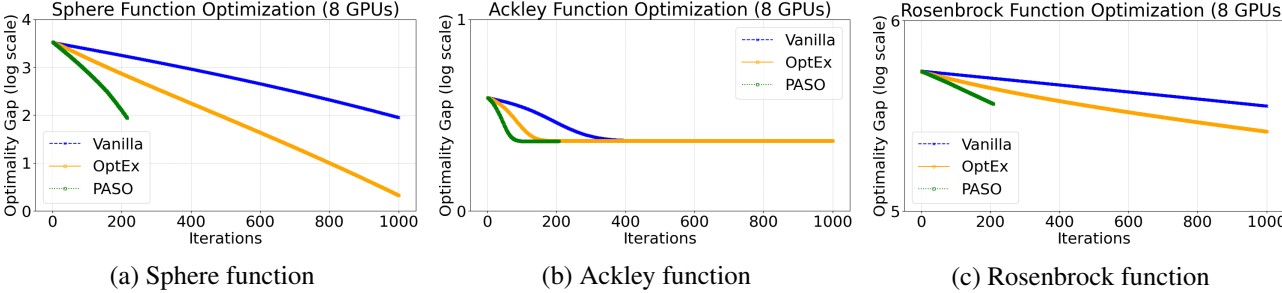

| (a) Sphere function | (b) Ackley function | (c) Rosenbrock function |

*Figure 8.* The comparison of optimality gap when optimizing the Sphere function, the Ackley function, and the Rosenbrock function.

### H.2. Metrics and Results

The primary metric for comparison is the **Optimality Gap**, defined as the final objective value $f(x_T)$ achieved by the Vanilla (Adam) baseline after $T = 1000$ iterations. (Note: for all three functions, the true minimum is $f^* = 0$, so the final loss $f(x_T)$ is equivalent to the optimality gap).

We then measure the number of sequential **Iterations** (i.e., communication rounds) required for OptEx and PASO to reach this same final optimality gap. The **Speedup** is calculated as the ratio of iterations taken by the Vanilla baseline (1000) to the iterations required by the parallel method.

The quantitative results are presented in Table 15 and the complete training processes are shown in Figure 8. The results

clearly demonstrate the superior efficiency of PASO.

- **On the Sphere function**, all methods converge to the same optimality gap of 1.95. However, PASO requires only **214** iterations, achieving a **4.67×** speedup. This is **2.36×** faster than OptEx, which requires 504 iterations (1.98× speedup).

- **On the Ackley function**, the performance gap is even more pronounced. PASO reaches the target gap of 0.4 in just **86** iterations, resulting in a massive **11.63×** speedup. OptEx requires 162 iterations (6.17× speedup).

- **On the complex Rosenbrock function**, PASO maintains a significant advantage, achieving a **4.76×** speedup (210 iterations) compared to OptEx's 2.08× speedup (479 iterations) to reach the 5.5 optimality gap.

Across all three standard benchmarks, PASO consistently and significantly outperforms OptEx, requiring far fewer communication rounds to converge to the same solution quality. This highlights the effectiveness of PASO's adaptive error-based stopping criterion, which achieves a more efficient parallel rollout compared to OptEx's kernel-based approximation approach.

## I. Use of LLM

During the preparation of this work, we used Large Language Models (LLMs) to assist with the writing process. The primary uses included polishing and improving the fluency of the text, generating preliminary drafts of proofs, and assisting in the creation and formatting of tables. After using these tools, the author(s) reviewed and edited the content extensively. We take full responsibility for the entire content of this publication, including the ideas, proofs, and presentations ultimately contained in the final manuscript.

## J. The Impact of PASO Hyperparameters

**Impact of Tolerance $\delta$ and EMA Decay Rate $\lambda$.** The Fig. 9 illustrates the impact of different tolerance ($\delta$) and EMA decay rate ($\lambda$) on model performance. The results show that different combinations of $\delta$ and $\lambda$ achieve a speedup of 4.61× (13000 v.s. 60000) to 4.81× (12450 v.s. 60000) while maintaining the similar model quality as Adam. In addition, the interplay between $\delta$ and $\lambda$ highlights a trade-off: aggressive smoothing ($\lambda \uparrow$) with loose tolerance ($\delta \uparrow$) may reduce computational effort, while finer tolerance ($\delta \downarrow$) with moderate $\lambda$ could enhance model quality at the expense of convergence speed.

In summary, these new parameters do not require extensive tuning and are quite intuitive in selection:

- **Tolerance** ($\delta$): This parameter controls the convergence precision of the fixed-point iteration. Manually setting this could be tedious. For this reason, we employ an adaptive tolerance schedule. The tolerance starts loose and automatically tightens as training progresses. This makes the method robust and largely removes $\delta$ from the list of parameters requiring manual tuning.

- **EMA Decay Rate** ($\lambda$): This is used within our adaptive tolerance schedule. Like most EMA parameters in deep learning (e.g., in batch normalization or Adam), it is not highly sensitive.

- **Window Size** ($p$): This is less of a hyperparameter and more of a hardware configuration parameter. For good efficiency, $p$ can be simply set as the number of available processors.

## K. Detailed Computation and Communication Analysis

In this section, we provide the detailed derivations and analyses of computational cost, memory footprint, and speedup ratios for sequential SGD and various parallel training methods, as summarized in Table 1.

### K.1. Computational Cost and Memory Footprint Analysis

To quantify the overhead of PASO, let $T$ be the total number of training steps for a standard sequential method. PASO converges in $K$ iterations, with each iteration performing $p$ parallel gradient computations (where $p$ is the window size). The total maximum number of gradient computations is therefore $p \times K$. Since the sliding window size $p$ will gradually decrease

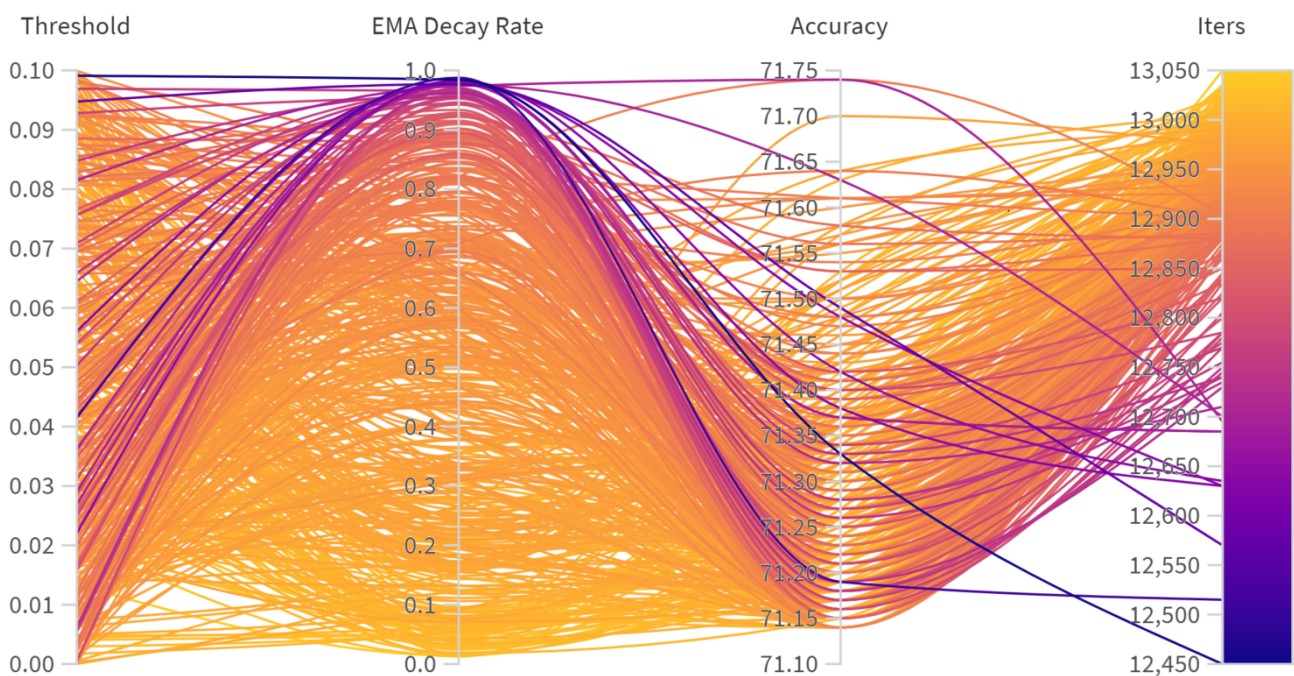

*Figure 9.* The impact of $\delta$ and $\lambda$ over CIFAR-10 by running 1200 experiments. We use PASO with $p = 7$ to accelerate Adam with 60000 steps. Darker lines indicate runs with fewer iterations.

at the end of the convergence, the practical number for gradients computations (we denote it as $G$) is less than $pK$. We define the *computational cost ratio* $m$ as the ratio of PASO's total gradient computations to that of the sequential method:

$$m = \frac{pK}{T}$$

Empirically, as shown in Figure 10, our experiments for $T = 10000$ demonstrate that $m$ remains close to 1 and does not exceed 1.5 across various window sizes. This indicates that PASO introduces minimal computational overhead.

In terms of memory, PASO requires storing only one model and one optimizer state per device. This is identical to the requirements of sequential, model, and pipeline parallelism. It is also significantly more memory-efficient than data parallelism, where the storage for optimizer states typically scales with the number of devices $N$.

### K.2. Speedup Ratio Analysis

In this section, we provide a detailed derivation of the speedup ratios for sequential SGD and various parallel training methods, as summarized in Table 1.

### K.2.1. DEFINITIONS AND ASSUMPTIONS

For a clear and consistent analysis, we define the following notations:

- $N$: The number of GPUs, assumed to have identical compute capabilities.

- $T$: The total number of iterations (steps) required for a model to converge using sequential SGD.

- $t_{\text{comp}}$: The time required for the computation within one SGD step on a single GPU. For simplicity, we normalize this to $t_{\text{step}}$ in some contexts.

- $t_{\text{comm}}$: The time required for necessary communication (e.g., synchronization) per parallel step.

- $\alpha \triangleq t_{\text{comm}}/t_{\text{comp}}$: The communication-to-computation ratio, a critical factor in parallel efficiency.

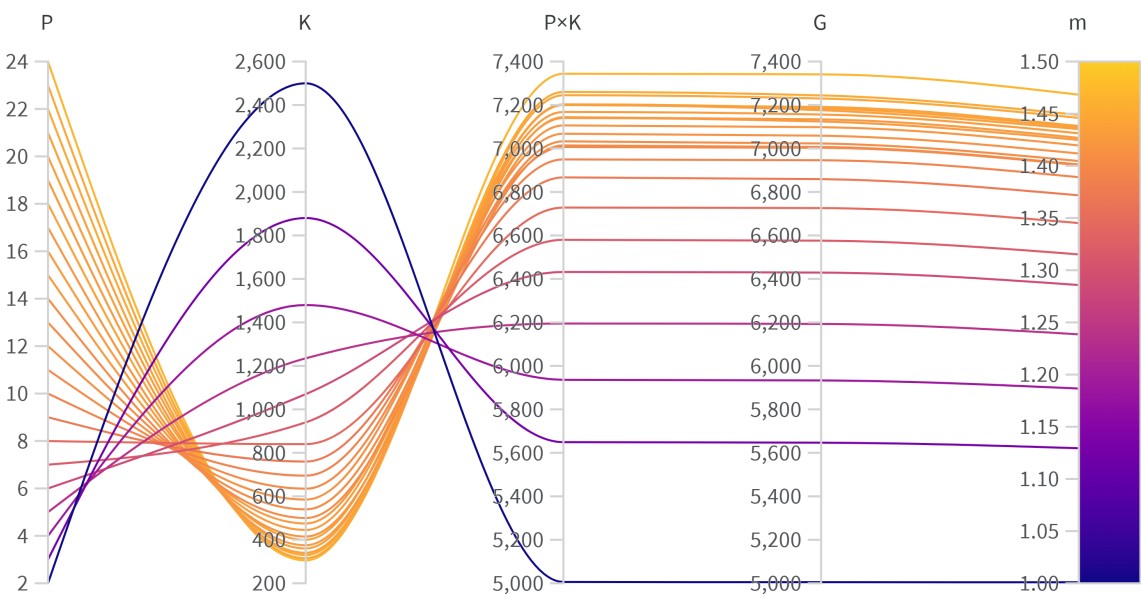

*Figure 10.* Empirical evaluation of the computational cost ratio $m = pK/T$ for $T = 10000$ across different window sizes $p$. Since the sliding window size $p$ will gradually decrease at the end of the convergence, the actual total number of gradient computations (we denote it as $G$) is marginally less than $pK$. The ratio remains close to 1, indicating minimal computational overhead.

The **speedup ratio** $S$ for any parallel method is defined as the ratio of the total time taken by sequential SGD to the time taken by the parallel method:

$$S = \frac{T_{\text{sequential}}}{T_{\text{parallel}}}$$

### K.2.2. BASELINE: SEQUENTIAL SGD

The total time for sequential SGD is the product of the number of iterations and the time per iteration.

$$T_{\text{sequential}} = T \times t_{\text{comp}}$$

By definition, its speedup ratio is $S_{\text{sequential}} = 1$.

### K.2.3. DATA-PARALLEL TRAINING

In synchronous data parallelism, the computation for each step is divided across $N$ GPUs, but a communication step (e.g., AllReduce) is required to synchronize gradients. The time for one parallel step is $(\frac{t_{\text{comp}}}{N} + t_{\text{comm}})$. The total time over $T$ iterations is:

$$T_{\text{data}} = T \times \left( \frac{t_{\text{comp}}}{N} + t_{\text{comm}} \right)$$

The speedup ratio is therefore:

$$S_{\text{data}} = \frac{T \cdot t_{\text{comp}}}{T \left( \frac{t_{\text{comp}}}{N} + t_{\text{comm}} \right)} = \frac{t_{\text{comp}}}{\frac{t_{\text{comp}}}{N} + \alpha \cdot t_{\text{comp}}} = \frac{1}{\frac{1}{N} + \alpha} = \frac{N}{1 + \alpha N}$$

In the communication-bound limit ($N \to \infty$), the speedup is capped at $S_{\text{data}} \to 1/\alpha$.

### K.2.4. MODEL-PARALLEL AND PIPELINE-PARALLEL TRAINING

For both model and pipeline parallelism, assuming perfect load balancing and ignoring initial pipeline-filling latency for large $T$, the computation is similarly distributed. The model is partitioned across $N$ devices, and each device computes its

part in parallel, followed by communication of activations or gradients between devices. The total time can be approximated as:

$$T_{\text{model/pipeline}} \approx T \times \left( \frac{t_{\text{comp}}}{N} + t_{\text{comm}} \right)$$

This yields the same speedup ratio form as data parallelism:

$$S_{\text{model/pipeline}} = \frac{N}{1 + \alpha N}$$

Practical limitations such as load imbalance or pipeline bubble latency often result in a lower effective speedup.

### K.2.5. STEP-PARALLEL TRAINING (PASO)

PASO operates differently by parallelizing across training steps. We introduce three key parameters for its analysis:

- $p$: The window size, representing the number of gradient steps computed in parallel.

- $K$: The total number of PASO iterations required for convergence.

- $m$: The computational cost ratio, $m = \frac{pK}{T}$, where $pK$ is the total number of gradient computations performed by PASO. Our empirical results show $m \approx 1$.

In each of the $K$ iterations, $p$ gradients are computed in parallel across $N$ devices. The computation time per iteration is $\frac{p \cdot t_{\text{comp}}}{N}$, followed by a single communication phase $t_{\text{comm}}$. The total time for PASO is:

$$T_{\text{PASO}} = K \times \left( \frac{p \cdot t_{\text{comp}}}{N} + t_{\text{comm}} \right)$$

To compare this with sequential training over $T$ steps, we substitute $K = \frac{mT}{p}$:

$$T_{\text{PASO}} = \frac{mT}{p} \left( \frac{p \cdot t_{\text{comp}}}{N} + t_{\text{comm}} \right) = mT \left( \frac{t_{\text{comp}}}{N} + \frac{t_{\text{comm}}}{p} \right)$$

The speedup ratio for PASO is then:

$$S_{\text{PASO}} = \frac{T \cdot t_{\text{comp}}}{mT \left( \frac{t_{\text{comp}}}{N} + \frac{t_{\text{comm}}}{p} \right)} = \frac{t_{\text{comp}}}{m \left( \frac{t_{\text{comp}}}{N} + \frac{\alpha \cdot t_{\text{comp}}}{p} \right)} = \frac{1}{m \left( \frac{1}{N} + \frac{\alpha}{p} \right)} = \frac{N}{m(1 + \alpha N/p)}$$

Since our experiments show $m \approx 1$ (see Figure 10), the speedup is approximately:

$$S_{\text{PASO}} \approx \frac{N}{1 + \alpha N/p}$$

As $p > 1$, it follows that $1 + \alpha N/p < 1 + \alpha N$, which confirms that $S_{\text{PASO}} > S_{\text{data/model/pipeline}}$. In the communication-bound limit ($N \to \infty$), the speedup is capped at $S_{\text{PASO}} \to p/(m\alpha)$, which is $p$ times higher than other methods.

## L. Notation Summary

Table 16 summarizes the notations.

## M. Update Rules for Various Optimizers in Definition 3.2

**Adam Optimizer**. At each iteration $t$, Adam computes the gradient of the loss function $\mathcal{L}(w_t, \zeta_t)$ over the mini-batch $\zeta_t$. It then updates two key quantities: the first moment $m_t$, which captures the momentum of the gradients, and the second moment $v_t$, which estimates the variability of the gradients. These updates are governed by exponential moving averages:

$$m_t = \beta_1 m_{t-1} + (1 - \beta_1) \nabla_{w_{t-1}} \mathcal{L}(w_{t-1}, \zeta_{t-1}), v_t = \beta_2 v_{t-1} + (1 - \beta_2)(\nabla_{w_{t-1}} \mathcal{L}(w_{t-1}, \zeta_{t-1}))^2, \quad (12)$$

*Table 16.* Summary of Notations

| Notation | Description |
|---|---|
| $T$ | Total number of gradient descent steps |
| $t$ | Current step index, $t \in \{0, 1, \ldots, T-1\}$ |
| $w_t$ | Model parameters at step $t$ |
| $\eta_t$ | Learning rate at step $t$ |
| $\zeta_t$ | Mini-batch of data used at step $t$ |
| $\mathcal{L}(w_t, \zeta_t)$ | Loss function evaluated at parameters $w_t$ with data $\zeta_t$ |
| $\nabla_{w_t} \mathcal{L}(w_t, \zeta_t)$ | Gradient of loss with respect to $w_t$ |
| $g_t(\cdot)$ | Update function specific to optimizer (SGD, Adam, etc.) |
| $r$ | The number of history weights used for existing autoregressive optimizers |
| $F_t(\cdot)$ | Nonlinear equation function at step $t$ |
| $\hat{w}_t^{(k)}$ | Estimated parameters at step $t$, iteration $k$ |
| $K$ | Number of parallel iterations |
| $p$ | Sliding window size for parallel computation |
| $\delta$ | Convergence tolerance threshold |
| $\lambda$ | Exponential moving average decay rate |
| $n$ | Dimension of model parameters |
| $L$ | Lipschitz constant for gradients |
| $M$ | Bound on gradient norm |
| $\epsilon$ | Small constant for numerical stability |
| $\beta_1, \beta_2$ | Exponential decay rates for optimizer with momentum |

where $\beta_1$ and $\beta_2$ are hyperparameters controlling the decay rates of the moving averages. Adam applies bias correction to the moments:

$$\hat{m}_t = \frac{m_t}{1-\beta_1^t}, \hat{v}_t = \frac{v_t}{1-\beta_2^t}. \tag{13}$$

The model parameters $w$ are then updated using the following rule:

$$w_t = w_{t-1} - \eta_{t-1} \frac{\hat{m}_t}{\sqrt{\hat{v}_t} + \epsilon}, \tag{14}$$

where the division is defined as the Hadamard division, and $\epsilon$ is a small constant.

For Adam , reformulating Eq. (12) produces the formulas of their general terms:

$$m_t = (1-\beta_1) \sum_{\tau=0}^{t-1} \beta_1^{t-1-\tau} \nabla_{w_\tau} \mathcal{L}(w_\tau, \zeta_\tau), v_t = (1-\beta_2) \sum_{\tau=0}^{t-1} \beta_2^{t-1-\tau} \left( \nabla_{w_\tau} \mathcal{L}(w_\tau, \zeta_\tau) \right)^2. \tag{15}$$

Through the combination of Eq. (13), Eq. (14), and Eq. (15), we derive $g_{t-1}$ with $r = t$ for Adam as follows:

$$g_{t-1}(w_{t-1}, \cdots, w_0; \zeta_{t-1}, \cdots, \zeta_0) = \frac{\frac{1-\beta_1}{1-\beta_1^t} \sum_{\tau=0}^{t-1} \beta_1^{t-1-\tau} \nabla_{w_\tau} \mathcal{L}(w_\tau, \zeta_\tau)}{\sqrt{\frac{(1-\beta_2) \sum_{\tau=0}^{t-1} \beta_2^{t-1-\tau} (\nabla_{w_\tau} \mathcal{L}(w_\tau, \zeta_\tau))^2}{1-\beta_2^t}} + \epsilon}. \tag{16}$$

**AdamW Optimizer**. The explicit form of $g_\tau$ for AdamW is derived by decoupling weight decay from the Adam update rule. Let $r = \tau$, then:

$$g_\tau(w_\tau, \ldots, w_0; \zeta_\tau, \ldots, \zeta_0) = \frac{\frac{1-\beta_1}{1-\beta_1^{\tau+1}} \sum_{k=0}^{\tau} \beta_1^{\tau-k} \nabla_{w_k} \mathcal{L}(w_k, \zeta_k)}{\sqrt{\frac{(1-\beta_2)}{1-\beta_2^{\tau+1}} \sum_{k=0}^{\tau} \beta_2^{\tau-k} \left( \nabla_{w_k} \mathcal{L}(w_k, \zeta_k) \right)^2} + \epsilon} + \lambda w_\tau,$$

where $\lambda$ is the weight decay coefficient. The term $\lambda w_\tau$ is explicitly added to the original Adam update, independent of gradient history.

**Adagrad Optimizer**. For Adagrad, the update function $g_\tau$ is defined using the explicit sum of squared gradients up to iteration $\tau$:

$$g_\tau(w_\tau, \ldots, w_0; \zeta_\tau, \ldots, \zeta_0) = \frac{\nabla_{w_\tau} \mathcal{L}(w_\tau, \zeta_\tau)}{\sqrt{\sum_{k=0}^{\tau} \left(\nabla_{w_k} \mathcal{L}(w_k, \zeta_k)\right)^2 + \epsilon}}.$$

Here, the denominator is the square root of the *non-decaying cumulative sum* of all historical squared gradients. $\epsilon$ is a small constant added for numerical stability.

**SAM Optimizer**. The explicit form of $g_\tau$ for SAM (Sharpness-Aware Minimization) involves computing the gradient at a perturbed point. Here in SAM $r = 1$, then:

$$g_\tau(w_\tau; \zeta_\tau) = \nabla_{w_\tau + \varepsilon_\tau} \mathcal{L}(w_\tau + \varepsilon_\tau, \zeta_\tau),$$

where the perturbation $\varepsilon_\tau$ is defined as:

$$\varepsilon_\tau = \rho \cdot \frac{\nabla_{w_\tau} \mathcal{L}(w_\tau, \zeta_\tau)}{\|\nabla_{w_\tau} \mathcal{L}(w_\tau, \zeta_\tau)\|_2 + \delta}.$$

Substituting the expression for $\varepsilon_\tau$ into the gradient formula, we get:

$$g_\tau(w_\tau; \zeta_\tau) = \nabla_{w_\tau} \mathcal{L}\left(w_\tau + \rho \cdot \frac{\nabla_{w_\tau} \mathcal{L}(w_\tau, \zeta_\tau)}{\|\nabla_{w_\tau} \mathcal{L}(w_\tau, \zeta_\tau)\|_2 + \epsilon}, \zeta_\tau\right).$$

This formulation explicitly shows SAM computes the gradient at a point that is perturbed in the direction of steepest ascent within a neighborhood of radius $\rho$, seeking parameters that are robust to adversarial perturbations. Here, $\rho$ is the perturbation radius that controls the magnitude of the perturbation, and $\epsilon$ is a small constant added for numerical stability.

# N. Proof of Proposition 3.3

## N.1. Proof of Uniqueness

Let $\{A_t\}_{t=0}^{T}$ and $\{B_t\}_{t=0}^{T}$ be two sets of solutions satisfying the Triangular Nonlinear Equations (TNEs) system. We prove $A_t = B_t$ for all $t \in [0, T]$ via induction:

- **Base Case ($t = 0$):** According to the first line of the TNEs system, $\hat{w}_0 - w_0 = 0$. This implies both solutions must satisfy $A_0 = w_0$ and $B_0 = w_0$, hence $A_0 = B_0$.

- **Inductive Hypothesis:** Assume that for some $k \in [0, T-1]$, the solutions are identical up to step $k$, such that $A_\tau = B_\tau$ for all $\tau \in \{0, \ldots, k\}$.

- **Inductive Step:** For $t = k + 1$, by the definition of the mapping $F_k$:

$$\begin{cases} A_{k+1} = F_k(A_0, \ldots, A_k; \zeta_0, \ldots, \zeta_k) \\ B_{k+1} = F_k(B_0, \ldots, B_k; \zeta_0, \ldots, \zeta_k) \end{cases} \tag{17}$$

  Since $F_k$ is a deterministic function of its parameters and the mini-batches $\zeta_\tau$ are fixed, the inductive hypothesis $A_\tau = B_\tau$ directly leads to:

$$A_{k+1} = F_k(A_0, \ldots, A_k; \zeta_0, \ldots, \zeta_k) = F_k(B_0, \ldots, B_k; \zeta_0, \ldots, \zeta_k) = B_{k+1}. \tag{18}$$

By the principle of mathematical induction, the TNEs system has a unique solution.

## N.2. Proof of Unbiasedness

We now prove that the unique solution $\{\hat{w}_t\}_{t=0}^{T}$ satisfies $E[\hat{w}_t] = E[w_t]$ for all $t \in [0, T]$, where $\{w_t\}_{t=0}^{T}$ is the trajectory generated by the autoregressive GD process (as defined in Eq. (3)).

- **Base Case ($t = 0$):** By the definition of the system, $\hat{w}_0 = w_0$. Therefore, $E[\hat{w}_0] = E[w_0]$, as the initial weights are identical.

- **Inductive Hypothesis:** Assume that for some $k \in [0, T-1]$, the estimator is unbiased for all steps up to $k$, i.e., $E[\hat{w}_\tau] = E[w_\tau]$ for all $\tau \in \{0, \ldots, k\}$.

- **Inductive Step:** For $t = k+1$, taking the expectation of the autoregressive GD process yields:

$$
\begin{aligned}
E[w_{k+1}] &= E[F_k(w_0, \ldots, w_k; \zeta_0, \ldots, \zeta_k)] \\
&= E[w_0] - \sum_{\tau=0}^{k} \eta_\tau E\left[g_\tau(w_\tau, \ldots, w_{\tau-r+1}; \zeta_\tau, \ldots, \zeta_{\tau-r+1})\right].
\end{aligned}
\tag{19}
$$

For the TNEs solution $\hat{w}_{k+1}$, the expectation is:

$$
\begin{aligned}
E[\hat{w}_{k+1}] &= E[F_k(\hat{w}_0, \ldots, \hat{w}_k; \zeta_0, \ldots, \zeta_k)] \\
&= E[\hat{w}_0] - \sum_{\tau=0}^{k} \eta_\tau E\left[g_\tau(\hat{w}_\tau, \ldots, \hat{w}_{\tau-r+1}; \zeta_\tau, \ldots, \zeta_{\tau-r+1})\right].
\end{aligned}
\tag{20}
$$

The mini-batches $\zeta_\tau$ used at step $\tau$ are identical between the sequential and parallel regimes. Given that the update rules $g_\tau$ are consistent and $E[\hat{w}_\tau] = E[w_\tau]$ for $\tau \leq k$, it follows that $E[\hat{w}_{k+1}] = E[w_{k+1}]$.

This completes the induction, proving that the TNEs solution unbiasedly estimates the GD trajectory:

$$
E[\hat{w}_t] = E[w_t], \quad \forall\, 0 \leq t \leq T.
$$

## O. Proof of Convergence for Fixed-Point Iteration in Proposition 3.4

### O.1. Assumptions and Lemmas

To give the proof, we first state the underlying assumptions and lemmas used:

**Assumption O.1.** The gradient $\nabla_{w_\tau} \mathcal{L}(w_\tau, \zeta_\tau)$ is $L$-Lipschitz continuous:

$$
\|\nabla_{w_\tau} \mathcal{L}(w_\tau, \zeta_\tau) - \nabla_{w_\tau} \mathcal{L}(x_\tau, \zeta_\tau)\| \leq L \|w_\tau - x_\tau\|.
\tag{21}
$$

**Assumption O.2.** The gradient norm is bounded:

$$
\|\nabla_{w_\tau} \mathcal{L}(w_\tau, \zeta_\tau)\| \leq M.
\tag{22}
$$

This implies bounded model weights $w_\tau$. For example, consider the simply quadratic loss $\mathcal{L}(w, \zeta) = w^2$ (with $w \in \mathbb{R}$ independent of $\zeta$). Here:

$$
\nabla_w \mathcal{L} = 2w, \quad \text{so} \quad |\nabla_w \mathcal{L}| = |2w|.
$$

The bounded gradient condition $|2w| \leq M$ directly implies $|w| \leq M/2$, proving $w$ is constrained to a compact set.

**Lemma O.3.** *If $U, V \in \mathbb{R}^{n \times t}$ satisfy $U_{ij}, V_{ij} \geq \mu$ for all $i, j$, then*

$$
\|\sqrt{U} - \sqrt{V}\|_F \;\leq\; \frac{1}{2\sqrt{\mu}} \|U - V\|_F,
$$

*where the squareroot is taken elementwise.*

*Proof.* For any scalars $a, b \geq \mu > 0$,

$$
\left|\sqrt{a} - \sqrt{b}\right| = \frac{|a - b|}{\sqrt{a} + \sqrt{b}} \;\leq\; \frac{|a - b|}{2\sqrt{\mu}},
$$

because $\sqrt{a} + \sqrt{b} \geq 2\sqrt{\mu}$.

Applying this entrywise with $a = U_{ij}$ and $b = V_{ij}$ yields

$$\left| \sqrt{U_{ij}} - \sqrt{V_{ij}} \right| \leq \frac{1}{2\sqrt{\mu}} \left| U_{ij} - V_{ij} \right| \qquad (\forall i, j).$$

Squaring and summing over $(i, j)$,

$$\sum_{i,j} \left( \sqrt{U_{ij}} - \sqrt{V_{ij}} \right)^2 \leq \frac{1}{4\mu} \sum_{i,j} (U_{ij} - V_{ij})^2.$$

The lefthand side equals $\|\sqrt{U} - \sqrt{V}\|_F^2$ and the righthand side equals $\frac{1}{4\mu} \|U - V\|_F^2$.

Taking square roots gives

$$\|\sqrt{U} - \sqrt{V}\|_F \leq \frac{1}{2\sqrt{\mu}} \|U - V\|_F,$$

$\square$

### O.2. Problem Restatement

**Notation.** We denote the collection of weights up to time $\tau$ as $W_\tau = [\hat{w}_0, \ldots, \hat{w}_\tau]$ and note $W_{T-1} = [\hat{w}_0, \ldots, \hat{w}_{T-1}]$ as $W$. The norm $\|\cdot\|$ is the Frobenius norm. For model weights $w \in \mathbb{R}^n$ with $n > 1$, multiplication and division are element-wise (Hadamard product and division).

**Definition O.4** (Iterative Mapping). Let the iterative mapping $\mathcal{H} : \mathbb{R}^{n \times T} \to \mathbb{R}^{n \times T}$ ($T$ components) be defined as follows for a sequence of model weights $W = [\hat{w}_0, \hat{w}_1, \ldots, \hat{w}_{T-1}]$:

$$\mathcal{H}(\hat{w}_0, \cdots, \hat{w}_{T-1}) = \begin{cases} \hat{w}_0 = w_0^{seq}, \\ F_0(\hat{w}_0; \zeta_0), \\ F_1(\hat{w}_0, \hat{w}_1; \zeta_0, \zeta_1), \\ \cdots, \\ F_{T-1}(\hat{w}_0, \cdots, \hat{w}_{T-1}; \zeta_0, \ldots, \zeta_{T-1}), \end{cases} \tag{23}$$

where $w_0^{seq}$ denotes the initialized model for the sequential gradient descent and each sub-mapping $F_{t-1}$ is of the form:

$$F_{t-1}(\hat{w}_0, \cdots, \hat{w}_{t-1}) = \hat{w}_0 - \sum_{\tau=0}^{t-1} \eta_\tau g_\tau(\hat{w}_\tau, \ldots, \hat{w}_0). \tag{24}$$

The fixed-point iteration is thus defined by the sequence $W^k = \mathcal{H}(W^{k-1})$.

**Definition O.5** (Autoregressive Gradient Descent Trajectory). The target fixed point, denoted by $W^{seq} = [w_0^{seq}, w_1^{seq}, \ldots, w_{T-1}^{seq}]$, is the trajectory generated by autoregressive gradient descent:

$$w_0^{seq} = \text{initial model weight} \tag{25}$$

$$w_t^{seq} = w_0^{seq} - \sum_{\tau=0}^{t-1} \eta_\tau g_\tau(w_\tau^{seq}, \ldots, w_0^{seq}) \quad \text{for } t \geq 1 \tag{26}$$

It is straightforward to see that $W^{seq}$ is a fixed point of $\mathcal{H}$, since $\mathcal{H}(W^{seq}) = W^{seq}$.

### O.3. Objectives

We aim to prove two key properties of this iterative process:

1. **Convergence:** The fixed-point iteration $W^k = \mathcal{H}(W^{k-1})$ converges to the unique fixed point $W^{seq}$, which corresponds to the trajectory of autoregressive gradient descent.

2. **Finite Convergence Steps:** Even in the worst case, exact convergence happens in $K \leq T$.

## O.4. Proof of Convergence (Objective 1)

We will prove by mathematical induction on the time step $t$ that for each $t \in \{0, \ldots, T-1\}$, the sequence of iterates $\{\hat{w}_t^k\}_{k=1}^{\infty}$ converges to $w_t^{seq}$.

*Proof.* Let $W^k = [\hat{w}_0^k, \ldots, \hat{w}_{T-1}^k]$ be the iterates at step $k$. From the definition of $\mathcal{H}$, we have:

$$\hat{w}_0^k = w_0^{seq} \tag{27}$$

$$\hat{w}_t^k = \hat{w}_0^{k-1} - \sum_{\tau=0}^{t-1} \eta_\tau g_\tau(\hat{w}_\tau^{k-1}, \ldots, \hat{w}_0^{k-1}) \quad \text{for } t \geq 1 \tag{28}$$

**Base Case ($t = 0$):** From the definition of $\mathcal{H}$, $\hat{w}_0^k = w_0^{seq}$ for all $k \geq 1$. Thus,

$$\lim_{k \to \infty} \left\| \hat{w}_0^k - w_0^{seq} \right\|_F = 0$$

The base case holds trivially.

**Inductive Hypothesis:** Assume for a given $t \geq 0$ that for all $\tau \in \{0, \ldots, t\}$, we have:

$$\lim_{k \to \infty} \left\| \hat{w}_\tau^k - w_\tau^{seq} \right\|_F = 0$$

**Inductive Step:** We must show that the statement holds for $t + 1$, i.e., $\lim_{k \to \infty} \left\| \hat{w}_{t+1}^k - w_{t+1}^{seq} \right\|_F = 0$.

The iterate $\hat{w}_{t+1}^k$ and the target $w_{t+1}^{seq}$ are given by:

$$\hat{w}_{t+1}^k = \hat{w}_0^{k-1} - \sum_{\tau=0}^{t} \eta_\tau g_\tau(W_\tau^{k-1})$$

$$w_{t+1}^{seq} = w_0^{seq} - \sum_{\tau=0}^{t} \eta_\tau g_\tau(W_\tau^{seq})$$

Since $\hat{w}_0^{k-1} = w_0^{seq}$ for $k - 1 \geq 1$, the difference is:

$$\hat{w}_{t+1}^k - w_{t+1}^{seq} = \sum_{\tau=0}^{t} \eta_\tau \left( g_\tau(W_\tau^{seq}) - g_\tau(W_\tau^{k-1}) \right)$$

Taking the norm and applying the triangle inequality:

$$\left\| \hat{w}_{t+1}^k - w_{t+1}^{seq} \right\|_F \leq \sum_{\tau=0}^{t} \eta_\tau \left\| g_\tau(W_\tau^{k-1}) - g_\tau(W_\tau^{seq}) \right\|_F$$

From Appendix O.6, O.7, and O.8. we can know that the gradient function $g_\tau$ for various optimizers is upper bounded with respect to its arguments. Denote uniformly by these boundaries $C$, we have:

$$\left\| \hat{w}_{t+1}^k - w_{t+1}^{seq} \right\|_F \leq \sum_{\tau=0}^{t} \eta_\tau C \left\| W_\tau^{k-1} - W_\tau^{seq} \right\|_F$$

By the inductive hypothesis, for each $\tau \in \{0, \ldots, t\}$, every component of $W_\tau^{k-1}$ converges to the corresponding component of $W_\tau^{seq}$ as $k \to \infty$. This implies that:

$$\lim_{k \to \infty} \left\| W_\tau^{k-1} - W_\tau^{seq} \right\|_F = \lim_{k \to \infty} \left( \sum_{j=0}^{\tau} \left\| \hat{w}_j^{k-1} - w_j^{seq} \right\|_F^2 \right)^{1/2} = 0$$

Since the sum on the right-hand side is a finite sum of terms each converging to zero, the entire expression converges to zero:

$$\lim_{k \to \infty} \left\| \hat{w}_{t+1}^k - w_{t+1}^{seq} \right\|_F \leq \sum_{\tau=0}^{t} \eta_\tau C \cdot 0 = 0$$

As the norm is non-negative, we conclude $\lim_{k \to \infty} \left\| \hat{w}_{t+1}^k - w_{t+1}^{seq} \right\|_F = 0$. This completes the inductive step.

By the principle of mathematical induction, $\hat{w}_t^k \to w_t^{seq}$ for all $t \in \{0, \ldots, T-1\}$. Therefore, the iteration $W^k = \mathcal{H}(W^{k-1})$ converges to $W^{seq}$. □

### O.5. Proof of Convergence Steps (Objective 2)

We now prove a stronger result: in worst-case scenario, exact convergence happen in $K \leq T$ iterations.

**Worst-Case Scenario Analysis.** The structure of the mapping $\mathcal{H}$ imposes a causal dependency: the calculation of $\hat{w}_t^k$ depends only on the components $\hat{w}_0^{k-1}, \ldots, \hat{w}_{t-1}^{k-1}$ from the previous iteration. The initial models for the fixed-point iteration and the autoregressive gradient descent are identical at $t = 0$ ($\hat{w}_0^k = w_0^{seq}$). Consequently, convergence cannot occur "out of order". The component $\hat{w}_1$ can only converge after $\hat{w}_0$ has, $\hat{w}_2$ can only converge after $\hat{w}_0$ and $\hat{w}_1$ have, and so on.

The worst-case scenario occurs when each iteration $k$ can only ensure the convergence of one component, leading to the convergence proceeding sequentially, one component at a time. This sequential "locking-in" of the correct values is equivalent in its step-by-step nature to the autoregressive gradient descent. We will formalize this intuition below.

*Proof.* We will prove by induction on the component index $t$ the statement $P(t)$:

$$P(t): \quad \hat{w}_t^k = w_t^{seq} \quad \text{for all } k \geq t+1.$$

**Base Case ($t = 0$):** We must prove $P(0)$: $\hat{w}_0^k = w_0^{seq}$ for all $k \geq 1$. By the definition of $\mathcal{H}$ in Eq. (23), $\hat{w}_0^k$ is set to $w_0^{seq}$ for every iteration $k \geq 1$. The base case holds.

**Inductive Hypothesis:** Assume for some $t \geq 1$ that $P(\tau)$ holds for all $\tau \in \{0, 1, \ldots, t-1\}$. This means for each such $\tau$:

$$\hat{w}_\tau^k = w_\tau^{seq} \quad \text{for all } k \geq \tau + 1.$$

**Inductive Step:** We must prove that $P(t)$ holds: $\hat{w}_t^k = w_t^{seq}$ for all $k \geq t+1$.

Consider an arbitrary iteration $k$ such that $k \geq t+1$. This implies $k - 1 \geq t$. The iterate $\hat{w}_t^k$ is defined as:

$$\hat{w}_t^k = \hat{w}_0^{k-1} - \sum_{\tau=0}^{t-1} \eta_\tau g_\tau(\hat{w}_\tau^{k-1}, \ldots, \hat{w}_0^{k-1}).$$

The arguments to the functions $g_\tau$ are the components of $W^{k-1}$. Let's examine an arbitrary component $\hat{w}_\tau^{k-1}$ in this expression, where $\tau \in \{0, 1, \ldots, t-1\}$. From our condition on $k$, we have $k - 1 \geq t > \tau$, which implies $k - 1 \geq \tau + 1$.

According to our inductive hypothesis, since $k - 1 \geq \tau + 1$, each of these components has already converged to its final value:

$$\hat{w}_\tau^{k-1} = w_\tau^{seq} \quad \text{for each } \tau \in \{0, 1, \ldots, t-1\}.$$

This demonstrates that for any iteration $k \geq t+1$, all the inputs required to compute $\hat{w}_t^k$ have already stabilized to their fixed-point values at the preceding step, $k - 1$.

Substituting these converged values back into the expression for $\hat{w}_t^k$:

$$\hat{w}_t^k = w_0^{seq} - \sum_{\tau=0}^{t-1} \eta_\tau g_\tau(w_\tau^{seq}, \ldots, w_0^{seq}).$$

The right-hand side of this equation is precisely the definition of the target sequential weight $w_t^{seq}$. Therefore,

$$\hat{w}_t^k = w_t^{seq}.$$

Since our choice of $k \geq t+1$ was arbitrary, this equality holds for all such $k$. This proves $P(t)$ and completes the inductive step.

**Conclusion on Iteration Count.** By induction, we have shown that $\hat{w}_t^k = w_t^{seq}$ for all $k \geq t+1$. For the entire vector $W^k = [\hat{w}_0^k, \ldots, \hat{w}_{T-1}^k]$ to converge, every component must have converged. The last component to converge is $\hat{w}_{T-1}^k$. Applying our result for $t = T-1$:

$$\hat{w}_{T-1}^k = w_{T-1}^{seq} \quad \text{for all } k \geq (T-1)+1 = T.$$

At iteration $k = T$, we have $T \geq t+1$ for all $t \in \{0, \ldots, T-1\}$. This implies that every component $\hat{w}_t^T$ has converged to $w_t^{seq}$. Thus, the entire vector has converged:

$$W^T = W^{seq}.$$

Therefore, it is also easy to see that even in the worst case, exact convergence happens in $K \leq T$ iterations since the first k model weight $\hat{w}_{0:k}$ must equal the sequential ones $w_{0:k}^{seq}$ after $k$ iterations. $\qquad\square$

## O.6. Upper Bound for the Difference of $g_t$ in SGD

For SGD, the update function $g_t$ takes the form:

$$g_t(w_t; \zeta_t) = \nabla_{w_t} \mathcal{L}(w_t, \zeta_t) \tag{29}$$

We aim to find an upper bound for $\|g_t(w_t) - g_t(x_t)\|$. By directly applying Assumption O.1 (L-Lipschitz continuity), we get:

$$\|g_t(w_t) - g_t(x_t)\| = \|\nabla_{w_t} \mathcal{L}(w_t, \zeta_t) - \nabla_{x_t} \mathcal{L}(x_t, \zeta_t)\| \leq L\|w_t - x_t\| \tag{30}$$

Therefore, for SGD, the Lipschitz constant of the update function $g_t$ is $L$.

## O.7. Upper Bound for the Difference of $g_t$ in Adam

**Notation.** We denote the collection of weights up to time $t$ as $W_\tau = [w_0, \ldots, w_t]$ and note $W_{T-1} = [w_0, \ldots, w_{T-1}]$ as $W$. Analogously, $X_t = [x_0, \ldots, x_t]$ and $X = [x_0, \ldots, x_{T-1}]$.

Our objective is to derive an upper bound for the difference $\|g_{t-1}(W_{t-1}) - g_{t-1}(X_{t-1})\|_F$ for any $W_{t-1}$ and $X_{t-1}$.

The function $g_{t-1}$ is defined as:

$$g_{t-1}(W_{t-1}) = \frac{A(W_{t-1})}{\sqrt{B(W_{t-1}) + \epsilon}} \tag{31}$$

where the division and square root are element-wise operations. The numerator $A(W_{t-1})$ and denominator component $B(W_{t-1})$ are defined as the bias-corrected first and second moment estimates:

$$A(W_{t-1}) = \frac{1 - \beta_1}{1 - \beta_1^t} \sum_{\tau=0}^{t-1} \beta_1^{t-1-\tau} \nabla_{w_\tau} \mathcal{L}(w_\tau, \zeta_\tau) \tag{32}$$

$$B(W_{t-1}) = \frac{1 - \beta_2}{1 - \beta_2^t} \sum_{\tau=0}^{t-1} \beta_2^{t-1-\tau} \left(\nabla_{w_\tau} \mathcal{L}(w_\tau, \zeta_\tau)\right)^2 \tag{33}$$

This proof relies on two standard assumptions:

**1. $L$-Lipschitz Gradient**: The gradient of the loss function is $L$-Lipschitz continuous, i.e., $\|\nabla \mathcal{L}(w) - \nabla \mathcal{L}(x)\|_F \leq L\|w - x\|_F$.

**2. Bounded Gradient Norm**: The Frobenius norm of the stochastic gradients is uniformly bounded by a constant $M$, i.e., $\|\nabla \mathcal{L}(w, \zeta)\|_F \leq M$.

For clarity, we will temporarily omit the subscript $t-1$ from $W$ and $X$ within the derivation and re-introduce it in the final result. We begin by decomposing the difference $g(W) - g(X)$ by adding and subtracting an intermediate term:

$$g(W) - g(X) = \left(\frac{A(W) - A(X)}{\sqrt{B(W) + \epsilon}}\right) + \left(\frac{A(X)}{\sqrt{B(W) + \epsilon}} - \frac{A(X)}{\sqrt{B(X) + \epsilon}}\right) \tag{34}$$

This can be expressed using the element-wise Hadamard product ($\odot$) as:

$$g(W) - g(X) = (A(W) - A(X)) \odot \frac{1}{\sqrt{B(W) + \epsilon}} + A(X) \odot \left( \frac{1}{\sqrt{B(W) + \epsilon}} - \frac{1}{\sqrt{B(X) + \epsilon}} \right) \tag{35}$$

By applying the triangle inequality to the Frobenius norm, we get:

$$\|g(W) - g(X)\|_F \leq \left\| (A(W) - A(X)) \odot \frac{1}{\sqrt{B(W) + \epsilon}} \right\|_F + \left\| A(X) \odot \left( \frac{1}{\sqrt{B(W) + \epsilon}} - \frac{1}{\sqrt{B(X) + \epsilon}} \right) \right\|_F \tag{36}$$

Next, we use the property of the Hadamard product, $\|U \odot V\|_F \leq \|U\|_{\max} \|V\|_F$, where $\|U\|_{\max}$ is the maximum absolute value of any element in $U$. This yields our main inequality:

$$\|g(W) - g(X)\|_F \leq \left\| \frac{1}{\sqrt{B(W) + \epsilon}} \right\|_{\max} \|A(W) - A(X)\|_F + \|A(X)\|_{\max} \left\| \frac{1}{\sqrt{B(W) + \epsilon}} - \frac{1}{\sqrt{B(X) + \epsilon}} \right\|_F \tag{37}$$

We now bound the four terms in Eq. (37).

**1. Bound for $\|A(W_{t-1}) - A(X_{t-1})\|_F$**

From the definition in Eq. (32), we have:

$$A(W) - A(X) = \frac{1 - \beta_1}{1 - \beta_1^t} \sum_{\tau=0}^{t-1} \beta_1^{t-1-\tau} \left( \nabla_{w_\tau} \mathcal{L}(w_\tau, \zeta_\tau) - \nabla_{x_\tau} \mathcal{L}(x_\tau, \zeta_\tau) \right) \tag{38}$$

Taking the Frobenius norm and applying the triangle inequality, then using the $L$-Lipschitz assumption and the fact that $\|w_\tau - x_\tau\|_F \leq \|W - X\|_F$:

$$\begin{aligned}
\|A(W) - A(X)\|_F &\leq \frac{1 - \beta_1}{1 - \beta_1^t} \sum_{\tau=0}^{t-1} \beta_1^{t-1-\tau} \|\nabla_{w_\tau} \mathcal{L}(w_\tau, \zeta_\tau) - \nabla_{x_\tau} \mathcal{L}(x_\tau, \zeta_\tau)\|_F \\
&\leq \frac{1 - \beta_1}{1 - \beta_1^t} \sum_{\tau=0}^{t-1} \beta_1^{t-1-\tau} L \|w_\tau - x_\tau\|_F \\
&\leq L \|W - X\|_F \left( \frac{1 - \beta_1}{1 - \beta_1^t} \sum_{\tau=0}^{t-1} \beta_1^{t-1-\tau} \right)
\end{aligned} \tag{39}$$

The sum of the bias-correction weights is equal to one. Thus, we have:

$$\|A(W_{t-1}) - A(X_{t-1})\|_F \leq L \|W_{t-1} - X_{t-1}\|_F \tag{40}$$

**2. Bound for $\left\| \frac{1}{\sqrt{B(W)+\epsilon}} \right\|_{\max}$**

Since each entry of $B(W)$ is a weighted average of squared gradients, $B_{ij}(W) \geq 0$ for all $i, j$. It follows that $\sqrt{B_{ij}(W) + \epsilon} \geq \sqrt{\epsilon}$. Taking the reciprocal gives the bound:

$$\left\| \frac{1}{\sqrt{B(W) + \epsilon}} \right\|_{\max} = \max_{i,j} \frac{1}{\sqrt{B_{ij}(W) + \epsilon}} \leq \frac{1}{\sqrt{\epsilon}} \tag{41}$$

**3. Bound for $\|A(X)\|_{\max}$**

Given the bounded gradient assumption $\|\nabla \mathcal{L}\|_F \leq M$, and since $\| \cdot \|_{\max} \leq \| \cdot \|_F$, we have $\|\nabla \mathcal{L}\|_{\max} \leq M$.

$$\begin{aligned}
\|A(X)\|_{\max} &\leq \left\| \frac{1 - \beta_1}{1 - \beta_1^t} \sum_{\tau=0}^{t-1} \beta_1^{t-1-\tau} \nabla_{x_\tau} \mathcal{L}(x_\tau, \zeta_\tau) \right\|_{\max} \\
&\leq \frac{1 - \beta_1}{1 - \beta_1^t} \sum_{\tau=0}^{t-1} \beta_1^{t-1-\tau} \|\nabla_{x_\tau} \mathcal{L}\|_{\max} \leq M
\end{aligned} \tag{42}$$

**4. Bound for** $\left\| \frac{1}{\sqrt{B(W)+\epsilon}} - \frac{1}{\sqrt{B(X)+\epsilon}} \right\|_F$

Let $u = B(W) + \epsilon$ and $v = B(X) + \epsilon$. We have:

$$\left\| \frac{1}{\sqrt{u}} - \frac{1}{\sqrt{v}} \right\|_F = \left\| \frac{\sqrt{v} - \sqrt{u}}{\sqrt{u}\sqrt{v}} \right\|_F \leq \left\| \frac{1}{\sqrt{uv}} \right\|_{\max} \|\sqrt{v} - \sqrt{u}\|_F \leq \frac{1}{\epsilon} \|\sqrt{v} - \sqrt{u}\|_F \tag{43}$$

The function $f(x) = \sqrt{x}$ is $\frac{1}{2\sqrt{\epsilon}}$-Lipschitz on $[\epsilon, \infty)$, which implies $\|\sqrt{v} - \sqrt{u}\|_F \leq \frac{1}{2\sqrt{\epsilon}}\|v - u\|_F$ (see Lemma O.3). Therefore:

$$\left\| \frac{1}{\sqrt{B(W)+\epsilon}} - \frac{1}{\sqrt{B(X)+\epsilon}} \right\|_F \leq \frac{1}{2\epsilon^{3/2}}\|B(W) - B(X)\|_F \tag{44}$$

To complete this bound, we must bound $\|B(W) - B(X)\|_F$. From Eq. (33), we analyze the difference of squares term $(\nabla_{w_\tau}\mathcal{L})^2 - (\nabla_{x_\tau}\mathcal{L})^2 = (\nabla_{w_\tau}\mathcal{L} - \nabla_{x_\tau}\mathcal{L}) \odot (\nabla_{w_\tau}\mathcal{L} + \nabla_{x_\tau}\mathcal{L})$. Taking the norm:

$$\begin{aligned}
\|(\nabla_{w_\tau}\mathcal{L})^2 - (\nabla_{x_\tau}\mathcal{L})^2\|_F &\leq \|\nabla_{w_\tau}\mathcal{L} - \nabla_{x_\tau}\mathcal{L}\|_F \cdot \|\nabla_{w_\tau}\mathcal{L} + \nabla_{x_\tau}\mathcal{L}\|_{\max} \\
&\leq (L\|w_\tau - x_\tau\|_F) \cdot (\|\nabla_{w_\tau}\mathcal{L}\|_{\max} + \|\nabla_{x_\tau}\mathcal{L}\|_{\max}) \\
&\leq (L\|w_\tau - x_\tau\|_F) \cdot (M + M) = 2LM\|w_\tau - x_\tau\|_F
\end{aligned} \tag{45}$$

Summing over $\tau$ with the bias-corrected weights gives $\|B(W) - B(X)\|_F \leq 2LM\|W - X\|_F$. Substituting this into Eq. (44):

$$\left\| \frac{1}{\sqrt{B(W)+\epsilon}} - \frac{1}{\sqrt{B(X)+\epsilon}} \right\|_F \leq \frac{2LM}{2\epsilon^{3/2}}\|W - X\|_F = \frac{LM}{\epsilon^{3/2}}\|W - X\|_F \tag{46}$$

**Final Result**. We now substitute the bounds from Eq. (40), Eq. (41), Eq. (42), and Eq. (46) into our main inequality Eq. (37).

$$\begin{aligned}
\|g_{t-1}(W) - g_{t-1}(X)\|_F &\leq \left( \frac{1}{\sqrt{\epsilon}} \right) \cdot (L\|W - X\|_F) + (M) \cdot \left( \frac{LM}{\epsilon^{3/2}}\|W - X\|_F \right) \\
&= \left( \frac{L}{\sqrt{\epsilon}} + \frac{M^2 L}{\epsilon^{3/2}} \right) \|W_{t-1} - X_{t-1}\|_F
\end{aligned} \tag{47}$$

This final result provides an upper bound for the difference in the Adam update step that depends only on the problem constants $L, M, \epsilon$.

### O.8. Upper Bound for the Difference of $g_t$ in AdamW

The AdamW update function $g_t$ can be decomposed into the Adam update term and a decoupled weight decay term:

$$g_t(W_t) = g_t^{\text{Adam}}(W_t) + \lambda_t w_t \tag{48}$$

where $\lambda_t$ is the weight decay coefficient. We analyze the norm of its difference using the triangle inequality:

$$\|g_t(W) - g_t(X)\|_F = \|(g_t^{\text{Adam}}(W) - g_t^{\text{Adam}}(X)) + \lambda_t(w_t - x_t)\|_F \tag{49}$$

$$\leq \|g_t^{\text{Adam}}(W) - g_t^{\text{Adam}}(X)\|_F + \lambda_t\|w_t - x_t\|_F \tag{50}$$

We now substitute the final bound derived for the Adam component in Appendix O.7:

$$\|g_t^{\text{Adam}}(W) - g_t^{\text{Adam}}(X)\|_F \leq \left( \frac{L}{\sqrt{\epsilon}} + \frac{M^2 L}{\epsilon^{3/2}} \right) \|W_t - X_t\|_F \tag{51}$$

Assuming an upper bound for the weight decay coefficient, $\lambda_t \leq \lambda_{\max}$, and noting that $\|w_t - x_t\|_F \leq \|W_t - X_t\|_F$, we have:

$$\|g_t(W) - g_t(X)\|_F \leq \left( \frac{L}{\sqrt{\epsilon}} + \frac{M^2 L}{\epsilon^{3/2}} \right) \|W_t - X_t\|_F + \lambda_{\max}\|W_t - X_t\|_F \tag{52}$$

$$= \left( \lambda_{\max} + \frac{L}{\sqrt{\epsilon}} + \frac{M^2 L}{\epsilon^{3/2}} \right) \|W_t - X_t\|_F \tag{53}$$

This provides a rigorous upper bound for the difference in the AdamW update step.

# P. Proof of Convergence for PASO with Sliding Window

## P.1. Notation and Problem Restatement

**Basic Notation.** We denote the full sequence of weights up to time $T - 1$ as $W = [\hat{w}_0, \ldots, \hat{w}_{T-1}]$. The norm $\| \cdot \|_F$ denotes the Frobenius norm. For model weights $w \in \mathbb{R}^n$ with $n > 1$, operations are assumed to be element-wise. The target fixed point $W^{seq} = [w_0^{seq}, w_1^{seq}, \ldots, w_{T-1}^{seq}]$ is the exact trajectory generated by standard autoregressive gradient descent.

The base update function for a single model state at index $t$ is defined as:

$$F_{t-1}(\hat{w}_0, \cdots, \hat{w}_{t-1}) = \hat{w}_0 - \sum_{\tau=0}^{t-1} \eta_\tau g_\tau(\hat{w}_\tau, \ldots, \hat{w}_0) \tag{54}$$

We denote the prefix sequence up to time $\tau$ as $W_{\leq \tau} = [\hat{w}_0, \ldots, \hat{w}_\tau]$.

**Block Decomposition & Core Variables.** We systematically partition the total sequence into $m$ consecutive functional blocks, $\mathcal{B}_1, \ldots, \mathcal{B}_m$, where $m \in [\lceil T/p \rceil, T]$.

- **Block Indices:** $\mathcal{B}_i = \{t_i, \ldots, t_{i+|\mathcal{B}_i|-1}\}$. The parallel capacity is strictly bounded by the maximum window size: $|\mathcal{B}_i| \leq p$.

- **Block State:** $W_{\mathcal{B}_i} = [\hat{w}_{t_i}, \ldots, \hat{w}_{t_{i+|\mathcal{B}_i|-1}}]$. The exact sequential target for this block is $W_{\mathcal{B}_i}^{seq}$.

- **Historical Context:** $H_{i-1}$ is the actual historical input received by block $i$ from previously processed blocks. The ideal sequential history is $H_{i-1}^{seq}$.

- **Block Operator:** For a fixed history $H_{i-1}$, the parallel evaluation mapping over the block at iteration $j$ is defined as:

$$F_{\mathcal{B}_i}(W_{\mathcal{B}_i}^{(j-1)} \mid H_{i-1}) = \left[ F_{t_i}(\hat{w}_{t_i} \mid H_{i-1}), \ldots, F_{t_{i+|\mathcal{B}_i|-1}}(\hat{w}_{t_i}, \ldots, \hat{w}_{t_{i+|\mathcal{B}_i|-1}} \mid H_{i-1}) \right]$$

- **Block Error:** $E_i^{(j)} = \|W_{\mathcal{B}_i}^{(j)} - W_{\mathcal{B}_i}^{seq}\|_F$.

## P.2. Assumptions.

**Assumption P.1** (Lipschitz Continuity). The gradient update function $g_\tau$ is uniformly Lipschitz continuous. There exists a constant $C > 0$ such that for any two sequences $X_\tau$ and $Y_\tau$, $\|g_\tau(X_\tau) - g_\tau(Y_\tau)\|_F \leq C\|X_\tau - Y_\tau\|_F$.

**Assumption P.2** (Bounded Initial Distance). For any functional block $i$, the initial bounded drift before its localized parallel updates begin is bounded by a constant $D > 0$, defined as $\max_i \|W_{\mathcal{B}_i}^{(0)} - W_{\mathcal{B}_i}^{seq}\|_F \leq D$.

## P.3. Main Theorem

**Theorem P.3** (Convergence and Parallel Speedup of PASO). *Consider a sequence of length $T$ processed with a maximum sliding window size of $p \in [1, T]$, partitioned into $m$ functional blocks. The system undergoes $K$ total parallel iterations, yielding an average of $K/m$ iterations per block. Suppose the update function satisfies Assumption P.1 and the initial drift is bounded by $D$ (Assumption P.2).*

*Let the internal block contraction factor $\rho_1$ and the historical propagation factor $\rho_2$ be conservatively bounded by the window size $p$ across all blocks as:*

$$\rho_1 = \sqrt{p} \cdot \max_i \sum_{\tau \in \mathcal{B}_i} \eta_\tau C \tag{55}$$

$$\rho_2 = \sqrt{p} \cdot \max_i \sum_{\tau \in H_{i-1}} \eta_\tau C \tag{56}$$

*If the learning rate schedule ensures strict internal contraction such that $\rho_1 \in (0, 1)$, the PASO algorithm mathematically guarantees:*

*(1) Global Convergence Error Bound:* The final approximation error of the terminal block $m$ is strictly bounded by:

$$\|W_{\mathcal{B}_m}^{(K/m)} - W_{\mathcal{B}_m}^{seq}\|_F \leq D \cdot \rho_1^{\frac{K}{m}} \cdot \left(1 + \frac{\rho_2(1 - \rho_1^{\frac{K}{m}})}{1 - \rho_1}\right)^{m-1}$$

*(2) Parallel Speedup Condition* ($K < T$): *To achieve $\epsilon$-precision convergence in strictly fewer iterations than standard sequential execution, the system must satisfy the condition at $K = T$:*

$$\rho_1^{\frac{T}{m}} \cdot \left(1 + \frac{\rho_2(1 - \rho_1^{\frac{T}{m}})}{1 - \rho_1}\right)^{m-1} < \frac{\epsilon}{D}$$

*This mathematical threshold guarantees that the exponential internal decay ($\rho_1^{T/m}$) governed by the window capacity is sufficient to overpower the historical structural error aggregation.*

**Corollary P.4.** *The decoupled bound established in Theorem P.3 serves as a strict generalization of the uniform global contraction mapping. If we consider the stricter condition where the global trajectory operates under a single uniform contraction factor $\rho \in (0, 1)$, we can symmetrically bound both the internal block mapping and the historical propagation by $\rho$ (i.e., $\rho_1 \leq \rho$ and $\rho_2 \leq \rho$). Under this uniform global assumption, the historical multiplier term elegantly collapses via standard algebraic simplification:*

$$1 + \frac{\rho(1 - \rho^{\frac{K}{m}})}{1 - \rho} = \frac{1 - \rho + \rho - \rho^{\frac{K}{m}+1}}{1 - \rho} = \frac{1 - \rho^{\frac{K}{m}+1}}{1 - \rho} \tag{57}$$

*Substituting this collapsed term back into the global convergence error bound strictly reduces our generalized result to the simplified uniform bound:*

$$\|W_{\mathcal{B}_m}^{(K/m)} - W_{\mathcal{B}_m}^{seq}\|_F \leq D \cdot \rho^{\frac{K}{m}} \cdot \left(\frac{1 - \rho^{\frac{K}{m}+1}}{1 - \rho}\right)^{m-1} \tag{58}$$

## P.4. Phase 1: Finite Bounds on Iteration Count

**Lemma P.5** (Finite Bounds of Termination). *For the PASO algorithm to iteratively process the trajectory via sliding windows, the structural hyperparameters and resulting iteration counts are bounded by $\lceil T/p \rceil \leq K \leq T$ and $\lceil T/p \rceil \leq m \leq T$.*

*Proof.* The causal triangularity inherently forces the first element of any active window to reach its fixed point in exactly one iteration, yielding a successive difference of $0$. Because adaptive tolerance $\delta^{(k)} \geq 0$, the condition $0 \leq \delta^{(k)}$ structurally guarantees a minimum forward stride of $s_k \geq 1$. Thus, the algorithm naturally terminates in at most $T$ iterations ($K \leq T$). Conversely, since the maximum advance step is upper-bounded by the window capacity ($s_k \leq p$), completing the sequence requires $\sum_{k=1}^{K} s_k = T \implies K \cdot p \geq T$, establishing $K \geq \lceil T/p \rceil$. The number of structural blocks $m$ is analogously bounded. $\square$

## P.5. Phase 2: Block-wise Contraction Mapping

**Lemma P.6** (Block-wise Contraction Mapping). *For any functional block $\mathcal{B}_i$ and fixed historical input $H_{i-1}$, the operator $F_{\mathcal{B}_i}(\cdot \mid H_{i-1})$ forms a strict contraction mapping bounded by $\rho_1$.*

*Proof.* Let $W$ and $V$ be two distinct sequence states for block $\mathcal{B}_i$. Since history $H_{i-1}$ is identical, gradient updates prior to index $t_i$ perfectly cancel out. For any specific index $t \in \mathcal{B}_i$:

$$F_t(W \mid H_{i-1}) - F_t(V \mid H_{i-1}) = -\sum_{\tau=t_i}^{t} \eta_\tau \left(g_\tau(W) - g_\tau(V)\right) \tag{59}$$

By Assumption P.1, because $W$ and $V$ share the identical history prefix, the gradient difference is strictly bounded by their distance across the current block:

$$\|g_\tau(W) - g_\tau(V)\|_F \leq C\|W_{\leq\tau} - V_{\leq\tau}\|_F \leq C\|W_{\mathcal{B}_i} - V_{\mathcal{B}_i}\|_F \tag{60}$$

Applying the triangle inequality to the state update and substituting this bound:

$$\|F_t(W \mid H_{i-1}) - F_t(V \mid H_{i-1})\|_F \leq \left( \sum_{\tau=t_i}^{t} \eta_\tau C \right) \|W_{\mathcal{B}_i} - V_{\mathcal{B}_i}\|_F \tag{61}$$

Since the learning rate and Lipschitz constant are non-negative, we can strictly bound the partial sum up to index $t$ by extending it to the entire block $\mathcal{B}_i$. By definition of the Frobenius norm over the evaluated block (bounded by capacity $p$):

$$
\begin{aligned}
\|F_{\mathcal{B}_i}(W \mid H_{i-1}) - F_{\mathcal{B}_i}(V \mid H_{i-1})\|_F &= \sqrt{\sum_{t \in \mathcal{B}_i} \|F_t(W \dots) - F_t(V \dots)\|_F^2} \\
&\leq \sqrt{\sum_{k=0}^{p-1} \left( \sum_{\tau \in \mathcal{B}_i} \eta_\tau C \right)^2} \cdot \|W_{\mathcal{B}_i} - V_{\mathcal{B}_i}\|_F \\
&= \left( \sqrt{p} \sum_{\tau \in \mathcal{B}_i} \eta_\tau C \right) \|W_{\mathcal{B}_i} - V_{\mathcal{B}_i}\|_F
\end{aligned}
\tag{62}
$$

By the definition in Theorem P.3, this bounding coefficient is exactly $\rho_1$. Because the learning rate schedule ensures $\rho_1 < 1$, it mathematical guarantees a strict internal contraction mapping. $\qquad\square$

### P.6. Phase 3: Global Convergence Error Bound

*Proof of Theorem P.3.* **Step 1: Single Iteration Recurrence & Historical Expansion.** Consider block $i$ at iteration $j \geq 1$. Bounding the target distance via the triangle inequality with a cross-term:

$$E_i^{(j)} \leq \underbrace{\|F_{\mathcal{B}_i}(W_{\mathcal{B}_i}^{(j-1)} \mid H_{i-1}) - F_{\mathcal{B}_i}(W_{\mathcal{B}_i}^{seq} \mid H_{i-1})\|_F}_{\text{Internal Contraction: } \leq \rho_1 E_i^{(j-1)}} + \underbrace{\|F_{\mathcal{B}_i}(W_{\mathcal{B}_i}^{seq} \mid H_{i-1}) - F_{\mathcal{B}_i}(W_{\mathcal{B}_i}^{seq} \mid H_{i-1}^{seq})\|_F}_{\text{Historical Error Propagation}} \tag{63}$$

For the historical term, internal gradient contributions ($\tau \geq t_i$) cancel out since the target block states $W_{\mathcal{B}_i}^{seq}$ are identical. The residual difference is driven purely by the history $\tau \in H_{i-1}$:

$$\|F_t(\cdots \mid H_{i-1}) - F_t(\cdots \mid H_{i-1}^{seq})\|_F \leq \left( \sum_{\tau \in H_{i-1}} \eta_\tau C \right) \|H_{i-1} - H_{i-1}^{seq}\|_F \tag{64}$$

Aggregating via the root sum of squares over the parallel components, and explicitly bounding the dynamic block size by the window capacity ($|\mathcal{B}_i| \leq p$), we obtain the $p$-dependent historical bound $\rho_2$:

$$
\begin{aligned}
\|F_{\mathcal{B}_i}(\dots) - F_{\mathcal{B}_i}^{seq}(\dots)\|_F &\leq \left( \sqrt{|\mathcal{B}_i|} \sum_{\tau \in H_{i-1}} \eta_\tau C \right) \|H_{i-1} - H_{i-1}^{seq}\|_F \\
&\leq \left( \sqrt{p} \sum_{\tau \in H_{i-1}} \eta_\tau C \right) \|H_{i-1} - H_{i-1}^{seq}\|_F \\
&\leq \rho_2 \|H_{i-1} - H_{i-1}^{seq}\|_F
\end{aligned}
\tag{65}
$$

Letting $\delta_{hist} = \rho_2 \|H_{i-1} - H_{i-1}^{seq}\|_F$, the single iteration recurrence strictly resolves to:

$$E_i^{(j)} \leq \rho_1 E_i^{(j-1)} + \delta_{hist} \tag{66}$$

**Step 2: Unrolling Iterations within the Block.** For block $i$ processing over $K/m$ iterations, unrolling the linear recurrence yields:

$$E_i \leq \rho_1^{\frac{K}{m}} E_i^{(0)} + \frac{1 - \rho_1^{\frac{K}{m}}}{1 - \rho_1} \delta_{hist} \tag{67}$$

The initial error is bounded by $D$. The historical delta concatenates previous block errors $\delta_{hist} \leq \rho_2 \sum_{k=1}^{i-1} E_k$, establishing the strictly decoupled causal propagation:

$$E_i \leq \rho_1^{\frac{K}{m}} D + \frac{\rho_2(1 - \rho_1^{\frac{K}{m}})}{1 - \rho_1} \sum_{k=1}^{i-1} E_k \tag{68}$$

**Step 3: Solving the Global Propagation to the Terminal Block.** Let constants $A = \rho_1^{\frac{K}{m}} D$ and $B = \frac{\rho_2(1 - \rho_1^{\frac{K}{m}})}{1 - \rho_1}$. The propagation strictly follows the recursive sequence:

$$E_i \leq A + B \sum_{k=1}^{i-1} E_k \tag{69}$$

We solve this recurrence strictly using strong mathematical induction to prove that the closed-form solution is $E_i \leq A(1 + B)^{i-1}$ for all $i \geq 1$.

*Base Case ($i = 1$):* The historical summation is empty, yielding $E_1 \leq A = A(1 + B)^0$, which holds true trivially.

*Inductive Step:* Assume the bound $E_k \leq A(1 + B)^{k-1}$ holds for all previous blocks $k \in [1, i - 1]$. We evaluate the error for block $i$:

$$\begin{aligned}
E_i &\leq A + B \sum_{k=1}^{i-1} E_k \\
&\leq A + B \sum_{k=1}^{i-1} A(1 + B)^{k-1}
\end{aligned} \tag{70}$$

Recognizing that the summation forms a standard geometric series with a common ratio of $(1 + B)$, we can evaluate it exactly:

$$\sum_{k=1}^{i-1} (1 + B)^{k-1} = \frac{(1 + B)^{i-1} - 1}{(1 + B) - 1} = \frac{(1 + B)^{i-1} - 1}{B} \tag{71}$$

Substituting this geometric sum back into the inequality effortlessly cancels out the $B$ multiplier:

$$\begin{aligned}
E_i &\leq A + B \cdot A \left( \frac{(1 + B)^{i-1} - 1}{B} \right) \\
&= A + A \left( (1 + B)^{i-1} - 1 \right) \\
&= A(1 + B)^{i-1}
\end{aligned} \tag{72}$$

This flawlessly concludes the induction. Evaluating this closed-form solution for the terminal functional block ($i = m$) provides $E_m \leq A(1 + B)^{m-1}$.

Finally, substituting the original system variables $A$ and $B$ back into this expression yields the exact global convergence bound parameterized by the window size $p$:

$$\|W_{\mathcal{B}_m}^{(K/m)} - W_{\mathcal{B}_m}^{seq}\|_F \leq D \cdot \rho_1^{\frac{K}{m}} \cdot \left( 1 + \frac{\rho_2(1 - \rho_1^{\frac{K}{m}})}{1 - \rho_1} \right)^{m-1} \tag{73}$$

**Step 4: Conditions for Parallel Speedup ($K < T$).** Because the internal mapping is a strict contraction ($\rho_1 \in (0, 1)$), the exponential decay of the leading factor $\rho_1^{\frac{K}{m}}$ strictly dominates the history aggregation as $K \to \infty$. The global error bound monotonically decreases with respect to iterations $K$.

Achieving $\epsilon$-precision in strictly fewer iterations than $T$ is mathematically guaranteed if and only if the error bound evaluated at exactly $K = T$ is less than $\epsilon$. Substituting $K = T$ yields:

$$D \cdot \rho_1^{\frac{T}{m}} \cdot \left( 1 + \frac{\rho_2(1 - \rho_1^{\frac{T}{m}})}{1 - \rho_1} \right)^{m-1} < \epsilon \tag{74}$$

Dividing by $D$ isolates the threshold requirement for early convergence:

$$\rho_1^{\frac{T}{m}} \cdot \left(1 + \frac{\rho_2(1 - \rho_1^{\frac{T}{m}})}{1 - \rho_1}\right)^{m-1} < \frac{\epsilon}{D} \tag{75}$$

$\square$

