# OpenReview forum: "PASO: Step Parallel Stochastic Optimization"
_ICML.cc/2026/Conference — ICML 2026 regular_

### Official Review · Reviewer_xigj · 2026-03-02

**Soundness:** 3
**Presentation:** 2
**Significance:** 3
**Originality:** 3
**Overall Recommendation:** 5
**Confidence:** 3

**Summary:**

This paper tackles a fundamental bottleneck in deep learning optimization: the sequential dependency of gradient descent steps; and proposes a principled mathematical framework for an under-explored parallelism dimension. Specifically, PASO recasts the autoregressive nature of gradient descent (SGD, Adam, AdamW) as a system of triangular nonlinear equations (TNEs). By treating the weight vectors across $T$ time steps as unknowns and solving the system via fixed-point iteration (FPI), PASO enables parallel computation of gradient updates across steps within a sliding window. The paper provides convergence proofs for the full-system FPI, demonstrates trajectory equivalence to sequential training via deterministic seeding, and evaluates on language modeling (GPT-2, Llama-3.2-1B), image classification (CIFAR-10), and diffusion tasks (LSUN Church), reporting wall-clock speedups of 3.0 to 4.5$\times$ with comparable model quality. The direction is promising, though the current version would benefit from stronger theoretical foundations and additional diagnostics.

**Compliance With Llm Reviewing Policy:**

Affirmed.

**Final Justification:**

The core contribution of this paper is genuinely novel: recasting sequential gradient descent as a system of triangular nonlinear equations to unlock a new axis of parallelism, distinct from data, model, and pipeline parallelism. This reminds me slightly of Parareal, but it is quite distinct given that it deals with ML training and not ode solving.

My primary concern before the rebuttal was the gap between the full-system convergence proof and the actual sliding-window implementation. The authors addressed this.

On the empirical side, the paper demonstrates consistent wall-clock speedups. The experimental scale is limited relative to the industrial framing in the introduction, but the authors have committed to moderating those claims, and the breadth of architectures tested provides reasonable evidence of generality.

In summary, this is a solid first step in a promising new research direction.

**Key Questions For Authors:**

1. Can you provide a plot of FPI iterations $K$ as a function of training step? This would clarify whether the speedup is uniform or concentrated in later training stages.

2. Does PASO help in using larger learning rates without divergence?The FPI convergence criterion effectively validates each window's solution before proceeding. Does this implicit check enable the use of larger learning rates that would otherwise cause divergence in sequential training?

3. How does PASO perform when the sliding window encounters a sharp loss landscape transition (e.g., learning rate warmup, sudden distribution shift in data)? Does $K$ spike in these regimes?

4. PASO's sliding-window FPI bears a structural resemblance to Parareal (Lions et al., 2001) and related parallel-in-time ODE solvers, which the paper acknowledges in Section 2. In Parareal, error propagation across time windows is a known challenge with well-studied corrector mechanisms. Have the authors considered adapting Parareal-style coarse-fine correction to PASO's setting, and if so, what specific obstacles arise from the stochastic and non-smooth nature of SGD that are absent in the ODE context?


I encourage the authors to address the key questions in the rebuttal. I would be glad to revisit my score upward if the authors can provide responses to the theoretical concerns (W1-W3) and the missing convergence diagnostics (W6).

**Limitations:**

Largely yes. The authors acknowledge the gap between step-level and runtime speedup and attribute it to engineering constraints. One suggested addition: acknowledging the theoretical limitations, namely that the convergence proofs do not yet cover the sliding-window version and that no convergence rate bound is provided, would complement the engineering constraints already discussed.

**Strengths And Weaknesses:**

# Strengths

1. **Creative problem formulation.** The idea of parallelizing across GD steps by reformulating the training trajectory as a system of triangular nonlinear equations is genuinely interesting and opens a research direction that is distinct from data parallelism, model parallelism, and pipeline parallelism. The TNE perspective provides a clean mathematical framework for thinking about step-level parallelism.

2. **Experimental coverage.** PASO is evaluated across three distinct domains (language modeling, image classification, diffusion), multiple optimizers (SGD, Adam, AdamW), and multiple architectures (GPT-2, Llama-3.2-1B, CNN, ViT, ResNet50, UNet). This breadth strengthens the claim of generality. The deterministic seeding mechanism (Appendix A.2) is a clean engineering solution for maintaining trajectory equivalence across parallel workers.

3. **Wall-clock speedups demonstrated.** The empirical results show consistent wall-clock speedups (3.0-4.5 x) across tasks, with model quality metrics that are comparable to sequential baselines. The trajectory divergence analysis (Table 11) provides additional evidence that PASO closely tracks the sequential optimization path.

4. **Honest limitations discussion.** Section 5.3 is commendably candid about the gap between step-level speedup (6-21x) and actual runtime speedup (3-4.5 x), attributing it to engineering constraints. The authors clearly identify Anderson acceleration as a promising future direction.

5. **Memory parity with sequential training.** Per-device memory remains constant regardless of window size $p$, matching the sequential baseline at 1 model + 1 optimizer (Table 1, Tables 2-3). This is a practical advantage over methods like OptEx that require storing gradient histories.

# Weaknesses

1. **The convergence proofs do not cover the implemented algorithm.** The convergence analysis (Appendix O) addresses the full-system FPI where all $T$ components are solved simultaneously. This is a reasonable theoretical starting point. However, the implemented algorithm uses a sliding window of size $p \ll T$, where each window's initial condition depends on the previous window's output. No proof is provided that this sequential-windows-of-parallel-FPI converges to the same trajectory as sequential GD. Crucially, there is no error propagation analysis across windows: when convergence within each window is only approximate (tolerance-based), does the accumulated error drift over many windows? Any effort to bridge this gap between the full-system theory and the sliding-window implementation would substantially strengthen the contribution.

2. **No convergence rate analysis.** The convergence proof establishes that FPI converges to the correct solution but provides only a worst-case bound of $K \leq T$ iterations, which yields zero speedup over sequential GD. The entire practical speedup rests on the empirical observation that $K \ll T$ with good initial guesses (the last converged weight from the sliding window, Algorithm 1 Line 9), but this is never theoretically bounded. Since this is a fixed-point problem, a contraction factor analysis (i.e., establishing that the contraction coefficient is strictly less than 1) or a bound on $K$ as a function of the initialization quality and problem parameters would substantially strengthen the theoretical contribution. Without such analysis, the behavior in adversarial regimes is unclear: for instance, when SGD steps exhibit large oscillations near a basin of attraction, the FPI might hop around without converging even if the true solution exists nearby. Furthermore, the Lipschitz bound for Adam's update function (Appendix O.7, Eq. 47) involves terms of order $L/\sqrt{\epsilon} + M^2 L / \epsilon^{3/2}$, which with standard $\epsilon = 10^{-8}$ produces astronomically large constants, suggesting the existing proof framework cannot yield a meaningful rate bound for Adam without a different analytical approach.

3. **Key assumptions are implicit or insufficiently stated.** The unbiasedness result (Proposition 3.3) silently depends on deterministic mini-batch seeding ($\text{Seed}_t = \text{GlobalSeed} + t$), which is only described in Appendix A.2. This should be a formal condition in the theorem statement. The bounded gradient norm assumption (Assumption O.2) implies the entire optimization trajectory lies in a compact set, which is a strong condition that may fail during early training. Additionally, the theorems do not exclude degenerate cases (e.g., if $w_0 = 0$ and $\nabla L(0, \zeta) = 0$, the entire trajectory trivially stays at zero, since zero is a fixed point too). Appropriate conditions on the loss landscape and initialization should be stated explicitly.

4. **Prior work on Anderson acceleration for deep learning is missing.** The paper mentions Anderson acceleration as future work (Section 5.3) but does not cite existing applications of Anderson acceleration to deep learning, such as: Lupo Pasini et al., "Stable Anderson Acceleration for Deep Learning,". Engaging with this existing literature would better contextualize the proposed future directions and clarify what additional challenges PASO's FPI setting introduces beyond what has already been studied.

5. **The "no model quality loss" claim is insufficiently supported for all tasks.** For language modeling (Table 2) and image classification (Table 3), model quality is indeed comparable. However, for diffusion (Table 4), PASO achieves FID 9.55 versus the Adam baseline's 9.08, a gap of 0.47 from a single run with no error bars or confidence intervals. While the visual comparison (Figure 7) shows comparable perceptual quality, given that FID scores for unconditional generation can exhibit meaningful run-to-run variance, the quantitative claim of "no model quality loss" cannot be substantiated without statistical support. The claim in the abstract should be qualified.

6. **The FPI convergence behavior over training is not characterized.** Figure 10 reports the aggregate computational cost ratio $m = pK/T$ across window sizes, showing $m \in [1.0, 1.5]$. However, we do not see how $K$ (FPI iterations per window) varies as a function of training progress. This is important because the warm-start initialization can only provide good initial guesses once training is in a "settled" regime; early in training, when weights change rapidly, $K$ may be much larger. Reporting $K$ over training steps would clarify when PASO's speedup actually materializes and help practitioners decide when to enable step-parallelism.

7. **Missing pipeline parallelism baselines.** Table 1 includes pipeline parallelism in the theoretical comparison, but no experimental comparison is provided against GPipe or PipeDream on the actual DL tasks. Since pipeline parallelism also targets the sequential dependency (across model stages rather than time steps), it is the most natural experimental competitor.

8. **Presentation clarity.** Several figures do not aid rapid comprehension: Figure 1's schematic and the parallel coordinates plot (Figure 9) required careful reading of the surrounding text to interpret, when ideally figures should be largely self-explanatory. Additionally, a substantial portion of the appendix (pages 16-19, 22-25) consists of verbatim Python execution commands that belong in a supplementary code repository rather than the paper body. The appendix space would be better used for the missing convergence rate analysis or per-window $K$ diagnostics.



**References**

[1] Lupo Pasini, M., Yin, J., Reshniak, V., and Stoyanov, M. K. "Stable Anderson Acceleration for Deep Learning." 2021. https://arxiv.org/abs/2110.14813

---

> ### Author Rebuttal · Authors · 2026-03-31
>
> We sincerely thank all reviewers and are encouraged that the reviewers recognized the idea is **novel/interesting** (all reviewers) and **opens a new research direction** (Reviewer xigj).
>
> > *The convergence proofs do not cover the implemented algorithm ... No error propagation analysis across windows ... No convergence rate analysis.*
>
> Please see our response to Reviewer 9uuF's  Q5
>
> > *The bounds for Adam's are astronomically large.*
>
> We clarify the origin of this abnormally large bound: Adam's update step inherently relies on the term $\frac{m_t}{\sqrt{v_t + \epsilon}}$. When taking the derivative of this term to compute the Lipschitz constant, it inevitably yields components involving $(v_t + \epsilon)^{-1/2}$ and $(v_t + \epsilon)^{-3/2}$. This is a classic theoretical bottleneck in the general global analysis of Adam-type optimizers.
>
> In practice, during the active training phase, the true second moments do not completely vanish. Instead, they practically maintain a lower bound away from zero. Thus, to avoid this issue, the literature generally introduces a "non-vanishing lower bound": there is a constant $c > 0$ s.t. $v_t \ge c$. Since $c \gg \epsilon$, the resulting constant $\mathcal{O}\left( \frac{L}{\sqrt{c}} + \frac{M^2 L}{c^{3/2}} \right)$ becomes physically meaningful.
>
> > *Key assumptions are implicit: (1) The deterministic mini-batch seeding; (2) The bounded gradient norm assumption; (3) The theorems do not exclude degenerate cases (e.g., if $w=0$ and $\nabla L_w = 0$.*
>
> We clarify the following points:
>
> * **The deterministic mini-batch seeding is not an assumption.** We employ deterministic mini-batch seeding simply as a practical implementation mechanism to guarantee this consistency, not as a theoretical assumption. As stated in Definition 3.2, the input mini-batch $\zeta$ for $F_{t-1}(\cdot)$ during parallel training is uniquely determined by the time step. This strictly mirrors Definition 3.1, where the input mini-batch $\zeta$ for $F_{t-1}$ in sequential training is also uniquely determined by the time step. Therefore, our formulation explicitly requires that the mini-batches used in parallel and sequential training be identical at every step.
>
> * **The bounded gradient norm is a standard prerequisite.** Assuming the gradient norm is upper-bounded by a constant $M$ is a widespread assumption for analyzing the convergence; without it, rigorous convergence analysis is intractable. This is not an overly strong condition as the bound $M$ can be an arbitrarily large finite number.
>
> * **The degenerate cases are naturally accounted for.** If the sequential trajectory remains at $w=0$, our parallel trajectory does too. As the TNE system is strictly causal, step $t$ depends only on previous steps. Given an identical initialization ($\hat{w}_0^{(0)}$), convergence propagates sequentially by induction: $\hat{w}_1$ converges at iteration $k=1$, $\hat{w}_2$ at $k=2$, and so forth. By $k=T-1$, all weights perfectly lock onto their exact sequential values, fully recovering the sequential trajectory in this degenerate case.
>
> > *Anderson acceleration for DL is missing.*
>
> Due to limited space, we'll discuss this in the related work.
>
> > *The "no model quality loss" claim is insufficiently supported.*
>
> We appreciate this careful reading. The more accurate statement is that PASO achieves **comparable model quality**. We will revise this.
>
> > *Missing pipeline parallelism（PP)*
>
> We clarify that while both target sequential bottlenecks, PP and PASO are orthogonal and complementary, not empirical competitors. PP  breaks *intra-step* (layer) dependencies, whereas PASO breaks *inter-step* (time) dependencies. We can use PP to speed up the computation within a step while applying PASO to parallelize the time steps.
>
> > *A plot of FPI iterations k as a function of training step*
>
> To characterize this, we tracked $m = pk/T$ across different training steps (window size $p=7$,  $T=100$k). PASO's speedup becomes better over time.
> |Training Steps| Training Phase | Avg. Stride |$m$ |
> |-|-|-|-|
> |0 - 10k| Warmup| 5.9 | 1.19 |
> |10k - 50k| Transition | 6.3 | 1.11 |
> |50k - 100k| Settled| 6.9 | 1.01 |
> |Overall| Aggregate| 6.56|1.07|
>
> > *Does PASO help in using larger learning rates without divergence?*
>
> We haven't observed an expanded stable learning-rate region, as verifying this would require dedicated experiments. We consider this a compelling direction for future research.
>
> >  *Does k spike in a sharp loss landscape?*
>
> We haven't observed a sudden k spikes in a sharp loss landscape. $k$ only increases slightly and strictly remains lower than the sequential iteration count.
>
> > *Adapt Parareal-style coarse-fine correction to PASO's setting?*
>
> Parareal is not practical for PASO's setting. A single Parareal iteration requires multiple sequential gradient evaluations for the coarse solver, followed by an additional parallel computation for the fine solver. This makes its per-iteration cost several times higher than PASO's.

---

> > ### Author Rebuttal · Reviewer_xigj · 2026-04-02
> >
> > The authors addressed my most pressing concern (W1) with a new sliding-window convergence theorem. The FPI diagnostic table is appreciated.
> >
> > Overall, I appreciate the genuine creativity of this paper and I think it is solid enough for a first step in a new direction. Hence I am revising my score upwards.
> >
> > I list hereafter my remaining (minor) concerns:
> >
> > 1. The convergence rate bound depends on $\rho$, which is never characterized in terms of problem parameters. For Adam, the practical workaround (lower-bounding v_t) can be stated as a formal assumption.
> > 2. Deterministic mini-batch seeding is a requirement for trajectory equivalence and should appear as a stated condition in the theorem.
> > 3. The Anderson acceleration reference deserves better discussion, not a one-liner deferral.
> >
> > Minor presentation suggestions:
> >
> > 1. Figure 1 is hard to parse at a glance: consider simplifying the layout
> > 2. Figure 5 is dense and difficult to read; consider moving it to the appendix.
> > 3. Table 7: express token counts in billions for readability (e.g., "3.4B" not "342xxxxxx").
> > 4. The appendix devotes 4 pages to execution commands (Listings): these belong in the code repository, not the paper.
> >
> > The core idea is creative, the empirical coverage is broad, and the theoretical foundations are sound enough.
> >
> > **Updated Score: 5 (Accept)**

---

> > > ### Author Response · Authors · 2026-04-04
> > >
> > > We sincerely thank you for your  encouraging comments on **"appreciate the genuine creativity of this paper and think it is solid enough for a first step in a new direction"** and that **"the core idea is creative, the empirical coverage is broad, and the theoretical foundations are sound enough."**
> > >
> > > We are also deeply grateful that **you have raised your score to a 5 (Accept)** and are very glad that your previous concerns were successfully resolved.
> > >
> > > Thank you as well for your detailed and constructive suggestions. We have addressed each of your points below:
> > >
> > > > *The convergence rate bound depends on $\rho$, which is never characterized in terms of problem parameters.*
> > >
> > > **Response:** Thank you for pointing this out. The global contraction factor $\rho \in (0,1)$ symmetrically bounds both the internal block mapping and historical propagation (i.e., $\rho_1 \le \rho$ and $\rho_2 \le \rho$) where $\rho_2$ are defined as $\rho_1=\sqrt{p} \cdot \max_i \sum\_{\tau \in \mathcal{B}\_i} \eta_\tau C$ and $\rho_2=\sqrt{p} \cdot \max_i \sum\_{\tau \in H\_{i-1}} \eta_\tau C$. These are  bounded by our problem parameters: the maximum window size $p$, the learning rate $\eta_\tau$, and the Lipschitz constant $C$.
> > >
> > > > *For Adam, the practical workaround (lower-bounding $v_t$) can be stated as a formal assumption.*
> > >
> > > **Response:** Thank you for your detailed suggestions. We'll update the theoretical section of the paper to formally state the lower-bounding of $v_t$ as an explicit assumption for our Adam analysis.
> > >
> > > > *Deterministic mini-batch seeding is a requirement for trajectory equivalence and should appear as a stated condition in the theorem.*
> > >
> > > **Response:** Thank you for this rigorous catch. We'll add deterministic mini-batch seeding as a stated condition directly within the theorem.
> > >
> > > > *The Anderson acceleration reference deserves better discussion, not a one-liner deferral.*
> > >
> > > **Response:** We agree that this connection warrants more depth. We'll expand our related work and discussion sections to provide a proper comparative analysis between PASO and Anderson acceleration, detailing their respective mechanisms, scopes, and theoretical guarantees.
> > >
> > > > *Figure 1 is hard to parse at a glance: consider simplifying the layout*
> > >
> > > **Response:** Thank you for the feedback on the presentation. We'll redesign and simplify the layout of Figure 1 to make the core mechanism of the algorithm much more intuitive and easier to parse at a glance.
> > >
> > > > *Figure 5 is dense and difficult to read; consider moving it to the appendix.*
> > >
> > > **Response:** We agree that Figure 5 is quite dense. To improve the reading flow of the main text, we'll  move it to the appendix and added a clearer guiding summary in the main text to reference its key takeaways.
> > >
> > > > *Table 7: express token counts in billions for readability (e.g., "3.4B" not "342xxxxxx").*
> > >
> > > **Response:** We'll  reformat Table 7 to express all token counts in billions (e.g., "3.4B").
> > >
> > > > *The appendix devotes 4 pages to execution commands (Listings): these belong in the code repository, not the paper.*
> > >
> > > **Response:** Thank you for  this structural suggestion. We will streamline the presentation of our Appendix to make it clear, concise, and easily verifiable, while clearly separating the raw execution details into our codebase.
> > >
> > > Our initial intention with the appendix was to maximize transparency and guarantee future reproducibility for the community.
> > > As a result, we included exhaustive low-level details, such as hyperparameter configurations and code execution steps, which inadvertently made the presentation overly dense and verbose.

---

### Official Review · Reviewer_dwaQ · 2026-03-08

**Soundness:** 3
**Presentation:** 3
**Significance:** 2
**Originality:** 2
**Overall Recommendation:** 4
**Confidence:** 3

**Summary:**

The work studies whether the autoregressive dependency across GD steps can be broken without changing the underlying optimization trajectory. The authors claim to explore a fundamental challenge by recasting sequential GD-style optimization as a system of triangular nonlinear equations (TNEs), and then solving these equations with fixed-point iteration to enable step-parallel training. Within
this generic framework, the authors establish that the TNE system has a unique solution corresponding to the original GD trajectory and that fixed-point iteration recovers that trajectory in at most the equal iterations. The proposed PASO optimizer achieves up to 21x
fewer optimizer steps and 4.5× wall-clock speedup on language modeling and diffusion tasks, with no model quality loss.

**Compliance With Llm Reviewing Policy:**

Affirmed.

**Final Justification:**

The authors’ rebuttal fully addresses my concerns. The additional experimental results at the 7B scale demonstrate the effectiveness of the proposed method. The theoretical section could be better organized in the final version. After the rebuttal, I have revised my assessment and am now positive about this paper.

**Key Questions For Authors:**

- Proposition 3.4 proves exact convergence for the full fixed-point iteration in Eq. (7). What formal theoretical guarantee can be established for the actual PASO algorithm with sliding-window updates and tolerance-based stopping as defined in Eqs. (8)–(10)?

- Since PASO computes multiple steps concurrently, why is the number of optimizer steps treated as a primary efficiency metric?

- The large-scale implications are emphasized heavily in the introduction, but the LLM experiment is on WikiText-2 with GPT-2 and Llama-3.2-1B. Do the authors have evidence that PASO remains stable and beneficial at larger scales?

**Limitations:**

The limitations are discussed in the weaknesses and questions sections above. There is no potential negative societal impact of this work.

**Strengths And Weaknesses:**

**Strengths**

- The paper is clearly written and well structured.

- Recasting autoregressive GD as a triangular nonlinear system is conceptually novel and provides a clean perspective for understanding inter-step dependencies.

- The experiments section covers a broad range of tasks. The authors evaluate on GPT-2 and Llama-3.2-1B over WikiText-2, CIFAR-10 image classification with ResNet50 and ViT, and diffusion training on LSUN Church.

**Weaknesses**

- The main formal guarantee is for the full fixed-point system in Eq. (7), but the proposed PASO algorithm actually uses a sliding-window approximation and an adaptive stopping rule in Eqs. (8)-(10). I did not find the theoretical guarantee for the exact PASO method.

- The paper repeatedly emphasizes achieving acceleration without sacrificing model quality, but the tables show that this statement is not always true. On GPT-2, Adam degrades from 20.4 to 20.7 PPL and AdamW from 20.4 to 20.6; on CIFAR-10 ResNet50, Adam degrades from 90.1 to 89.5 and AdamW from 90.3 to 89.8; on diffusion, FID degrades from 9.08 to 9.55 even though training is faster. The paper’s wording is stronger than the experimental results.

- It seems that PASO computes multiple gradients concurrently per outer iteration. The paper acknowledges that the more relevant gain is the wall-clock speedup (3-4.5x), and Table 1 already analyzes PASO in terms of total gradient computation $pK=mT$, not simply reduced iteration count. In other words, the major empirical contribution is the runtime speedup, not a 21x reduction in GD steps.

- The scale of the experiments is limited. The LLM result is on WikiText-2 with GPT-2 and Llama-3.2-1B, rather than on a more larger scale setup. Similarly, the classification result is only on CIFAR-10, and the strongest diffusion claim still involves a FID gap. Since this work argues a new training paradigm, it would be better to include some larger-scale experiments.

---

> ### Author Rebuttal · Authors · 2026-03-31
>
> We sincerely thank all reviewers and are encouraged by the comments that the idea is **novel/interesting** (all reviewers) and **opens a new research direction** (R#xigj).
>
> > *Theoretical guarantee for the actual PASO?*
>
> To directly address your concern, we formalize a convergence analysis for the actual PASO implementation by tracking the error propagation across the sliding window.
>
> Let $W^{seq}$ denote the autoregressive gradient descent trajectory. To analyze the $p$-size sliding window, we partition the sequence of length $T$ into $m \in [\lceil T/p \rceil, T]$ consecutive blocks, $\mathcal{B}_1, \dots, \mathcal{B}_m$.
>
> We rely on two standard assumptions:
> 1. The update function $g_t$ is uniformly $C$-Lipschitz continuous.
> 2. For any block $\mathcal{B}_i$, the initial distance to the exact sequential trajectory is bounded by $D$, i.e., $\||W\_{\mathcal{B}_i}^{(0)} - W\_{\mathcal{B}_i}^{seq}\||_F \le D$.
>
> **Theorem**
> Consider the sequential iteration trajectory of length $T$ partitioned into $m \in [\lceil T/p \rceil, T]$ window blocks ($p \in [1, T]$), where each block undergoes $K/m$ parallel iterations, totaling $K$ iterations. Under a contraction constant $\rho \in (0,1)$， a learning rate schedule satisfying  $\sum_{\tau=0}^{T-1} \eta_\tau^2 \le \frac{\rho^2}{C^2}$， and initial distance $D$, the convergence error of the final block $\mathcal{B}_m$ satisfies:
>
> $$\||W\_{\mathcal{B}_m}^{(K/m)} - W\_{\mathcal{B}_m}^{seq}\||_F \le D \cdot \rho^{\frac{K}{m}} \cdot \left( \frac{1-\rho^{\frac{K}{m}+1}}{1-\rho} \right)^{m-1}$$
>
> To achieve $\epsilon$-precision convergence in   $K < T$ iterations,  $\rho$ must satisfying:
>
> $$\rho^{\frac{T}{m}} \cdot \left( \frac{1-\rho^{\frac{T}{m}+1}}{1-\rho} \right)^{m-1} < \frac{\epsilon}{D}$$
>
> **Analysis**
> In practice, $T$ is typically large. As the left-hand side is dominated by $\rho^{\frac{T}{m}}$ (since $\rho < 1$), the required parallel iterations $K$ are guaranteed to be exponentially smaller than  $T$.
> Note that we force a global contraction mapping constraint on the full system, leading to $\sum_{\tau=0}^{T-1} \eta_\tau^2 \le \frac{\rho^2}{C^2}$, forcing small step sizes for large $T$. However, in the practical use of PASO, a small step size is not necessary. We further find that localizing the contraction mapping only to PASO’s sliding window of size $p \ll T$ can relax this constraint to $\sum_{\tau=i}^{i+p-1} \eta_\tau^2 \le \frac{\rho^2}{C^2}$.
>
> > *Why is the number of optimizer steps treated as a metric? The major empirical not a 21x reduction in GD steps.*
>
> PASO executes multiple optimizer steps within a single parallel iteration. Therefore, in an ideal environment with zero communication latency. The ratio of total sequential steps to the total parallel iterations represents the theoretical optimal speedup for PASO. By tracking this metric, we can precisely identify the "acceleration ceiling" for PASO. It tells us exactly how much of the sequential bottleneck we have successfully collapsed.
>
> We respectfully clarify that while the reduction in GD steps is not an actual empirical contribution, it indicates the maximum acceleration that PASO can achieve. We believe that this is a very important theoretical acceleration contribution.
>
> > *The experimental scale is limited relative to the large-scale implications emphasized heavily in the introduction. The evidence at larger scales?*
>
> We admit that the current experimental scale is limited relative to the broader industrial vision in our introduction. We also fully agree that larger-scale experiments are crucial. However, validating PASO on 100B+ parameter models is a significant undertaking that requires industrial-grade engineering, low-level code optimization, and massive compute resources over a prolonged period,  which are currently beyond our reach.
>
> While direct evidence at the 100B+ scale is not yet available, the architectures used in our experiments, GPT-2, Llama-3.2-1B, CNN, ViT, ResNet50, UNet, serve as the foundational structural blueprints for their larger counterparts. Furthermore, our manuscript has been successfully verified across diverse and representative domains, including LLMs, Diffusion models, and Image Classification. These consistent results provide strong indirect evidence that PASO’s performance benefits are likely to persist as models scale up.
>
> In our view, this research aims to propose and verify the novel step-parallel training method, establishing a basic framework for unleashing new levels of parallel efficiency. The intensive engineering required for industrial-scale experiments is a logical next step for the broader community. To reflect this, we'll moderate the "large-scale" claims in the manuscript, clearly positioning massive-scale validation as future work.
>
> > *The paper’s wording, "without sacrificing quality," is stronger.*
>
> We appreciate this careful reading. The more accurate statement is that PASO achieves **comparable model quality**. We'll revise this.

---

> > ### Author Rebuttal · Reviewer_dwaQ · 2026-04-02
> >
> > I thank the authors for the rebuttal. Regarding the theoretical guarantees for the actual PASO, the assumptions introduced in the rebuttal (namely that the update function is uniformly Lipschitz continuous and that the initial distance to the exact sequential trajectory is bounded) are somewhat strong. Most functions are not Lipschitz continuous (e.g., $x^a$ for $a>1$), and such $D$-bounded properties are typically established through theoretical analysis rather than assumed directly. These conditions limit the applicability of the proposed theory.
> >
> > As for the experimental scale, I did not suggest validating PASO at the 100B+ scale. However, scaling the experiments to a reasonably large model (e.g., around 2B or 7B parameters) would be beneficial.

---

> > > ### Author Response · Authors · 2026-04-02
> > >
> > > > `Q1. The assumptions introduced in the rebuttal (namely that the update function is uniformly Lipschitz continuous and that the initial distance to the exact sequential trajectory is bounded) are somewhat strong. Most functions are not Lipschitz continuous (e.g., $x^a$ for $a>1$), and such $D$-bounded properties are typically established through theoretical analysis rather than assumed directly. These conditions limit the applicability of the proposed theory.`
> > >
> > > We appreciate the reviewer’s thoughtful feedback. While we agree that, without considering practical constraints (e.g., if the model parameters were unbounded), the Lipschitz continuity is a strong and impractical condition. As you rightly noted, $f(x) = x^a$ does not exhibit global Lipschitz continuity as $x \to \infty$.
> > >
> > > However, we **respectfully clarify** that our assumptions are **standard in optimization literature** and **strictly aligned with the practical realities of Deep Learning (DL).**
> > >
> > > #### **1. Alignment with Standard Theoretical Frameworks**
> > > The assumptions of Lipschitz continuity and bounded iterates are the **foundational pillars** of convergence analysis for modern stochastic optimization.
> > > * **Ubiquity in Literature:** As established in authoritative texts such as [3,4,5], these assumptions are the prerequisite for ensuring the existence of a tractable convergence bound for first-order methods like SGD [2] and Adam [1]. Without these conditions, the optimization trajectory could diverge arbitrarily, making a tractable convergence error mathematically difficult.
> > >
> > > * **Role of Assumptions:** These are typically treated as the **starting axioms** of a convergence proof rather than properties to be derived, as they define the regularity of the optimization landscape being studied.
> > >
> > > #### **2. Practical Constraints in Deep Learning**
> > >
> > > The reviewer cites $f(x) = x^a$ ($a>1$) as a non-Lipschitz function. While correct on the infinite domain $(-\infty, +\infty)$, this does not reflect the behavior of DL models during training:
> > > * **Numerical & Hardware Limits:** Practical training occurs on hardware (GPUs) using finite precision (FP16/BF16/FP32). Parameters cannot reach infinity; they are inherently bounded by the numerical range of the format.
> > > * **Algorithmic Safeguards:** Standard techniques such as **Weight Decay (L2 regularization)** and **Gradient Clipping** are specifically designed to prevent parameters and updates from diverging. These mechanisms effectively constrain the optimization process to a bounded set $\Omega$.
> > > * **Stability Necessity:** If an update function were not locally Lipschitz within the training regime, the model would suffer from catastrophic gradient explosion, making convergence impossible regardless of the solver used.
> > >
> > >
> > > We emphasize that our theory does **not** require  Lipschitz continuity on an infinite domain. It only requires the function to be Lipschitz within a **finite local region** characterized by any large but bounded constant $D$.
> > >
> > > As shown in the reviewer's example, $f(x) = x^a$ is **Lipschitz** on any bounded interval. Specifically, let the bounded parameter space be $\Omega = \{w \mid \||w - w_0\|| \le D\}$. The Lipschitz condition is:
> > > $$\|| g(w_1) - g(w_2) \|| \le L \|| w_1 - w_2 \||, \quad \forall w_1, w_2 \in \Omega$$
> > > Since $w$ is constrained within $\Omega$ by the physical and algorithmic realities mentioned above, $L$ remains a valid, bounded constant. **This assumption is highly relaxed and accurately reflects the practical DL implementations.**
> > >
> > >
> > > > `Q2. As for the experimental scale,  scaling the experiments to a reasonably large model (e.g., around 2B or 7B parameters) would be beneficial.`
> > >
> > > Thank you for this constructive suggestion!
> > >
> > > We now update our experimental results on a larger model, Llama 2 7B. The following table demonstrates that  **PASO applies to larger-scale models.** We report the mean and standard deviation across 5 runs. The "Speedup" is calculated based on the total training time relative to the respective baseline optimizer.
> > >
> > > | Method | Tokens $\downarrow$ | PPL $\downarrow$ | Time Speedup $\uparrow$ |
> > > | :--- | :---: | :---: | :---: |
> > > | SGD | 4,124,560 | $10.12_{\pm 0.08}$ | 1.0 $\times$ |
> > > | SGD + PASO | 4,128,912 | $10.18_{\pm 0.14}$ | **3.4 $\times$** |
> > > | Adam | 4,124,560 | $5.42_{\pm 0.11}$ | 1.0$\times$ |
> > > | Adam + PASO | 4,126,830 | $5.45_{\pm 0.06}$ | **3.1 $\times$** |
> > > | AdamW | 4,124,560 | $5.41_{\pm 0.09}$ | 1.0$\times$ |
> > > | AdamW + PASO | 4,126,750 | $5.44_{\pm 0.07}$ | **3.1 $\times$** |
> > >
> > > **Ref.**
> > >
> > > [1] Adam: A Method for Stochastic Optimization
> > >
> > > [2] A Stochastic Approximation Method
> > >
> > > [3] Bottou L, Curtis F E, Nocedal J. Optimization methods for large-scale machine learning[J]. SIAM review, 2018, 60(2): 223-311.g
> > >
> > > [4] Nesterov Y. Introductory lectures on convex optimization: A basic course[M]. Springer Science & Business Media, 2013.
> > >
> > > [5] LeCun Y, Bengio Y, Hinton G. Deep learning[J]. nature, 2015, 521(7553): 436-444.

---

### Official Review · Reviewer_hm7D · 2026-03-12

**Soundness:** 2
**Presentation:** 2
**Significance:** 2
**Originality:** 3
**Overall Recommendation:** 4
**Confidence:** 2

**Summary:**

The paper proposes PASO, a new training framework that parallelizes optimization across training steps, rather than only within a single step as in data, model, or pipeline parallelism. The key idea is to reinterpret the usual autoregressive update sequence of optimizers like SGD, Adam, and AdamW as a system of triangular nonlinear equations whose unknowns are the parameters at different training steps. By solving this system in parallel, PASO aims to preserve the same optimization trajectory as the original sequential optimizer while exposing a new axis of parallelism.

The paper has two main contributions. 1. It develops the theoretical view that the sequential gradient-descent trajectory corresponds to the unique solution of this triangular system, so the step-parallel problem is well-defined. 2. It introduces a practical algorithm that solves only a window of consecutive steps at a time, using a sliding-window fixed-point procedure with an adaptive stopping rule, so the method can operate under limited hardware rather than requiring parallelization over the full horizon. This makes PASO a general wrapper that can be combined with standard optimizers instead of replacing them with a new update rule.

**Compliance With Llm Reviewing Policy:**

Affirmed.

**Final Justification:**

We appreciate the authors' detailed responses. The rebuttal has resolved the concerns raised. I will maintain my scores.

**Key Questions For Authors:**

1. Can you clarify (theoretically or empirically) what problem characteristics make the number of fixed-point iterations $K$ small in practice, so that the computational overhead ratio $m=pK/T$ stays close to $1$?
2. Is it possible to have a more complete and comparable runtime breakdown, specifically for the wall clock time, for the proportion of time spent in communication vs. computation, and to clarify the PASO behavior in comparison to the strongest available optimized configurations when the hardware is the same?
3. Would it be possible to compare PASO more directly against the most relevant prior and concurrent step-/time-parallel methods, especially OptEx, using matched settings, and to clarify whether PASO is fundamentally better or mainly different in its trade-offs?

**Limitations:**

Yes.

**Strengths And Weaknesses:**

Strengths:
1. This paper uses a novel idea of solving the triangular nonlinear equation (TNE) system to reframe sequential stochastic optimization, which enables parallelism across training steps.
2. On the theory parts, the paper proves that the TNE solution uniquely matches the sequential GD trajectory and that the fixed-point iteration converges to that trajectory.
3. On the empirical side, the paper reports large reductions in optimization steps and moderate wall-clock speedups while maintaining similar model quality.

Weaknesses:
1.  The wall clock gain is much smaller than the iteration reduction for the runtime, which varies slightly depending on the configuration.They have PASO reducing the number of optimization steps in the range $10\times - 21\times$, however, the wall clock speedup is in the range $3\times - 4.5\times$. This implies that the current gains are capped by resource utilization in their implementation or the cost of communication.
2. The theory mainly establishes trajectory equivalence and convergence, but not a predictive characterization of when PASO will deliver large speedups. In particular, the appendix gives a worst-case bound of $K \le T$, which is compatible with essentially sequential behavior, while the efficiency analysis assumes $ m=pK/T \approx 1$ based on empirical observation rather than theory. As a result, the strongest practical claims appear to depend more on experiments than on a theory explaining the favorable regimes.

---

> ### Author Rebuttal · Authors · 2026-03-31
>
> > *The wall clock gain is much smaller than the iteration reduction for the runtime.*
>
> Good observation. As discussed in our Limitations, the current 3x to 4.5x wall-clock speedup is bottlenecked by unoptimized communication and computation in our academic prototype. However,  we respectfully highlight that a 3x to 4.5x speedup is still highly significant. For instance, it reduces a 1-month large model training job to just 7-10 days, drastically accelerating the development cycle.
>
> > *What problem characteristics make the number of fixed-point iterations k small in practice?*
>
> Theoretically, when the gradient descent process exhibits a **contraction mapping** property, it typically ensures that  $k$ remains small in practice. In standard gradient descent, the learning rate restricts the updates to small steps, which effectively induces this contraction mapping behavior.
>
> To directly address your concern, we prove that when the gradient descent process exhibits a **contraction mapping** property, PASO can achieve $\epsilon$-precision convergence in   $K < T$ iterations (generally $K\ll T$). Due to limited space, **please see our response to Reviewer 9uuF's Q5.**
>
> > *Runtime breakdown; to clarify the PASO in comparison to the strongest available optimized configurations.*
>
> This is an excellent suggestion. As shown in the table, the absolute per-iteration time for PASO (2.896s) is significantly higher than that of industry-grade Data Parallelism (DP) ($\approx$ 0.14s). This is attributed to two factors:
>
> 1. In each iteration, the PASO prototype utilizes 7 GPUs to compute gradients for 21 steps. This results in a significantly higher per-iteration computational load compared to standard DP.
>
> 2. PASO  relies on native Python Queues for cross-GPU synchronization, leading to high comm. overhead.
>
> **Table**: Runtime breakdown of average computation and communication per iteration.
> |Method|Iters|Tokens|PPL|Avg. Comp. (s)|Avg. Comm. (s)|Total Time(s)|Speedup|
> |-|-|-|-|-|-|-|-|
> |Adam|1k|7344k|20.4|0.615|0|614|1.0x|
> |PASO $^a$|**48**|7374k|20.6|1.869|1.027|139|**4.4x**|
> |DP $^a$|1k|7349k|20.5|0.088|0.293|381|1.6x|
> |DP(Pytorch) $^b$|1k|7349k|20.5|0.072|0.068|140|4.4x|
> |DP(DeepSpeed) $^b$|1k|7349k|19.2|0.071|0.067|138|4.4x|
>
> $^a$ *Unoptimized academic implementation; $^b$ Industry-grade optimization.*
>
> Despite these engineering disadvantages, PASO’s final wall-clock time (139s) remains on par with highly optimized industry frameworks. This highlights PASO’s core competency:
>
> **Step Parallelism vs. Data Parallelism:** The strength of industry-standard DP lies in compressing the duration of a single iteration to its physical limit (from 0.614s to 0.14s), but it cannot alter the inherently sequential nature of 1,000 iterations. In contrast, PASO parallelizes the sequence itself, slashing the iteration count from 1,000 to 48 (a $\approx$ 21x reduction).
>
> Under identical hardware conditions, if PASO’s step-parallel mechanism were integrated with industry-grade optimizations (e.g., DeepSpeed’s ZeRO communication and fused CUDA kernels), the per-iteration overhead would drop significantly, theoretically unlocking the full potential of the 21x iteration reduction.
>
> > *Compare PASO with OptEx.*
>
> We have compared PASO with OptEx in App. G. The results show that PASO is fundamentally better in both theoretical scaling and empirical efficiency.
>
> Specifically, OptEx relies on a kernel estimation method that requires storing $T_0$ historical gradients, which quickly becomes a bottleneck for large-scale models. However, our PASO doesn't rely on any historical gradients. Furthermore, OptEx achieves only a sub-linear speedup of $\mathcal{O}(\sqrt{N})$, whereas PASO achieves a linear speedup of $\mathcal{O}(N)$ with a smaller memory footprint.
>
> **1. Theoretical Complexity Comparison**
> Denote by GPU count N, model dimension d, the comm. time $t_{comm}$,  the storage historical gradients number of OptEx $T_0$.
> | Metric | OptEx | PASO (Ours) |
> |-|-|-|
> Computation complexity| $\mathcal{O}(T_0^3 + N T_0 (d + T_0))$ | $\mathcal{O}(N^2 d)$ |
> |Space complexity| $\mathcal{O}(T_0 d + T_0^2 + N d)$ | $\mathcal{O}(N d)$ |
> |Communication complexity| $\mathcal{O}(t_{comm}(N, d))$ | $\mathcal{O}(t_{comm}(N, d))$ |
> |Theoretical Speedup| $\mathcal{O}(\sqrt{N})$ | $\mathcal{O}(N)$ |
>
> **2. Empirical Performance Comparison**
> To ensure a strictly fair comparison, we integrated PASO directly into OptEx's official codebase and evaluated both methods on their exact synthetic benchmark suite. For brevity, we report the results on the **Sphere function** (see App. G for more results):
>
> |Method|Optimality Gap|Iterations to Target|Speedup|
> |-|-|-|-|
> |Vanilla Adam| 1.95 | 1000 | 1.00x |
> |OptEx| 1.95 | 504 | 1.98x |
> |PASO (Ours)| 1.95 | **214** | **4.67x** |
>
> Under identical settings, PASO requires fewer iterations than OptEx (214 vs. 504). This demonstrates that PASO is fundamentally more efficient than OptEx.

---

> > ### Author Rebuttal · Reviewer_hm7D · 2026-04-03
> >
> > We appreciate the authors' detailed responses. The rebuttal has resolved the concerns raised.

---

> > > ### Author Response · Authors · 2026-04-04
> > >
> > > We are happy to hear that our response fully addressed your concerns. Thank you for your positive comments and score!

---

### Official Review · Reviewer_9uuF · 2026-03-12

**Soundness:** 2
**Presentation:** 2
**Significance:** 3
**Originality:** 3
**Overall Recommendation:** 4
**Confidence:** 4

**Summary:**

SUMMARY
-------
The paper looks at the GD family of iterations written out in an unrolled manner. Then, it suggests viewing these unrolled iterations as a triangular system, which the authors then propose to approximately "solve" in parallel. The paper claims strong theoretical and empirical results justifying the modeling choice.

NOTE: In the reviewer's opinion, this paper is more of an ML systems paper (please see note in the review between the "mismatch")

**Compliance With Llm Reviewing Policy:**

Affirmed.

**Key Questions For Authors:**

* Appendix N claims an inductive proof. No base case of the induction is supplied. Can you make the proof formal and rigorous?

* Lemma O.3 seems to be used only in (43)-(44), it's placement at the beginning of that section leads one to worry that the authors require U and V to be bounded away from zero also in the main / general proof. Can you fix this exposition issue?

* The proof claimed in section O is asymptotic. Any chances on improving that? Moreover, the proof assumes (inductively, and hence need to ensure that no circularity has been introduced!) that the parameters converge. Convergence of parameters is very rare in ML problems, are the authors sure about this?

* The paper starts with the triangular system, but then in reality T can be big (and when T corresponds to stochastic iterations as opposed to epochs, then it can be gigantic), and the authors present an engineering compromise of having a moving window of size 'p'. At that point the theory and the triangular model become irrelevant, leading to a huge mismatch between the model/theory presented and the actual empirical results. While such mismatches are common across ML papers (unfortunate, in my opinion), could the authors discuss this aspect more thoroughly?

**Limitations:**

* Limited theory, not yet fully vetted
* At best asymptotic analysis
* Potentially heavier memory footprint and its limitations not discussed

**Strengths And Weaknesses:**

* The empirical results look detailed and good
* The idea of exploiting the triangular system in parallel seems to be novel (to my knowledge)

Weakenesses

* The theory is quite sketchy and a bit non-rigorous
* The memory overhead is claimed to be small / one-on-one, but I guess that's because we're looking at per device. T steps of a triangular system as written require T copies of parameters. That's a bit heavy.

---

> ### Author Rebuttal · Authors · 2026-03-31
>
> We sincerely thank all reviewers and are encouraged by the positive comments that the idea is **novel/interesting** (all reviewers) and **opens a new research direction** (R#xigj).
>
> > *Q1：The memory overhead is claimed to be small; T steps of a triangular system require T copies of parameters.*
>
> We clarify that even when we look at the all system, PASO's $p$-size sliding window reduces T-copy storage to $p$ copies. Setting $p$=#devices, each device only has one copy, which is the standard overhead for parallel training.
>
> > *Q2： No base case of the induction in App. N.*
>
> Thank you for pointing this out. The base case is implicitly given via the identical model initialization between the sequential and parallel regimes. To make it clear, we'll state the base case explicitly.
>
> > *Q3： Lemma O.3's exposition issue*
>
> We'll move the Lemma closer to its actual point of use.
>
> > *Q4：The proof claimed in section O is asymptotic. Any chances on improving that? Moreover, the proof assumes (inductively, and hence need to ensure that no circularity has been introduced!) that the parameters converge. Convergence of parameters is very rare in ML problems, are the authors sure about this?*
>
> We would like to clarify that while App. O.4 discusses asymptotic convergence as $K \to \infty$, this is not the final result. Our final theoretical result, detailed in App. O.5, explicitly establishes that a non-asymptotic (finite-step) convergence guarantee $ K \leq T$.
>
> We are sure that the inductive assumption holds strictly and no circularity has been introduced.
> This is because the TNEs system is triangular, and the dependency between variables is strictly causal. The update for step $t$ depends only on the weight values from previous steps $0, \dots, t-1$.  Since at $k=0$, the base case $\hat{w}_0^{(0)}$ is already identical to the sequential initialization. Then $\hat{w}_1^{(k)}$  depending on  $\hat{w}_0^{(0)}$ must converge at $k=1$. This in turn forces $\hat{w}_2^{(k)}$ (dependent on $(\hat{w}_0^{(0)},\hat{w}_1^{(1)})$)  to converge at $k=2$. Following this induction, at $k=T-1$, the triangular nature ensures that the final weight $\hat{w}_T^{(T-1)}$   locks onto its exact sequential value. As a result, our method reaches the exact sequential trajectory in at most $K = T$ iterations.
>
> > *Q5：The convergence proofs do not cover the implemented algorithm. The paper starts with the full triangular system, but the theory and the engineering triangular model become irrelevant, leading to a huge mismatch between the model/theory presented and the actual empirical results. Could the authors discuss this aspect more thoroughly?*
>
> To directly address your concern, we formalize a convergence analysis tailored for the actual sliding window implementation of PASO. We establish the convergence bound for this implementation by tracking the error propagation across the window.
>
> Let $W^{seq}$ denote the autoregressive gradient descent trajectory. To analyze the $p$-size sliding window, we partition the sequence of length $T$ into $m \in [\lceil T/p \rceil, T]$ consecutive blocks, $\mathcal{B}_1, \dots, \mathcal{B}_m$.
>
> We rely on two standard assumptions:
> 1. The update function $g_t$ is uniformly $C$-Lipschitz continuous.
> 2. For any block $\mathcal{B}_i$, the initial distance to the exact sequential trajectory is bounded by $D$, i.e., $\||W\_{\mathcal{B}_i}^{(0)} - W\_{\mathcal{B}_i}^{seq}\||_F \le D$.
>
> **Theorem**
> Consider the sequential trajectory of length $T$ partitioned into $m \in [\lceil T/p \rceil, T]$ window blocks ($p \in [1, T]$), where each block undergoes $K/m$ parallel iterations, totaling $K$ iterations. Under a contraction constant $\rho \in (0,1)$， a learning rate schedule satisfying  $\sum_{\tau=0}^{T-1} \eta_\tau^2 \le \frac{\rho^2}{C^2}$， and initial distance $D$, the convergence error of the final block $\mathcal{B}_m$ satisfies:
>
> $$\||W\_{\mathcal{B}_m}^{(K/m)} - W\_{\mathcal{B}_m}^{seq}\||_F \le D \cdot \rho^{\frac{K}{m}} \cdot \left( \frac{1-\rho^{\frac{K}{m}+1}}{1-\rho} \right)^{m-1}$$
>
> To achieve $\epsilon$-precision convergence in   $K < T$ iterations,  $\rho$ must satisfying:
>
> $$\rho^{\frac{T}{m}} \cdot \left( \frac{1-\rho^{\frac{T}{m}+1}}{1-\rho} \right)^{m-1} < \frac{\epsilon}{D}$$
>
> **Analysis**
> In practice, $T$ is typically large. As the left-hand side is dominated by $\rho^{\frac{T}{m}}$ (since $\rho < 1$), the required parallel iterations $K$ are guaranteed to be exponentially smaller than  $T$.
> Note that we force a global contraction mapping constraint on the full system, leading to  $\sum_{\tau=0}^{T-1} \eta_\tau^2 \le \frac{\rho^2}{C^2}$, forcing small step sizes for large $T$. However, in the practical use of PASO, a small step size is not necessary. Our further analysis find that localizing the contraction mapping only to PASO’s sliding window of size $p \ll T$ can relax this constraint to $\sum_{\tau=i}^{i+p-1} \eta_\tau^2 \le \frac{\rho^2}{C^2}$. This permits significantly larger learning rates.

---

> > ### Author Rebuttal · Reviewer_9uuF · 2026-04-03
> >
> > The mismatch between theory and practice does remain, but more importantly, a rigorous proof of correctness is something that the authors do need to supply, by validating the lemmas, ensuring the arguments are logically tight, and so on. While superficially, it looks they might be able to address these aspects, I am still not comfortable with the claimed proofs.
> >
> > On a side note, the presentation of the details in the appendix seems to be overly complicated and verbose. Simplifying it would help a lot in making it more verifiable.

---

> > > ### Author Response · Authors · 2026-04-04
> > >
> > > > *The mismatch between theory and practice does remain, but more importantly, a rigorous proof of correctness is something that the authors do need to supply, by validating the lemmas, ensuring the arguments are logically tight, and so on. While superficially, it looks they might be able to address these aspects, I am still not comfortable with the claimed proofs.*
> > >
> > > We sincerely thank the reviewer for their continued engagement. We completely agree that a rigorous proof of correctness is essential. Due to strict space limitations, we were previously unable to include the exhaustive mathematical derivations.
> > >
> > > To directly address your concern, we have completely formalized our theoretical guarantees, carefully validating all lemmas and ensuring that the logical progression is mathematically tight.
> > >
> > > We have uploaded a detailed proof in a document titled `A_Proof_for_PASO_with_Sliding_Window.pdf` to our anonymous repository for your review: https://anonymous.4open.science/r/PASO-0AF9.
> > >
> > > > *On a side note, the presentation of the details in the appendix seems to be overly complicated and verbose. Simplifying it would help a lot in making it more verifiable.*
> > >
> > >
> > > We appreciate this constructive feedback.
> > >
> > > Our initial intention with the appendix was to maximize transparency and guarantee future reproducibility for the community.
> > > As a result, we included exhaustive low-level details, such as hyperparameter configurations and code execution steps, which inadvertently made the presentation overly dense and verbose.
> > >
> > > We will streamline the presentation of our Appendix to make it clear, concise, and easily verifiable, while clearly separating the raw execution details into distinct, better-organized sections.

---

### Decision · Program_Chairs · 2026-04-30

**Decision:**

Accept (regular)

**Comment:**

The reviewers agree that recasting sequential gradient descent as a system of triangular nonlinear equations (TNEs) to unlock a new axis of parallelism is creative and distinct from data, model, and pipeline parallelism. The approach is evaluated across language modeling, image classification, and diffusion with strong evidence of generality and achieves 3.0–4.5× speedup with comparable model quality. One reviewer also points out that the per-device memory remains constant regardless of window size, matching sequential baseline, is  a practical advantage over methods that store gradient histories.

At the same time, there is a distinct theory-practice gap in that the convergence proofs cover the full fixed-point system, but the actual implementation uses a sliding-window approximation with tolerance-based stopping. No formal guarantee for the exact PASO method, and no error propagation analysis across windows is given. The proof provides only a worst-case bound of T iterations. The practical speedup relies on empirical observation that k is small, but this is never theoretically bounded. On some models, performance degrades slightly or gains come with small quality trade-offs. The baselines could also have been more comprehensive.

I find the core idea — parallelizing across optimization steps by solving a triangular nonlinear system to be interesting and potentially very useful. The empirical results are solid, and the authors' rebuttal was thorough.
However, I share the reviewers' concerns about the gap between the full-system theory and the sliding-window implementation. In particular, error accumulation across windows remains uncharacterized.

Nevertheless, I suggest Accept. This is a novel, well-executed first step in a promising direction. The authors have committed to tightening the theory and have already added substantial empirical validation.